# Characterization and Learning of Causal Graphs from Hard Interventions

**Zihan Zhou**[*]
Department of Computer Science
Johns Hopkins University
zzhou150@jhu.edu

**Muhammad Qasim Elahi**[*]
School of Electrical and Computer Engineering
Purdue University
elahi0@purdue.edu

**Murat Kocaoglu**
Department of Computer Science
Johns Hopkins University
mkocaoglu@jhu.edu

## Abstract

A fundamental challenge in the empirical sciences involves uncovering causal structure through observation and experimentation. Causal discovery entails linking the conditional independence (CI) invariances in observational data to their corresponding graphical constraints via d-separation. In this paper, we consider a general setting where we have access to data from multiple experimental distributions resulting from hard interventions, as well as potentially from an observational distribution. By comparing different interventional distributions, we propose a set of graphical constraints that are fundamentally linked to Pearl's do-calculus within the framework of hard interventions. These graphical constraints associate each graphical structure with a set of interventional distributions that are consistent with the rules of do-calculus. We characterize the interventional equivalence class of causal graphs with latent variables and introduce a graphical representation that can be used to determine whether two causal graphs are interventionally equivalent, i.e., whether they are associated with the same family of hard interventional distributions, where the elements of the family are indistinguishable using the invariances from do-calculus. We also propose a learning algorithm to integrate multiple datasets from hard interventions, introducing new orientation rules. The learning objective is a tuple of augmented graphs which entails a set of causal graphs. We also prove the soundness of the proposed algorithm.

## 1 Introduction

Understanding the behavior of complex systems through their causal relationships is a fundamental problem in science. Researchers collect data and perform experiments to analyze how specific phenomena arise or to investigate the structure and function of underlying systems, whether social, biological, or economic [Hünermund and Bareinboim, 2023, Petersen et al., 2024, Sanchez et al., 2022]. Causal discovery focuses on identifying causal relationships from both observational and interventional data [Pearl, 1995, Spirtes et al., 2001, Peters et al., 2017]. A widely used approach for causal discovery models the underlying system as a causal graph, represented by a directed acyclic graph (DAG), where nodes denote random variables and directed edges between nodes ($A \rightarrow B$) signify causal relations [Pearl, 2009, Spirtes et al., 2001].

---

[*]Equal Contribution

39th Conference on Neural Information Processing Systems (NeurIPS 2025).

Causal discovery involves deriving constraints from data to infer the underlying causal graph. However, in practice, these constraints are rarely enough to identify the exact causal graph. Instead, they typically define a set of graphs consistent with the data, collectively referred to as an equivalence class (EC) [Ali et al., 2012, Meek, 2013a]. Conditional independence (CI) relations are the primary markers of the underlying causal structure and are used to define an equivalence class. These fundamental probabilistic invariances have been widely explored within the framework of graphical models [Pearl, 1995, Peters et al., 2017]. Conditional independencies (CIs) are a powerful tool and serve as the foundation for many causal discovery algorithms, e.g., PC and FCI [Spirtes et al., 2001].

When only observational data is available, the Markov equivalence class (MEC) comprises all causal graphs that exhibit the same set of conditional independences (CIs) among the measured variables, as defined by the d-separation criterion [Verma and Pearl, 1992]. The availability of interventional (experimental) data allows us to reduce the size of the equivalence class, potentially facilitating the recovery of the causal graph [Hauser and Bühlmann, 2012, Kocaoglu et al., 2017]. Hard and soft interventions are two methods for manipulating variables. A hard intervention directly sets a variable to a fixed value, removing natural dependencies, while a soft intervention alters the mechanism through which the parents of a variable influence the target variable. While soft interventions are more common in biology as, e.g., in gene knockout experiments [Meinshausen et al., 2016] since we do not have precise control of mechanisms, in computer systems hard interventions are feasible and have been used for learning causal relations, e.g., in microservice architectures [Wang et al., 2023].

Although a hard intervention can be seen as a special case of a soft one, it can be more informative in the presence of latent variables in many cases. For instance, consider the causal graph $\mathcal{D}_1 = \{X \to Z \to Y, Z \leftrightarrow Y\}$. We use the bidirected edge to represent a latent confounder between two nodes. A hard intervention on $Z$ breaks the inducing path[1] $\langle X, Z, Y \rangle$, which implies that after a hard intervention on $Z$, the variables $X$ and $Y$ are no longer dependent. In contrast, in the case of a soft intervention, the incoming edges to $Z$ are not removed, so a soft intervention will not break the inducing path. It is also worth noting that hard interventions may change d-separation statements non-locally, as seen here between $X, Y$ after $do(z)$, which the existing representations of interventional Markov equivalence classes cannot encode. Now consider another graph $\mathcal{D}_2 = \{X \to Z \to Y, Z \leftrightarrow Y, X \to Y\}$, which is the same as $\mathcal{D}_1$ with the additional edge $X \to Y$. With access to a hard intervention on $Z$, we can distinguish these graphs; however, with a soft intervention, one cannot differentiate between them. This example demonstrates that hard interventions can be more informative in the presence of latent variables and narrow down the search space more effectively than soft ones. We conducted empirical experiments in Appendix F to compare hard and soft interventions in learning the causal graph with latents. The results verify this observation. A fundamental question is how can we extract as much causal knowledge as possible from a collection of hard interventional datasets. To the best of our knowledge, this problem has been open before this work.

Motivated by this, our paper considers a general setting where multiple experimental distributions resulting from hard interventions are available alongside (optionally) observational distributions. Prior work has focused on characterizing the $\mathcal{I}$-Markov equivalence class through distributional invariances both within and across a set of observational and interventional distributions [Hauser and Bühlmann, 2012, Yang et al., 2018]. The closest work to ours is Kocaoglu et al. [2019]. However, they deal with soft interventions, whereas we consider a setting where experimental data comes from hard interventions. This can lead to more invariances that can be inferred from the experimental data compared to the soft intervention case, which can potentially further reduce the size of $\mathcal{I}$-Markov equivalence class. However, the existing work cannot utilize these additional invariances.

We propose using *do-constraints* with hard interventions, a concept that emerges from comparing observational and experimental distributions, extending Pearl's do-calculus to uncover new structural insights in causal graphs [Kocaoglu et al., 2019, Jaber et al., 2020]. They emerge as the converse of the causal calculus developed by Pearl [1995]. These constraints, distinct from traditional conditional independence (CI) relations, are derived by contrasting distributions such as $P(y|x)$ and $P(y|do(x))$ through a *do-see test*. When differences are detected, they reveal open backdoor paths in the graph, aiding structure learning. We leverage *F-nodes*, introduced by Pearl, to explicitly encode intervention effects in augmented graphs [Pearl, 1993]. These nodes make the effects of interventions visible within the graph, enabling the application of do-calculus tests and capturing key structural knowledge

---

[1]Inducing paths are paths between non-adjacent variables that cannot be blocked by conditioning on any subset of observed variables. They only exist in the presence of unobserved confounders.

such as the existence of a backdoor path from $X$ to $Y$ when $F_X$ is not d-separated from $Y$ given $X$. Such augmented representations, widely used in inference and identification, highlight the utility of do-constraints alongside CI relations for learning causal structures [Yang et al., 2018, Kocaoglu et al., 2019, Mooij et al., 2020].

We say that a set of interventional distributions satisfies the $\mathcal{I}$-Markov property with respect to a graph if these distributions adhere to the invariance constraints imposed by the causal calculus rules of that graph. We first extend the causal calculus rules to operate between arbitrary sets of hard interventions. We say that two causal graphs, $\mathcal{D}_1$ and $\mathcal{D}_2$, are $\mathcal{I}$-Markov equivalent if the set of distributions that are $\mathcal{I}$-Markov to both $\mathcal{D}_1$ and $\mathcal{D}_2$ is the same. Using the augmented graph, we identify a graphical condition that is both necessary and sufficient for two Causal Bayesian Networks (CBNs) with latents to be $\mathcal{I}$-Markov equivalent under the framework of hard interventions. Finally, we propose a sound algorithm for causal discovery from hard-interventional datasets. Our main contributions can be summarized as follows:

- We propose a characterization of $\mathcal{I}$-Markov equivalence between two causal graphs with latent variables for a given intervention set $\mathcal{I}$, based on a generalization of do-calculus under hard interventions.

- We provide a graphical characterization of $\mathcal{I}$-Markov equivalence for causal graphs with latent variables under the framework of hard interventions.

- We introduce a learning algorithm for inferring the graphical structure using a combination of different interventional data, while utilizing the corresponding new constraints. This procedure includes a new set of orientation rules, and we formally prove its soundness.

- We conducted experiments (Appendix F) to compare the size of $\mathcal{I}$-Markov equivalence class under hard and soft interventions to show that hard interventions on average provide more information about the causal graph.

## 2   Background and Related Works

In this section, we briefly describe related background knowledge and notations in this paper. Throughout this paper, we use upper case letters to denote variables, lower case letters to denote realizations, and bold letters for sets.

**Causal Bayesian Network (CBN):** Given a set of variables $\mathbf{V}$, $P(\mathbf{v})$ represents the joint distribution for $\mathbf{V} = \mathbf{v}$. A hard intervention $do(\mathbf{X} = \mathbf{x})$ refers to setting a subset $\mathbf{X} \subseteq \mathbf{V}$ to constants $\mathbf{x}$. It breaks the causal relationship between the intervened variables and their parents. The interventional distribution is $P_\mathbf{x}(\mathbf{v})$. Let $\mathcal{P}$ denote the tuple of all interventional distributions for all $\mathbf{X} \subseteq \mathbf{V}$. Then, a directed acyclic graph (DAG) $\mathcal{D} = (\mathbf{V}, \mathbf{E})$ is said to be a causal Bayesian network compatible with $\mathcal{P}$ if and only if, for all $\mathbf{X} \subseteq \mathbf{V}$, $P_\mathbf{x}(\mathbf{v}) = \prod_{\{i|V_i \notin \mathbf{X}\}} P(v_i|pa_i)$, for all $\mathbf{v}$ consistent with $\mathbf{x}$, and where $\mathbf{pa}_i$ is the set of parents of $V_i$ in $\mathcal{D}$. $\mathcal{D}$ is said to be causal if it satisfies this condition. $V[\mathcal{D}]$ and $E[\mathcal{D}]$ denote the set of all nodes and all edges of graph $\mathcal{D}$ respectively. Causal graphs entail specific conditional independence (CI) relationships among observable variables via d-separation statements. The d-separation serves as a criterion to determine whether a set of variables $\mathbf{X}$ is independent of another set $\mathbf{Y}$, given $\mathbf{Z}$.

If a causal graph has latent variables, it is denoted as $\mathcal{D} = (\mathbf{V} \cup \mathbf{L}, \mathbf{E})$ where $\mathbf{V}$ represents observable variables, $\mathbf{L}$ represents latent variables, and $\mathbf{E}$ denotes the edges. If a latent variable $L \in \mathbf{L}$ is a common cause of two observable variables, we use a curved bidirected edge between the two children variables. Such a causal graph is called Acyclic Directed Mixed Graph (ADMG). However, unlike causal graphs with sufficiency, the observed distribution is obtained by marginalizing $\mathbf{L}$ out as the Markovian condition does not hold in this case: $P(\mathbf{v}) = \sum_\mathbf{L} \prod_{\{i|T_i \in \mathbf{L} \cup \mathbf{V}\}} P(t_i|pa_i)$. Two causal graphs are called Markov equivalent if they encode the same set of CI statements over $\mathbf{V}$.

**Ancestral graphs:** Ancestral graphs are a graphical representation for a class of Markov equivalent causal graphs with latent variables. In an ADMG, $X$ is an ancestor of $Y$ if there is a directed path from $X$ to $Y$ [2]. $X$ is a spouse of $Y$ if $X \leftrightarrow Y$ is present. An inducing path relative to $\mathbf{Z}$ is a path on which every non-endpoint vertex $T \notin \mathbf{Z}$ is a collider on the path and every collider is an

---

[2]We follow the convention that a node is an ancestor of itself.

ancestor of one of the endpoints. A directed path is a cycle if it starts and ends at the same node. An almost directed cycle can be constructed by changing an arrowtail to an arrowhead in a cycle. An ADMG is ancestral if it does not contain any almost directed cycle. It is maximal if there is no inducing path (relative to the empty set) between any pair of non-adjacent vertices. It is called a Maximal Ancestral Graph (MAG) if it is both maximal and ancestral [Richardson and Spirtes, 2002]. In Zhang [2008a], the authors show how to uniquely construct a MAG for a causal graph with latents $\mathcal{D} = (\mathbf{V} \cup \mathbf{L}, \mathbf{E})$, such that all the (conditional) independence statements and ancestral relationships over $\mathbf{V}$ are preserved. Such CI statements are called m-separation statements in ADMGs.

In a graph $\mathcal{D}$, a triple $\langle X, Y, Z \rangle$ is unshielded if $X, Y$ are adjacent and $Y, Z$ are adjacent while $X, Z$ are not adjacent. If both edges are into $Y$, then it is an unshielded collider. A path between $X$ and $Y$, $p = \langle X, ..., W, Z, Y \rangle$, is a discriminating path for $Z$ if (1) $p$ includes at least three edges; (2) $Z$ is a non-endpoint node on $p$, and is adjacent to $Y$ on $p$; and (3) $X$ is not adjacent to $Y$, and every node between $X$ and $Z$ is a collider on $p$ and is a parent of $Y$. Two MAGs are Markov equivalent if and only if (1) they have the same skeleton; (2) they have the same unshielded colliders; and (3) if a path $p$ is a discriminating path for $Z$ in both MAGs, then Z is a collider on the path in one graph if and only if it is a collider on the path in the other. A partial ancestral graph (PAG) represents a Markov equivalence class of MAGs. It can be learned from CI statements over the observable variables under faithfulness. When observational data is provided, FCI algorithm is a commonly used algorithm to recover the PAG and is proved to be sound and complete in Zhang [2008b].

**Related works:** There are many works in the literature [Chickering, 2002, Hyttinen et al., 2013, Eberhardt, 2007, Shanmugam et al., 2015, Kocaoglu et al., 2017] related to learning the causal structure from a combination of observational and interventional data. Under the assumption of sufficiency, Hauser and Bühlmann [2012, 2014] introduced the Markov equivalence characterization. Yang et al. [2018] further showed that the same characterization can be used for both hard and soft interventions. More works aimed at the cases where latents are present in the graph. If only observational data is available, Zhang [2008b] showed the property and proposed the sound and complete FCI algorithm to learn a PAG. Spirtes et al. [2001], Colombo et al. [2012], Spirtes et al. [1991], Colombo et al. [2014], Ghassami et al. [2018], Kocaoglu et al. [2017] proposed FCI-variant algorithms under different settings. Kocaoglu et al. [2019] introduced a characterization for $\mathcal{I}$-Markov equivalence class for soft interventions using augmented graphs with $F$ node and proposed an FCI-variant algorithm to learn it. Following this, Jaber et al. [2020] characterized the $\psi$-MEC for unknown soft interventions and proposed a learning algorithm. Li et al. [2023] introduced the S-Markov property and learning algorithm when data from multiple domains are provided.

**Notations:** For disjoint sets of variables $\mathbf{X}, \mathbf{Y}, \mathbf{Z}$, a CI statement '$\mathbf{X}$ *is independent of* $\mathbf{Y}$ *conditioning on* $\mathbf{Z}$' is represented by $\mathbf{X} \perp\!\!\!\perp \mathbf{Y} | \mathbf{Z}$. Similarly, in a causal graph $\mathcal{D}$, the d-separation statement '$\mathbf{X}$ *is independent of* $\mathbf{Y}$ *conditioning on* $\mathbf{Z}$ *in graph* $\mathcal{D}$' is denoted as $(\mathbf{X} \perp\!\!\!\perp \mathbf{Y} | \mathbf{Z})_{\mathcal{D}}$. A set of interventions is $\mathcal{I} \subseteq 2^{\mathbf{V}}$, where $2^{\mathbf{V}}$ is the power set of $\mathbf{V}$. For two interventions $\mathbf{I}, \mathbf{J} \in \mathcal{I}$, the symmetric difference is $\mathbf{I} \Delta \mathbf{J} := (\mathbf{I} \setminus \mathbf{J}) \cup (\mathbf{J} \setminus \mathbf{I})$. $\mathcal{D}_{\overline{X}} / \mathcal{D}_{\underline{X}}$ is the graph obtained by removing all the edges into/out of $\mathbf{X}$ from $\mathcal{D}$. For $\mathcal{D}_{\overline{\mathbf{X}, \mathbf{Y}(\mathbf{Z})}}$, $\mathbf{Y}(\mathbf{Z})$ is the subset of $\mathbf{Y}$ that are not ancestors of $\mathbf{Z}$ in the graph $\mathcal{D}_{\overline{\mathbf{X}}}$. In a PAG, a circle mark in an edge $X o\rightarrow Y$ can be either an arrowtail or an arrowhead which is not determined. A star mark in an edge $X *\rightarrow Y$ is used as a wildcard which can be a circle, arrowhead, or arrowtail. We assume that there is no selection bias.

## 3   Combining Experimental Distributions under Do-Calculus

One of the most renowned contributions to causal inference is the development of do-calculus (also known as causal calculus) [Pearl, 1995, 2009]. Do-calculus is a set of three inference rules that enable the transformation of distributions associated with a causal graph. It leverages the graphical structure to determine when and how interventions can be adjusted or "removed" from expressions. In the context of hard interventions, the theorem is stated as follows[3]:

**Theorem 3.1.** *(Theorem 3 in Pearl [1995]). Let* $\mathcal{D} = (\mathbf{V} \cup \mathbf{L}, \mathbf{E})$ *be a causal graph. Then the following statements hold for any distribution that is consistent with* $\mathcal{D}$

*Rule 1 (see-see): For any* $\mathbf{X} \subseteq \mathbf{V}$ *and disjoint* $\mathbf{Y}, \mathbf{Z}, \mathbf{W} \subseteq \mathbf{V}$

---

[3]Here we put condition on $z$ for Rule 2. While this is redundant, we aim to show its clear connection with the corresponding $F$ node d-separations in the augmented graphs (see Section 4), which requires conditioning.

$P_{\mathbf{x}}(\mathbf{y}|\mathbf{w},\mathbf{z}) = P_{\mathbf{x}}(\mathbf{y}|\mathbf{w})$, *if* $\mathbf{Y} \perp\!\!\!\perp \mathbf{Z}|\mathbf{W},\mathbf{X}$ *in* $\mathcal{D}_{\overline{\mathbf{X}}}$

*Rule 2 (do-see): For any disjoint* $\mathbf{X},\mathbf{Y},\mathbf{Z} \subseteq \mathbf{V}$ *and* $\mathbf{W} \subseteq \mathbf{V} \setminus (\mathbf{Z} \cup \mathbf{Y})$

$P_{\mathbf{x},\mathbf{z}}(\mathbf{y}|\mathbf{w},\mathbf{z}) = P_{\mathbf{x}}(\mathbf{y}|\mathbf{w},\mathbf{z})$, *if* $\mathbf{Y} \perp\!\!\!\perp \mathbf{Z}|\mathbf{W},\mathbf{X}$ *in* $\mathcal{D}_{\overline{\mathbf{X}},\underline{\mathbf{Z}}}$

*Rule 3 (do-do): For any disjoint* $\mathbf{X},\mathbf{Y},\mathbf{Z} \subseteq \mathbf{V}$ *and* $\mathbf{W} \subseteq \mathbf{V} \setminus (\mathbf{Z} \cup \mathbf{Y})$

$P_{\mathbf{x},\mathbf{z}}(\mathbf{y}|\mathbf{w}) = P_{\mathbf{x}}(\mathbf{y}|\mathbf{w})$, *if* $\mathbf{Y} \perp\!\!\!\perp \mathbf{Z}|\mathbf{W},\mathbf{X}$ *in* $\mathcal{D}_{\overline{\mathbf{X}\mathbf{Z}(W)}}$

*where* $\mathbf{Z}(\mathbf{W}) \subseteq \mathbf{Z}$ *are non-ancestors of* $\mathbf{W}$ *in* $\mathcal{D}_{\overline{\mathbf{X}}}$.

Similar to the observations in Kocaoglu et al. [2019] that the converse of the rules can be utilized to derive insights of the graph structures, here we also need a set of statements for hard interventions. With soft intervention, the interventional graph remains the same as no causal relationship is broken by soft interventions. However, hard interventions induce changes to the causal graph, making it potentially more informative in learning the graph structure. The intuition is that with hard interventions, the causal graph becomes more sparse and thus more do-invariance statements can be found to constrain the graph. Accordingly, we show the following Proposition that characterizes the graph conditions from the invariance of two arbitrary intervention sets. Throughout the paper, for a pair of targets $\mathbf{I},\mathbf{J}$ and a conditioning set $\mathbf{W}$, we define the following useful sets: $\mathbf{K} = \mathbf{I}\Delta\mathbf{J}, \mathbf{K_I} = \mathbf{K} \setminus \mathbf{J}, \mathbf{K_J} = \mathbf{K} \setminus \mathbf{I}, \mathbf{W_I} = \mathbf{K_I} \cap \mathbf{W}, \mathbf{W_J} = \mathbf{K_J} \cap \mathbf{W}, \mathbf{R} = \mathbf{K} \setminus \mathbf{W}, \mathbf{R_I} = \mathbf{R} \cap \mathbf{K_I}, \mathbf{R_J} = \mathbf{R} \cap \mathbf{K_J}$.

**Proposition 3.2.** *(Generalized do-calculus for hard interventions). Let* $\mathcal{D} = (\mathbf{V} \cup \mathbf{L}, \mathbf{E})$ *be a causal graph with latents. Then, the following holds for any tuple of hard-interventional distributions* $(P_{\mathbf{I}})_{\mathbf{I} \in \mathcal{I}}$ *consistent with* $\mathcal{D}$*, where* $\mathcal{I} \subseteq 2^{\mathbf{V}}$.

*Rule 1 (conditional independence): For any* $\mathbf{I} \subseteq \mathbf{V}$ *and disjoint* $\mathbf{Y},\mathbf{Z},\mathbf{W} \subseteq (\mathbf{V} \setminus \mathbf{I})$

$\quad P_{\mathbf{I}}(\mathbf{y}|\mathbf{w},\mathbf{z}) = P_{\mathbf{I}}(\mathbf{y}|\mathbf{w})$, *if* $\mathbf{Y} \perp\!\!\!\perp \mathbf{Z}|\mathbf{W},\mathbf{I}$ *in* $\mathcal{D}_{\overline{\mathbf{I}}}$

*Rule 2 (do-see): For any* $\mathbf{I},\mathbf{J} \subseteq \mathbf{V}$ *and disjoint* $\mathbf{Y},\mathbf{W} \subseteq \mathbf{V} \setminus \mathbf{K}$*, where* $\mathbf{K} := \mathbf{I}\Delta\mathbf{J}$

$\quad P_{\mathbf{I}}(\mathbf{y}|\mathbf{w},\mathbf{k}) = P_{\mathbf{I},\mathbf{J}}(\mathbf{y}|\mathbf{w},\mathbf{k}) = P_{\mathbf{J}}(\mathbf{y}|\mathbf{w},\mathbf{k})$, *if* $(\mathbf{Y} \perp\!\!\!\perp \mathbf{K_J}|\mathbf{W},\mathbf{I})_{\mathcal{D}_{\overline{\mathbf{I}},\underline{\mathbf{K_J}}}} \wedge (\mathbf{Y} \perp\!\!\!\perp \mathbf{K_I}|\mathbf{W},\mathbf{J})_{\mathcal{D}_{\overline{\mathbf{J}},\underline{\mathbf{K_I}}}}$

*Rule 3 (do-do): For any* $\mathbf{I},\mathbf{J} \subseteq \mathbf{V}$ *and disjoint* $\mathbf{Y},\mathbf{W} \subseteq \mathbf{V} \setminus \mathbf{K}$*, where* $\mathbf{K} := \mathbf{I}\Delta\mathbf{J}$

$\quad P_{\mathbf{I}}(\mathbf{y}|\mathbf{w}) = P_{\mathbf{I},\mathbf{J}}(\mathbf{y}|\mathbf{w}) = P_{\mathbf{J}}(\mathbf{y}|\mathbf{w})$, *if* $(\mathbf{Y} \perp\!\!\!\perp \mathbf{K_J}|\mathbf{W},\mathbf{I})_{\mathcal{D}_{\overline{\mathbf{I},\mathbf{K_J}(W)}}} \wedge (\mathbf{Y} \perp\!\!\!\perp \mathbf{K_I}|\mathbf{W},\mathbf{J})_{\mathcal{D}_{\overline{\mathbf{J},\mathbf{K_I}(W)}}}$

*Rule 4 (mixed do-see/do-do): For any* $\mathbf{I},\mathbf{J} \subseteq \mathbf{V}$ *and disjoint* $\mathbf{Y},\mathbf{W} \subseteq \mathbf{V}$*, where* $\mathbf{K} := \mathbf{I}\Delta\mathbf{J}$

$\quad P_{\mathbf{I}}(\mathbf{y}|\mathbf{w}) = P_{\mathbf{I},\mathbf{J}}(\mathbf{y}|\mathbf{w},\mathbf{k}) = P_{\mathbf{J}}(\mathbf{y}|\mathbf{w})$, *if* $(\mathbf{Y} \perp\!\!\!\perp \mathbf{R_J}|\mathbf{W},\mathbf{I})_{\mathcal{D}_{\overline{\mathbf{I},\mathbf{R_J}(W)}}} \wedge (\mathbf{Y} \perp\!\!\!\perp \mathbf{W_J}|\mathbf{W},\mathbf{I})_{\mathcal{D}_{\overline{\mathbf{I}},\underline{\mathbf{W_J}}}} \wedge (\mathbf{Y} \perp\!\!\!\perp \mathbf{R_I}|\mathbf{W},\mathbf{J})_{\mathcal{D}_{\overline{\mathbf{J},\mathbf{R_I}(W)}}} \wedge (\mathbf{Y} \perp\!\!\!\perp \mathbf{W_I}|\mathbf{W},\mathbf{J})_{\mathcal{D}_{\overline{\mathbf{J}},\underline{\mathbf{W_I}}}}$

Note that Rule 2 and 3 are special cases of Rule 4. We present them to make the connection to standard causal calculus rules more explicit. In the following sections, we will show how the generalized rules can be crucial in characterizing and learning the $\mathcal{I}$-Markov Equivalence Class ($\mathcal{I}$-MEC).

# 4 $\mathcal{I}$-Markov Equivalence Class

In this section, we will characterize the graphical conditions for interventional Markov equivalence class. First of all, we start by introducing the definition of interventional Markov equivalence based on the new do-constraint rules.

**Definition 4.1.** Consider the tuples of absolutely continuous probability distributions $(P_{\mathbf{I}})_{\mathbf{I} \in \mathcal{I}}$ over a set of variables $\mathbf{V}$. A tuple $(P_{\mathbf{I}})_{\mathbf{I} \in \mathcal{I}}$ satisfies the $\mathcal{I}$-Markov property with respect to a causal graph $\mathcal{D} = (\mathbf{V} \cup \mathbf{L}, \mathbf{E})$ if the following holds for disjoint $\mathbf{Y},\mathbf{Z},\mathbf{W} \subseteq \mathbf{V}$:

1. For $\mathbf{I} \in \mathcal{I}$: $P_{\mathbf{I}}(\mathbf{y}|\mathbf{w},\mathbf{z}) = P_{\mathbf{I}}(\mathbf{y}|\mathbf{w})$ if $\mathbf{Y} \perp\!\!\!\perp \mathbf{Z}|\mathbf{W},\mathbf{I}$ in $\mathcal{D}_{\overline{\mathbf{I}}}$

2. For $\mathbf{I},\mathbf{J} \in \mathcal{I}$: $P_{\mathbf{I}}(\mathbf{y}|\mathbf{w}) = P_{\mathbf{J}}(\mathbf{y}|\mathbf{w})$ if $(\mathbf{Y} \perp\!\!\!\perp \mathbf{R_J}|\mathbf{W},\mathbf{I})_{\mathcal{D}_{\overline{\mathbf{I},\mathbf{R_J}(W)}}} \wedge (\mathbf{Y} \perp\!\!\!\perp \mathbf{W_J}|\mathbf{W},\mathbf{I})_{\mathcal{D}_{\overline{\mathbf{I}},\underline{\mathbf{W_J}}}} \wedge (\mathbf{Y} \perp\!\!\!\perp \mathbf{R_I}|\mathbf{W},\mathbf{J})_{\mathcal{D}_{\overline{\mathbf{J},\mathbf{R_I}(W)}}} \wedge (\mathbf{Y} \perp\!\!\!\perp \mathbf{W_I}|\mathbf{W},\mathbf{J})_{\mathcal{D}_{\overline{\mathbf{J}},\underline{\mathbf{W_I}}}}$

The set of all tuples that satisfy the $\mathcal{I}$-Markov property with respect to $\mathcal{D}$ are denoted by $\mathcal{P}_{\mathcal{I}}(\mathcal{D},\mathbf{V})$.

The two conditions of $\mathcal{I}$-Markov property correspond to the first Rule in Theorem 3.1 and Rule 4 in Proposition 3.2 respectively. When $\mathcal{I} = \emptyset$, i.e. we only have access to observational distribution, this definition aligns with the well-known definition of Markov equivalence. It only implies the first condition on the observational distribution $P(\mathbf{V})$. Accordingly, two causal graphs are said to be $\mathcal{I}$-Markov equivalent if they induce the same constraints to the interventional distribution tuple which we formalize as follows:

**Definition 4.2.** Given two causal graphs $\mathcal{D}_1 = (\mathbf{V} \cup \mathbf{L}_1, \mathbf{E}_1)$ and $\mathcal{D}_2 = (\mathbf{V} \cup \mathbf{L}_2, \mathbf{E}_2)$, and a set of intervention targets $\mathcal{I} \subseteq 2^V$, $\mathcal{D}_1$ and $\mathcal{D}_2$ are $\mathcal{I}$-Markov equivalent if $\mathcal{P}_{\mathcal{I}}(\mathcal{D}_1, \mathbf{V}) = \mathcal{P}_{\mathcal{I}}(\mathcal{D}_2, \mathbf{V})$.

The challenge of checking the $\mathcal{I}$-Markov property in Definition 4.1 involves checking multiple graph conditions in different graph mutilations of $\mathcal{D}$. In order to construct a more compact representation of $\mathcal{D}$ that captures all the graph conditions, we construct the augmented pair graph defined as follows[4].

**Definition 4.3** (Augmented Pair Graph). Given a causal graph $\mathcal{D} = (\mathbf{V} \cup \mathbf{L}, \mathbf{E})$ and a set of intervention targets $\mathcal{I} \subseteq 2^V$, for a pair of interventions $\mathbf{I}, \mathbf{J} \in \mathcal{I}$, the augmented pair graph of $\mathcal{D}$, denoted as $\mathrm{Aug}_{(\mathbf{I},\mathbf{J})}(\mathcal{D})$, is constructed as: $\mathrm{Aug}_{(\mathbf{I},\mathbf{J})}(\mathcal{D}) = (\mathbf{V}^{(\mathbf{I})} \cup \mathbf{V}^{(\mathbf{J})} \cup \{F^{(\mathbf{I},\mathbf{J})}\}, \mathbf{E}^{(\mathbf{I})} \cup \mathbf{E}^{(\mathbf{J})} \cup \mathcal{E})$, where $F^{(\mathbf{I},\mathbf{J})}$ is an auxiliary node with the superscript representing the pair of intervention targets it refers to, $\mathbf{E}^{(\mathbf{I})} = E[\mathcal{D}_{\overline{\mathbf{I}}}], \mathbf{E}^{(\mathbf{J})} = E[\mathcal{D}_{\overline{\mathbf{J}}}], \mathcal{E} = \{(F^{(\mathbf{I},\mathbf{J})}, S)\}_{S \in \mathbf{K}^{(\mathbf{I})} \cup \mathbf{K}^{(\mathbf{J})}}$, with $S$ as a singleton.

In words, for each pair of interventions $\mathbf{I}, \mathbf{J}$, we create the augmented pair graph by creating two copies of vertices $\mathbf{V}^{(\mathbf{I})}, \mathbf{V}^{(\mathbf{J})}$ and adding the edges between the vertices with those in the corresponding interventional graphs $\mathcal{D}_{\overline{\mathbf{I}}}, \mathcal{D}_{\overline{\mathbf{J}}}$, and then connecting the auxiliary node $F$ to all the nodes in the symmetric difference of $\mathbf{I}, \mathbf{J}$. We will omit the subscript for the graph and superscript for $F$ node when the pair of interventions is clear from the context. This kind of construction has been proposed and used in the causality literature before [Eberhardt and Scheines, 2007, Hauser and Bühlmann, 2012, Pearl, 2009, Dawid, 2002]. The constructed augmented pairs allow us to test the m-separation statements as listed in Definition 4.1 without looking into mutilations of the original graph $\mathcal{D}$. This is illustrated by the following Proposition.

**Proposition 4.4.** *Given a causal graph $\mathcal{D} = (\mathbf{V} \cup \mathbf{L}, \mathbf{E})$ and a set of intervention targets $\mathcal{I} \subseteq 2^V$, for each pair of interventions $\mathbf{I}, \mathbf{J} \in \mathcal{I}$, $\mathbf{K} = \mathbf{I} \Delta \mathbf{J}$, and the corresponding augmented pair graph $\mathrm{Aug}_{(\mathbf{I},\mathbf{J})}(\mathcal{D}) = (\mathbf{V}^{(\mathbf{I})} \cup \mathbf{V}^{(\mathbf{J})} \cup \{F^{(\mathbf{I},\mathbf{J})}\}, \mathbf{E}^{(\mathbf{I})} \cup \mathbf{E}^{(\mathbf{J})} \cup \mathcal{E}), \mathcal{E} = \{(F^{(\mathbf{I},\mathbf{J})}, S)\}_{S \in K^{(\mathbf{I})} \cup K^{(\mathbf{J})}}$, we have the following equivalence statements:*

*For disjoint $\mathbf{Y}, \mathbf{Z}, \mathbf{W} \subseteq \mathbf{V}$:*

$$(\mathbf{Y} \perp\!\!\!\perp \mathbf{Z} | \mathbf{W}, \mathbf{I})_{\mathcal{D}_{\overline{\mathbf{I}}}} \iff (\mathbf{Y} \perp\!\!\!\perp \mathbf{Z} | \mathbf{W}, \mathbf{I}, F^{(\mathbf{I},\mathbf{J})})_{\mathrm{Aug}_{(\mathbf{I},\mathbf{J})}(\mathcal{D})} \tag{1}$$

*For disjoint $\mathbf{Y}, \mathbf{Z}, \mathbf{W} \subseteq \mathbf{V}$ :*

$$\begin{cases} (\mathbf{Y} \perp\!\!\!\perp \mathbf{R_J} | \mathbf{W}, \mathbf{I})_{\mathcal{D}_{\overline{\mathbf{I}, \mathbf{R_J(W)}}}} \\ (\mathbf{Y} \perp\!\!\!\perp \mathbf{W_J} | \mathbf{W} \setminus \mathbf{W_J}, \mathbf{I})_{\mathcal{D}_{\overline{\mathbf{I}}, \underline{\mathbf{W_J}}}} \\ (\mathbf{Y} \perp\!\!\!\perp \mathbf{R_I} | \mathbf{W}, \mathbf{J})_{\mathcal{D}_{\overline{\mathbf{J}, \mathbf{R_I(W)}}}} \\ (\mathbf{Y} \perp\!\!\!\perp \mathbf{W_I} | \mathbf{W} \setminus \mathbf{W_I}, \mathbf{J})_{\mathcal{D}_{\overline{\mathbf{J}}, \underline{\mathbf{W_I}}}} \end{cases} \iff \begin{cases} (F^{(\mathbf{I},\mathbf{J})} \perp\!\!\!\perp \mathbf{Y}^{(\mathbf{I})} | \mathbf{I}^{(\mathbf{I})}, \mathbf{W}^{(\mathbf{I})})_{\mathrm{Aug}_{(\mathbf{I},\mathbf{J})}(\mathcal{D})} \\ (F^{(\mathbf{I},\mathbf{J})} \perp\!\!\!\perp \mathbf{Y}^{(\mathbf{J})} | \mathbf{J}^{(\mathbf{J})}, \mathbf{W}^{(\mathbf{J})})_{\mathrm{Aug}_{(\mathbf{I},\mathbf{J})}(\mathcal{D})} \end{cases} \tag{2}$$

While the augmented pair graphs encode the same CI statements as the original graph, we know that different graphs may entail the same CI statements. To characterize the $\mathcal{I}$-Markov equivalence, we utilize the structure of Maximal Ancestral Graphs (MAGs). MAGs represent the Markov equivalence class of the original graph, making it possible to compare and analyze equivalence classes without needing the full graph with latent variables. We introduce the following definition to construct a graph structure that captures the $\mathcal{I}$-Markov equivalence.

**Definition 4.5.** (Twin Augmented MAG). Given a causal graph $\mathcal{D} = (\mathbf{V} \cup \mathbf{L}, \mathbf{E})$ and a set of interventions $\mathcal{I} \subseteq 2^V$, for each pair of intervention targets $\mathbf{I}, \mathbf{J} \in \mathcal{I}$, $\mathbf{K} = \mathbf{I} \Delta \mathbf{J}$, and the corresponding augmented pair graph $Aug_{(\mathbf{I},\mathbf{J})}(\mathcal{D}) = (\mathbf{V}^{(\mathbf{I})} \cup \mathbf{V}^{(\mathbf{J})} \cup \{F^{(\mathbf{I},\mathbf{J})}\}, \mathbf{E}^{(\mathbf{I})} \cup \mathbf{E}^{(\mathbf{J})} \cup \mathcal{E}), \mathcal{E} = \{(F^{(\mathbf{I},\mathbf{J})}, S)\}_{S \in \mathbf{K}^{(\mathbf{I})} \cup \mathbf{K}^{(\mathbf{J})}}$, construct the MAG[5] of the augmented pair graph and denote it as

---

[4]Throughout this paper, we use the superscript for (a set of) nodes to denote the interventional domain.

[5]We use the conventional steps to construct the MAG from an ADMG: For each pair of nodes $X, Y$, if $X$ is $Y$'s ancestor/descendant/spouse and there is an inducing path between them, we orient $X \to Y / X \leftarrow Y / X \leftrightarrow Y$ between them, otherwise they are not adjacent.

$\mathrm{MAG}(\mathrm{Aug}_{(\mathbf{I},\mathbf{J})}(\mathcal{D}))$. The twin augmented MAG, denoted as $\mathrm{Twin}_{(\mathbf{I},\mathbf{J})}(\mathcal{D})$, is constructed by adding edges $(F, S^{(\mathbf{I})})$ and $(F, S^{(\mathbf{J})})$ to $\mathrm{MAG}(\mathrm{Aug}_{(\mathbf{I},\mathbf{J})}(\mathcal{D}))$ if for the singleton $S \in \mathbf{V}$, $S^{(\mathbf{I})}$ or $S^{(\mathbf{J})}$ is adjacent to $F$ in $\mathrm{MAG}(\mathrm{Aug}_{(\mathbf{I},\mathbf{J})}(\mathcal{D}))$.

**Lemma 4.6.** *Twin augmented MAGs are valid MAGs.*

The motivation of adding extra edges to $F$ nodes comes from the fact that a query $F \perp\!\!\!\perp Y^{(\mathbf{I})}, Y \in \mathbf{V}$ itself is not testable by comparing the invariances using $P_{\mathbf{I}}, P_{\mathbf{J}}$. It requires access to $P_{\mathbf{I},\mathbf{J}}$ which is not necessarily given. Therefore, when the invariance does not hold, we cannot distinguish if $F$ is non-separable to $Y$ in only one domain or in both domains. Next, we give a graphical characterization of $\mathcal{I}$-Markov equivalence between causal graphs using Definition 4.5.

**Theorem 4.7.** *Given two causal graphs $\mathcal{D}_1 = (\mathbf{V} \cup \mathbf{L}_1, \mathbf{E}_1)$ and $\mathcal{D}_2 = (\mathbf{V} \cup \mathbf{L}_2, \mathbf{E}_2)$, and a set of intervention targets $\mathcal{I} \subseteq 2^V$, $\mathcal{D}_1$ and $\mathcal{D}_2$ are $\mathcal{I}$-Markov equivalent with respect to $\mathcal{I}$ if and only if for each pair of interventions $\mathbf{I}, \mathbf{J} \in \mathcal{I}$, $\mathcal{M}_1 = \mathrm{Twin}_{(\mathbf{I},\mathbf{J})}(\mathcal{D}_1), \mathcal{M}_2 = \mathrm{Twin}_{(\mathbf{I},\mathbf{J})}(\mathcal{D}_2)$:*

1. *$\mathcal{M}_1$ and $\mathcal{M}_2$ have the same skeleton;*

2. *$\mathcal{M}_1$ and $\mathcal{M}_2$ have the same unshielded colliders;*

3. *If a path $p$ is a discriminating path for a node $Y$ in both $\mathcal{M}_1$ and $\mathcal{M}_2$, then $Y$ is a collider on the path if and only if it is a collider on the path in the other.*

To illustrate how we construct twin augmented MAGs step by step and compare $\mathcal{I}$-Markov equivalence for two causal graphs, we construct the examples in Figure 2 in Appendix D.

# 5 $\mathcal{I}$-Augmented MAG

We have demonstrated that the characterization of $\mathcal{I}$-Markov equivalence between two causal graphs can be effectively captured using the proposed twin augmented MAGs. However, for an intervention target set of size $k$, we will have to inspect all $\binom{k}{2}$ such structures. This is undesirable, as there exists only one underlying causal graph, and a more compact graph representation is preferred. Additionally, each twin augmented MAG encodes information for only a single pair of distributions, whereas we aim for an objective that encapsulates as much information as possible. To address these challenges, we propose a new graphical structure defined as follows:

**Definition 5.1** ($\mathcal{I}$-augmented MAG). Given a causal graph $\mathcal{D} = (\mathbf{V}, \mathbf{E})$ and a set of intervention targets $\mathcal{I}$, construct all twin augmented MAGs $\mathrm{Twin}_{(\mathbf{I},\mathbf{J})}(\mathcal{D})$ for all $\mathbf{J} \in \mathcal{I} \setminus \{\mathbf{I}\}$. For each $\mathbf{I} \in \mathcal{I}$, the $\mathcal{I}$-augmented MAG related to $\mathcal{I}$ is defined as $\mathrm{Aug}_{\mathbf{I}}(\mathcal{D}, \mathcal{I}) = (\mathbf{V} \cup \mathcal{F}, E(\mathcal{D}_{\bar{\mathbf{I}}}) \cup \mathcal{E})$ where $\mathcal{F} = \{F^{(\mathbf{I},\mathbf{J})}\}_{\mathbf{J} \in \mathcal{I} \setminus \{\mathbf{I}\}}, \mathcal{E} = \{(F^{(\mathbf{I},\mathbf{J})}, X^{(\mathbf{I})})\}_{(F^{(\mathbf{I},\mathbf{J})}, X^{(\mathbf{I})}) \in E(Twin_{(\mathbf{I},\mathbf{J})}(\mathcal{D})), \mathbf{J} \in \mathcal{I} \setminus \{\mathbf{I}\}}$. In other words, it is the graph union of each twin augmented MAG's induced subgraph on $\mathbf{V}^{(\mathbf{I})} \cup \{F^{(\mathbf{I},\mathbf{J})}\}$.
The $\mathcal{I}$-augmented MAG tuple is a tuple of all $\mathcal{I}$-augmented MAGs $\mathcal{N}_{\mathcal{I}}(\mathcal{D}) = (\mathrm{Aug}_{\mathbf{I}}(\mathcal{D}, \mathcal{I}))_{\mathbf{I} \in \mathcal{I}}$.

**Remark:** The $\mathcal{I}$-augmented MAG $\mathrm{Aug}_{\mathbf{I}}(\mathcal{D}, \mathcal{I})$ preserves all the m-separation statements in the domain of $do(\mathbf{I})$ from the twin augmented MAGs with $\mathbf{I}$ in the intervention pair, $\mathrm{Twin}_{(\mathbf{I},\mathbf{J})}(\mathcal{D}), \mathbf{J} \in \mathcal{I} \setminus \{\mathbf{I}\}$. The structure of $\mathbf{V}^{(\mathbf{I})} \cup F$ within each twin augmented MAG is preserved in the $\mathcal{I}$-augmented MAG and is not affected by the other domains.
The constructed $\mathcal{I}$-augmented MAG tuple consists of only $k$ graphs, each of which encapsulates more information on the domain than a twin augmented MAG. Specifically, the set of ADMGs consistent with a twin augmented MAG in one domain is a superset of those consistent with an $\mathcal{I}$-augmented MAG, as the $\mathcal{I}$-augmented MAG imposes stricter constraints of separations across other domains on the causal graph. Furthermore, the graphical conditions for two causal graphs to be $\mathcal{I}$-Markov equivalent, as stated in Theorem 4.7, remain valid when using the $\mathcal{I}$-augmented MAG. Hence, the $\mathcal{I}$-augmented MAG serves as the unified and compact graphical representation. Below, we illustrate the steps of how to construct an $\mathcal{I}$-augmented MAG.
In Figure 1, we demonstrate the construction of $\mathcal{I}$-augmented MAGs with respect to three datasets: observational data and interventions on $X$ and $Z$, i.e., $\mathcal{I} = \{\mathbf{I}_1 = \emptyset, \mathbf{I}_2 = \{X\}, \mathbf{I}_3 = \{Z\}\}$, for the graph $\mathcal{D}_1 = [X \to Y \to Z, Y \leftrightarrow Z]$. For simplicity, we relabel the observational domain and the interventional domains for $X$ and $Z$ as 1, 2, and 3, respectively, in the $\mathcal{I}$-augmented MAGs shown in Figure 1. Figure 1a, Figure 1b, and Figure 1c are the twin augmented MAGs given

$(\emptyset, \{X\}), (\emptyset, \{Z\})$, and $(\{X\}, \{Z\})$ respectively. Based on the twin augmented MAGs, we construct the $\mathcal{I}$-augmented MAGs as shown in Figure 1d, Figure 1e, and Figure 1f for the domains $\emptyset, \{X\}$ and $\{Z\}$ respectively. Each $\mathcal{I}$-augmented MAG has the domain-specific skeleton in the center with the $F$ nodes around it indicating the invariances from other domains. The $\mathcal{I}$-augmented MAGs entail the same information about the causal graph as the twin augmented MAGs, but they have a much more compact representation. Proposition 5.2 shows the equivalence between the two representations.

Figure 1: Illustration of the construction of $\mathcal{I}$−augmented MAGs from twin augmented MAGs. Figure 2a is the ground truth graph. The intervention targets are $\mathcal{I} = \{\mathbf{I}_1 = \emptyset, \mathbf{I}_2 = \{X\}, \mathbf{I}_3 = \{Z\}\}$. (a), (b), and (c) are the twin augmented MAGs. (d), (e), and (f) are the $\mathcal{I}$-augmented MAGs.

**Proposition 5.2.** *Given two causal graphs $\mathcal{D}_1 = (\mathbf{V} \cup \mathbf{L}_1, \mathbf{E}_1), \mathcal{D}_2 = (\mathbf{V} \cup \mathbf{L}_2, \mathbf{E}_2)$ and a set of intervention targets $\mathcal{I} \subseteq 2^V$, construct the $\mathcal{I}$-augmented MAGs following the steps in Definition 5.1 of $\mathcal{D}_1, \mathcal{D}_2$. $\mathcal{D}_1$ and $\mathcal{D}_2$ are $\mathcal{I}$-Markov equivalent with respect to $\mathcal{I}$ if and only if for each $\mathbf{I} \in \mathcal{I}$, $\mathrm{Aug}_{\mathbf{I}}(\mathcal{D}_1, \mathcal{I})$ and $\mathrm{Aug}_{\mathbf{I}}(\mathcal{D}_2, \mathcal{I})$ satisfy the 3 conditions in Theorem 4.7.*

## 6 Learning by Combining Experiments

In this section, we develop an algorithm to learn the causal structure from given datasets. We do not assume that observational data is given. Like any learning algorithm, a faithfulness assumption is necessary to infer graphical properties from the distributional constraints. Accordingly, we assume that the provided interventional distributions are h-faithful to the causal graph $\mathcal{D}$, defined below.

**Definition 6.1** (h-faithful). For a causal graph $\mathcal{D} = (\mathbf{V} \cup \mathbf{L}, \mathbf{E})$, a tuple of distributions $(P_{\mathbf{I}})_{\mathbf{I} \in \mathcal{I}} \in \mathcal{P}(\mathcal{D}, \mathbf{V})$ is called h-faithful to $\mathcal{D}$ if the converse for each of the conditions in Definition 4.1 holds.

### 6.1 Learning Objective

Similar to the case when only observational data is available, it is hard to recover the whole graph in general. Therefore, the objective of the algorithm is to learn a graphical representation that demonstrates a set of $\mathcal{I}$-Markov equivalent graphs. However, although the MAG of the augmented pair graphs proposed in Definition 4.5 denotes the ground truth, it is not always a fundamentally learnable structure from the distributions. To witness, let us consider the example in Figure 2c. The edge $(F, Y^{(2)})$ is not in $\mathrm{MAG}(\mathrm{Aug}_{(\emptyset, \{Z\})}(\mathcal{D}_1))$. While we can learn only that from the distributions, $P_{obs}(y|\mathbf{w}) \neq P_Z(y|\mathbf{w})$ for some $\mathbf{W} \subseteq \{X, Z\}$. The inequality tells us that there is an inducing path from $F$ to $Y^{(1)}$ or $Y^{(2)}$ which we cannot distinguish. Therefore, we proposed the twin augmented MAG to be able to capture the characterization for $\mathcal{I}$-Markov equivalent graphs. Based on that, we construct the $\mathcal{I}$-augmented MAG which is a more compact graphical structure and we use it as the learning objective. Accordingly, we define the $\mathcal{I}$-augmented graph as follows.

**Definition 6.2** ($\mathcal{I}$-augmented Graph). Given a causal graph $\mathcal{D}$ and a set of intervention targets $\mathcal{I}$, for each $\mathbf{I} \in \mathcal{I}$, let $\mathcal{M} = \mathrm{Aug}_{\mathbf{I}}(\mathcal{D}, \mathcal{I})$ and let $[\mathcal{M}]$ be the set of $\mathcal{I}$-augmented MAGs corresponding to all the causal graphs that are $\mathcal{I}$-Markov equivalent to $\mathcal{D}$ given $\mathcal{I}$. For any $\mathbf{I} \in \mathcal{I}$, the $\mathcal{I}$-augmented graph, denoted as $\mathcal{G}_{\mathbf{I}}(\mathcal{D}, \mathcal{I})$, is a graph such that:

1. $\mathcal{G}_{\mathbf{I}}(\mathcal{D}, \mathcal{I})$ has the same adjacencies as $\mathcal{M}$, and any member of $[\mathcal{M}]$ does; and

2. every non-circle mark in $\mathcal{G}_{\mathbf{I}}(\mathcal{D}, \mathcal{I})$ is an invariant mark in $[\mathcal{M}]$.

The $\mathcal{I}$-augmented graph tuple $\mathcal{L}_{\mathcal{I}}(\mathcal{D})$ is a tuple of all $\mathcal{I}$-augmented graphs $\mathcal{L}_{\mathcal{I}}(\mathcal{D}) = (\mathcal{G}_{\mathbf{I}}(\mathcal{D}, \mathcal{I}))_{\mathbf{I} \in \mathcal{I}}$. We will omit the graph $\mathcal{D}$ or the intervention targets $\mathcal{I}$ when it is clear from the context for simplicity.

## 6.2 The Learning Algorithm

---

**Algorithm 1** Main Causal Discovery Algorithm

---

**Input:** Intervention targets $\mathcal{I}$, interventional distributions $(P_{\mathbf{I}})_{\mathbf{I} \in \mathcal{I}}$, observable variables $\mathbf{V}$
Initialize $\mathcal{L}_{\mathcal{I}}$ as an empty tuple, $SepSet$ as an empty set;
**Phase I:** Initialize with Complete Graphs;
**for I in $\mathcal{I}$ do**
    Duplicate $\mathbf{V}$ to create $\mathbf{V}^{(\mathbf{I})}$, run Algorithm 2 on $\mathcal{I}, \mathbf{I}$ to get $\mathcal{F}_{\mathbf{I}}$ as $F$ nodes;
    Put a circle edge ($o$—$o$) between every pair of $X \in \mathbf{V}^{(\mathbf{I})}$ and $Y \in \mathbf{V}^{(\mathbf{I})} \cup \mathcal{F}_{\mathbf{I}}$;
**Phase II:** Learning the Skeleton and Separating Sets;
**for I in $\mathcal{I}$ do**
    $E(\mathcal{G}_{\mathbf{I}}) \leftarrow \emptyset, V(\mathcal{G}_{\mathbf{I}}) \leftarrow \mathbf{V}^{(\mathbf{I})} \cup \mathcal{F}_{\mathbf{I}}$;
    **for J in $\mathcal{I}$ do**
        Run Algorithm 3 on $\mathbf{I}, \mathbf{J}, (P_{\mathbf{I}})_{\mathbf{I} \in \mathcal{I}}, \mathbf{V}, \mathcal{F}_{\mathbf{I}}$ to get $\mathcal{E}_{\mathbf{J}}, SepSet_{\mathbf{J}}$;
        $E(\mathcal{G}_{\mathbf{I}}) \leftarrow E(\mathcal{G}_{\mathbf{I}}) \cup \mathcal{E}_{\mathbf{J}}, SepSet \leftarrow SepSet \cup SepSet_{\mathbf{J}}$;
    $\mathcal{L}_{\mathcal{I}} \leftarrow \mathcal{L}_{\mathcal{I}} \cup \mathcal{G}_{\mathbf{I}}$;
**Phase III:** Apply Orientation Rules to each $\mathcal{G}_{\mathbf{I}}, \mathbf{I} \in \mathcal{I}$;
**Rule 0:** For every unshielded triple $\langle X^{(\mathbf{I})}, Y^{(\mathbf{I})}, Z^{(\mathbf{I})} \rangle$ in $\mathcal{G}_{\mathbf{I}}, \mathbf{I} \in \mathcal{I}$, orient it as $X^{(\mathbf{I})} * \!\!\rightarrow Y^{(\mathbf{I})} \leftarrow \!\! * Z^{(\mathbf{I})}$ if $Y \notin SepSet_{\mathbf{I}}(X, Z)$.
First apply Rule 0, then apply 7 FCI rules in Zhang [2008b] together with the following 4 additional rules to each $\mathcal{G}_{\mathcal{I}}$ until none applies.
**Rule 8:** For any edge adjacent to an $F$ node, orient the edge out of the $F$ node.
**Rule 9:** For any $\mathbf{I} \in \mathcal{I}$, if $X \in \mathbf{I}, X^{(\mathbf{I})}, Y^{(\mathbf{I})}$ are adjacent in $\mathcal{G}_{\mathbf{I}}$, then orient $X^{(\mathbf{I})} o$—$* Y^{(\mathbf{I})}$ as $X^{(\mathbf{I})} \rightarrow Y^{(\mathbf{I})}$.
**Rule 10:** If $X^{(\mathbf{I})} \rightarrow Y^{(\mathbf{I})}$ in $\mathcal{G}_{\mathbf{I}}$ for some $\mathbf{I} \in \mathcal{I}$, replace the circle mark at $Y^{(\mathbf{J})}$ between $X^{(\mathbf{J})}$ and $Y^{(\mathbf{J})}$ in $\mathcal{G}_{\mathbf{J}}$ with an arrowhead for any $\mathbf{J} \in \mathcal{I} \setminus \{\mathbf{I}\}$.
**Rule 11:** In $\mathcal{G}_{\mathbf{I}}, \mathbf{I}, \mathbf{J} \in \mathcal{I}$, if $\mathbf{J} = \mathbf{I} \cup \{X\}$, $F^{(\mathbf{I}, \mathbf{J})}$ is adjacent to $Y^{(\mathbf{I})}, Y \notin \mathbf{J}$, then orient $X^{(\mathbf{I})} * \!\!-\!\! * Y^{(\mathbf{I})}$ as $X^{(\mathbf{I})} \rightarrow Y^{(\mathbf{I})}$.
**Output:** $\mathcal{I}$-augmented graph tuple $\mathcal{L}_{\mathcal{I}}$

---

We propose Algorithm 1 to learn the $\mathcal{I}$-augmented graph tuple from the given experiments. The algorithm is inspired by the FCI algorithm. It learns the $\mathcal{I}$-augmented graph tuple $\mathcal{L}_{\mathcal{I}}$ by iteratively recovering $\mathcal{G}_{\mathbf{I}}$ for each $\mathbf{I} \in \mathcal{I}$. In Phase I, it initializes the $\mathcal{I}$-augmented graph $\mathcal{G}_{\mathbf{I}}$ for each $\mathbf{I} \in \mathcal{I}$. It puts a circle edge between each pair of nodes $X, Y \in \mathbf{V}$. This constructs the domain-specific skeleton $G^{(\mathbf{I})}$ under target $\mathbf{I}$. After that, we attach the $F$ nodes to the $\mathcal{I}$-augmented graph using Algorithm 2 and then put a circle edge between any $F$ node and any $X \in \mathbf{V}$. In Phase II, we learn the skeleton for each $\mathcal{I}$-augmented graph $\mathcal{G}_{\mathbf{I}}$. For each $\mathbf{I} \in \mathcal{I}$, we retrieve the $\mathcal{I}$-augmented graph $\mathcal{G}_{\mathbf{I}}$ with all circle edges. Algorithm 3 tests if there is a separating set between any pair of nodes $X, Y$ in $\mathcal{G}_{\mathbf{I}}$. If both nodes are non-$F$ nodes, this can be tested by checking whether $P_{\mathbf{I}}(y|\mathbf{w}, x)$ and $P_{\mathbf{I}}(y|\mathbf{w})$. If one of them is an $F$ node, this can be tested by checking the equality between $P_{\mathbf{I}}(y|\mathbf{w})$ and $P_{\mathbf{J}}(y|\mathbf{w})$ for $Y \in \mathbf{V}, \mathbf{W} \subseteq \mathbf{V}$. Two $F$ nodes are separated by default. If there is no such set $\mathbf{W}$, we preserve the circle edge between $X$ and $Y$. Otherwise, we remove the circle edge from $\mathcal{G}_{\mathbf{I}}$. In Phase III, we apply orientation rules to learn more edges in the constructed $\mathcal{I}$-augmented graphs. Notice that the skeleton $G^{(\mathbf{I})}$ in $\mathcal{G}_{\mathbf{I}}$ is a PAG; thus, the FCI rules are still applicable here. We first use Rule 0 to orient all the unshielded triples by checking if the node in the center of the triple is in the separating set of the two end nodes. After that, Algorithm 1 will repeatedly apply the FCI rules (Rules 5 to 7 are not included here as they are related to selection bias nodes) together with 4 new rules until none apply. Here we briefly describe the intuition of Rule 9, 10, and 11 (Rule 8 is sound by construction).
**Rule 9 (Intervened nodes):** The intuition of this rule is that since $X$ is intervened, all the non-descendants of $X$ become separable from $X$ in $\mathcal{D}_{\bar{\mathbf{I}}}$. Thus, $Y$ has to be a descendant of $X$ in $\mathcal{D}_{\bar{\mathbf{I}}}$.
**Rule 10 (Consistency of Skeletons):** The intuition of this rule stems from the fact that each skeleton is obtained from the same causal graph $\mathcal{D}$. The ancestral relationship between any pair of nodes

cannot be reversed by hard interventions.

**Rule 11 (Inducing Path):** The intuition is that the $F$ nodes cannot be separated from $Y \notin \mathbf{K}$, meaning there is an inducing path to $Y^{(\mathbf{I})}$ or $Y^{(\mathbf{J})}$ through $X^{(\mathbf{I})}$ or $X^{(\mathbf{J})}$. If $X$ is intervened, the inducing path cannot go through $X^{(\mathbf{J})}$.

To illustrate how each step in Algorithm 1 works, we show an example in Appendix E. We establish the soundness of the proposed algorithm.

**Theorem 6.3.** *Consider a set of interventional distributions $(P_\mathbf{I})_{\mathbf{I} \in \mathcal{I}}$ that are h-faithful to a causal graph $\mathcal{D} = (\mathbf{V} \cup \mathbf{L}, \mathbf{E})$, where $\mathcal{I}$ is a set of intervention targets. Algorithm 1 is sound, i.e., every adjacency and arrowhead/tail orientation in the returned $\mathcal{I}$-augmented graph $\mathcal{G}_\mathbf{I}(\mathcal{D}, \mathcal{I})$ is common for all $\mathcal{I}$-augmented MAGs of $\mathcal{D}'$, $\mathcal{G}_\mathbf{I}(\mathcal{D}', \mathcal{I})$ for any $\mathcal{D}'$ which is $\mathcal{I}$-Markov equivalent to $\mathcal{D}$.*

## 7 Experiments

In this experiment, we compare the $\mathcal{I}$-MEC size under hard and soft interventions. For a given number of observable nodes $n$, we create an arbitrary ADMG by first constructing a DAG and then adding bidirected edges to it. Then, we

Table 1: Comparison of $\mathcal{I}$-MEC size under hard and soft interventions

| $n$ | Mean of Hard | Mean of Soft | Graph | Ratio |
|---|---|---|---|---|
| 2 | $2.03 \pm 0.15$ | $2.93 \pm 0.29$ | Random | $0.69 \pm 0.05$ |
| 2 | $2.37 \pm 0.12$ | $3.67 \pm 0.22$ | Complete | $0.65 \pm 0.05$ |
| 3 | $19.50 \pm 3.41$ | $30.57 \pm 4.36$ | Random | $0.64 \pm 0.11$ |
| 3 | $14.03 \pm 2.69$ | $24.70 \pm 4.12$ | Complete | $0.57 \pm 0.05$ |
| 4 | $677.13 \pm 227.72$ | $1218.83 \pm 361.83$ | Random | $0.56 \pm 0.18$ |
| 4 | $721.37 \pm 276.36$ | $1529.57 \pm 368.68$ | Complete | $0.47 \pm 0.07$ |

enumerate all ADMGs of the same size and check if the ADMG is in the $\mathcal{I}$-MEC. For hard interventions, we construct the $\mathcal{I}$-augmented MAGs according to the steps in Definition 5.1, and then check if Theorem 4.7 holds. For soft interventions, we refer to the construction in Definition 4 and criteria in Theorem 2 in Kocaoglu et al. [2019]. We count the number of ADMGs that are in the $\mathcal{I}$-MEC and take average over 50 random ADMGs and compute the standard error. The results are shown in Table 1. 'Complete' means the DAG is complete while 'Random' means the DAG has a density of $0.5$. It is obvious that on average, the size of $\mathcal{I}$-MEC is smaller under hard interventions and the ratio tends to decrease while $n$ increases, meaning hard interventions become more powerful for larger graphs. We can only enumerate the ADMGs for small graphs as the total number grows super-exponentially with $n$. For the experiment details and additional experiments, see Appendix F.

## 8 Conclusion

We address the challenge of learning the causal structure underlying a phenomenon of interest using a combination of several experimental data. Different from Kocaoglu et al. [2019], we study the problem under hard interventions. The motivation comes from the observation that hard interventions provide more information about the causal graph than soft ones. We verify the observation through empirical experiments that compare the $\mathcal{I}$-MEC size under hard and soft interventions (Appendix F). Our approach builds on a generalization of the converse of Pearl's do-calculus, which introduces new tests that can be applied to data. These tests translate into structural constraints. We define $\mathcal{I}$-MEC based on these criteria (Definition 4.1) and provide a graphical characterization for the equivalence of two causal graphs (Theorem 4.7) using the proposed twin augmented MAG structure (Definition 4.5). To construct a unified graphical representation that is closer to the ground truth ADMG, we combine the twin augmented MAGs into an $\mathcal{I}$-augmented MAG (Definition 5.1) and show the equivalence of the two representations (Proposition 5.2). Finally, we propose an algorithm (Algorithm 1) to learn the interventional equivalence class represented by the $\mathcal{I}$-augmented graphs (Definition 6.2) from data, incorporating novel orientation rules. We also prove the soundness of the proposed learning algorithm (Theorem 6.3). The proofs can be found in Appendix B.

## Acknowledgement

This research has been supported in part by NSF CAREER 2239375, IIS 2348717, Amazon Research Award, Adobe Research and Intuit.

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

# Appendix Contents

## A  Detailed Related Works

**Equivalence Class:** The learning process uses data constraints to infer the causal diagram. However, these constraints often cannot uniquely identify the complete diagram. As a result, the analysis typically focuses on an equivalence class (EC) of causal diagrams that captures the constraints implied by the underlying causal system. When only observational data is available, such EC is called MEC. It characterizes causal graphs with the same set of d-separation statements over observable variables [Spirtes et al., 2001, Verma and Pearl, 1992, Spirtes et al., 2013, Meek, 2013a]. Under causal sufficiency, Hauser and Bühlmann [2012], Yang et al. [2018] characterizes the $\mathcal{I}$-MEC. Tian and Pearl [2001] first considers the equivalence class under local changes. When there are latents under soft intervention with known targets, Kocaoglu et al. [2019] characterize the $\mathcal{I}$-MEC while with unknown targets, it is called $\psi$-MEC [Jaber et al., 2020]. When there is access to multiple domains, Li et al. [2023] propose S-MEC.

**Learning from Combined Datasets:** There are plenty of works in the literature on learning the causal structure from experiments (or across domains). However, most of them provide empirical approaches without theoretical analysis. Approaches like Perry et al. [2022], Peters et al. [2016], Ghassami et al. [2017], Heinze-Deml et al. [2018], Huang et al. [2020] often assume Markovianity or specific functional models (e.g., linearity), or combinations of observational and interventional data with both known and unknown targets. In contrast, JCI combines all data into a single dataset and performs learning on the pooled data [Mooij et al., 2020]. Ke et al. [2019], Brouillard et al. [2020] introduces neural-network-based frameworks that leverages observational and interventional data to identify causal structures, even when intervention targets are unknown, demonstrating superior performance in structure recovery tasks but there is no soundness analysis. Acharya et al. [2018] demonstrates that with $O(\log n)$ interventions and $O(n/\epsilon^2)$ samples per intervention, one can effectively distinguish whether an unknown CBN matches a given model or differs by more than $\epsilon$ in total variation distance. Jiang and Aragam [2023] offers the first results to characterize conditions under which causal representations are identifiable without parametric assumptions, even in settings with unknown interventions and without assuming faithfulness. Addanki et al. [2021] aims to determine the directions of all causal or ancestral relations in $G$ using a minimum-cost set of interventions. GIES [Hauser and Bühlmann, 2012] and IGSP [Wang et al., 2017] are score-based and aim at learning a single DAG under known targets; there is no soundness or equivalence class analysis either. Lopez et al. [2022] developed a differentiable causal discovery for large and high-dimensional data. Tigas et al. [2022] proposed an experiment design algorithm to adaptively choose the intervention targets. Zhou et al. [2024], Mascaro and Castelletti [2023] proposes Bayesian causal discovery algorithms under causal sufficiency.

## B  Proofs

### B.1  Proof for Proposition 3.2

In this section, we extend the do-calculus rules to enable their application across two arbitrary interventions. This extension is crucial for characterizing our equivalence class when arbitrary sets of interventional distributions are provided.

*Proof.* **Rule 1:** The results follow from rule 1 of Theorem 3.1.

**Rule 2:** From $(\mathbf{Y} \perp\!\!\!\perp \mathbf{K_J}|\mathbf{W}, \mathbf{I})_{\mathcal{D}_{\bar{\mathbf{I}}, \underline{\mathbf{K_J}}}}$, we can derive from rule 2 of Theorem 3.1 that $P_\mathbf{I}(\mathbf{y}|\mathbf{w}, \mathbf{z}) = P_{\mathbf{I},\mathbf{J}}(\mathbf{y}|\mathbf{w}, \mathbf{z})$. Similarly, given $(\mathbf{Y} \perp\!\!\!\perp \mathbf{K_I}|\mathbf{W}, \mathbf{J})_{\mathcal{D}_{\bar{\mathbf{J}}, \underline{\mathbf{K_I}}}}$, we have $P_\mathbf{J}(\mathbf{y}|\mathbf{w}, \mathbf{z}) = P_{\mathbf{I},\mathbf{J}}(\mathbf{y}|\mathbf{w}, \mathbf{z})$. Together, we have $P_\mathbf{I}(\mathbf{y}|\mathbf{w}, \mathbf{z}) = P_\mathbf{J}(\mathbf{y}|\mathbf{w}, \mathbf{z})$. Here, we use $P_{\mathbf{I},\mathbf{J}}$ as the intermediate distribution to show the equality and use rule 2 in Theorem 3.1 twice. This is necessary, as for any $\mathbf{S} \subset \mathbf{I} \cup \mathbf{J}$, $\mathcal{D}_{\overline{\mathbf{S}}}$ would be a denser graph that contains fewer m-separation statements than $\mathcal{D}_{\overline{\mathbf{I},\mathbf{J}}}$. Next, we show that it is also sufficient.

**Lemma B.1.** *If* $\exists \mathbf{S} \subset \mathbf{W} \setminus (\mathbf{I} \cup \mathbf{J})$*, s.t.* $(\mathbf{Y} \perp\!\!\!\perp \mathbf{K}_J, \mathbf{S}|\mathbf{W}, \mathbf{I})_{\mathcal{D}_{\bar{\mathbf{I}}, \underline{\mathbf{K_J}}, \mathbf{S}}} \wedge (\mathbf{Y} \perp\!\!\!\perp \mathbf{K_I}, \mathbf{S}|\mathbf{W}, \mathbf{J})_{\mathcal{D}_{\bar{\mathbf{J}}, \underline{\mathbf{K_I}}, \mathbf{S}}}$, *then* $(\mathbf{Y} \perp\!\!\!\perp \mathbf{K_J}|\mathbf{W}, \mathbf{I})_{\mathcal{D}_{\bar{\mathbf{I}}, \underline{\mathbf{K_J}}}} \wedge (\mathbf{Y} \perp\!\!\!\perp \mathbf{K_I}|\mathbf{W}, \mathbf{J})_{\mathcal{D}_{\bar{\mathbf{J}}, \underline{\mathbf{K_I}}}}$.

*Proof.* Suppose otherwise $(\mathbf{Y} \not\perp\!\!\!\perp \mathbf{K}_J|\mathbf{W}, \mathbf{I})_{\mathcal{D}_{\bar{\mathbf{I}}, \underline{\mathbf{K_J}}}}$, then there is an m-connecting path $p$ from $Y \in \mathbf{Y}$ to $U \in \mathbf{K_J}$ in $\mathcal{D}_{\bar{\mathbf{I}}, \underline{\mathbf{K_J}}}$. Given $(\mathbf{Y} \perp\!\!\!\perp \mathbf{K_J}, \mathbf{S}|\mathbf{W}, \mathbf{I})_{\mathcal{D}_{\bar{\mathbf{I}}, \underline{\mathbf{K_J}}, \mathbf{S}}}$, we have $(\mathbf{Y} \perp\!\!\!\perp \mathbf{K}_J|\mathbf{W}, \mathbf{I})_{\mathcal{D}_{\bar{\mathbf{I}}, \underline{\mathbf{K_J}}, \mathbf{S}}}$.

Comparing $(\mathbf{Y} \not\perp\!\!\!\perp \mathbf{K}_J|\mathbf{W}, \mathbf{I})_{\mathcal{D}_{\overline{\mathbf{I}}, \underline{\mathbf{K_J}}}}$ and $(\mathbf{Y} \perp\!\!\!\perp \mathbf{K_J}|\mathbf{W}, \mathbf{I})_{\mathcal{D}_{\overline{\mathbf{I}}, \underline{\mathbf{K_J}}, \mathbf{S}}}$, the only difference between the two statements is the edges outgoing from $\mathbf{S}$. Removing the edges creates $p$. Consider the case that $S \in \mathbf{S}$ is in $p$. For $p$ to be m-connecting, $S$ can only be a collider. However, such a m-connecting path indicates that there is also a m-connecting path from $Y$ to $S$ which contradicts the given conditions. Thus, $\mathbf{S}$ cannot be in $p$. Then, there has to be a collider in $p$ activated by some $S \in \mathbf{S}$ in $\mathcal{D}_{\overline{\mathbf{I}}, \underline{\mathbf{K_J}}}$. Consider a collider $C$ that is closest to $Y$ in $p$. $C$ is an ancestor of $S$. Then the path created by concatenating the subpath from $Y$ to $C$ and $C$ to $S$ is an m-connecting path which is also in $\mathcal{D}_{\overline{\mathbf{I}}, \underline{\mathbf{K_J}}, \mathbf{S}}$. This is a contradiction. Following the same process we can also show that $(\mathbf{Y} \perp\!\!\!\perp \mathbf{K_I}|\mathbf{W}, \mathbf{J})_{\mathcal{D}_{\overline{\mathbf{J}}, \underline{\mathbf{K_I}}}}$. $\qquad\square$

Lemma B.1 shows that the graphical conditions for $P_{\mathbf{I}}(\mathbf{y}|\mathbf{w}, \mathbf{z}) = P_{\mathbf{I}, \mathbf{J}}(\mathbf{y}|\mathbf{w}, \mathbf{z}) = P_{\mathbf{J}}(\mathbf{y}|\mathbf{w}, \mathbf{z})$ are sufficient to use another interventional distribution $P_{\mathbf{I}, \mathbf{J}, \mathbf{S}}$ as the intermediate distribution to show the equality.

**Rule 3:** From $(\mathbf{Y} \perp\!\!\!\perp \mathbf{K_J}|\mathbf{W}, \mathbf{I})_{\mathcal{D}_{\overline{\mathbf{I}, \mathbf{K_J}(\mathbf{W})}}}$, we can tell using rule 3 of Theorem 3.1 that $P_{\mathbf{I}}(\mathbf{y}|\mathbf{w}) = P_{\mathbf{I}, \mathbf{J}}(\mathbf{y}|\mathbf{w})$. From $(\mathbf{Y} \perp\!\!\!\perp \mathbf{K_I}|\mathbf{W}, \mathbf{J})_{\mathcal{D}_{\overline{\mathbf{J}, \mathbf{K_I}(\mathbf{W})}}}$, we have $P_{\mathbf{I}, \mathbf{J}}(\mathbf{y}|\mathbf{w}) = P_{\mathbf{J}}(\mathbf{y}|\mathbf{w})$. Together, we have $P_{\mathbf{I}}(\mathbf{y}|\mathbf{w}) = P_{\mathbf{J}}(\mathbf{y}|\mathbf{w})$. Similarly, we apply the do-do rule twice and use $P_{\mathbf{I}, \mathbf{J}}$ as an intermediate distribution to show the equality. Likewise, any distribution that corresponds to a denser graph indicates the conditions. Next, we show the sufficiency of the conditions.

**Lemma B.2.** *If* $\exists \mathbf{S} \subset \mathbf{V} \setminus (\mathbf{I} \cup \mathbf{J} \cup \mathbf{W})$, *such that* $(\mathbf{Y} \perp\!\!\!\perp \mathbf{K_J}, \mathbf{S}|\mathbf{W}, \mathbf{I})_{\mathcal{D}_{\overline{\mathbf{I}, \mathbf{K_J}(\mathbf{W})}, \mathbf{S}(\mathbf{W})}} \wedge (\mathbf{Y} \perp\!\!\!\perp \mathbf{K_I}, \mathbf{S}|\mathbf{W}, \mathbf{J})_{\mathcal{D}_{\overline{\mathbf{J}, \mathbf{K_I}(\mathbf{W})}, \mathbf{S}(\mathbf{W})}}$, *then* $(\mathbf{Y} \perp\!\!\!\perp \mathbf{K_J}|\mathbf{W}, \mathbf{I})_{\mathcal{D}_{\overline{\mathbf{I}, \mathbf{K_J}(\mathbf{W})}}} \wedge (\mathbf{Y} \perp\!\!\!\perp \mathbf{K_I}|\mathbf{W}, \mathbf{J})_{\mathcal{D}_{\overline{\mathbf{J}, \mathbf{K_I}(\mathbf{W})}}}$.

*Proof.* Suppose otherwise $(\mathbf{Y} \not\perp\!\!\!\perp \mathbf{K_J}|\mathbf{W}, \mathbf{I})_{\mathcal{D}_{\overline{\mathbf{I}, \mathbf{K_J}(\mathbf{W})}}}$, then there is an m-connecting path $p$ from $Y \in \mathbf{Y}$ to $U \in \mathbf{K_J}(\mathbf{W})$ in $\mathcal{D}_{\overline{\mathbf{I}, \mathbf{K_J}(\mathbf{W})}}$. Given $(\mathbf{Y} \perp\!\!\!\perp \mathbf{K_J}, \mathbf{S}|\mathbf{W}, \mathbf{I})_{\mathcal{D}_{\overline{\mathbf{I}, \mathbf{K_J}(\mathbf{W})}, \mathbf{S}(\mathbf{W})}}$, we have $(\mathbf{Y} \perp\!\!\!\perp \mathbf{K_J}|\mathbf{W}, \mathbf{I})_{\mathcal{D}_{\overline{\mathbf{I}, \mathbf{K_J}(\mathbf{W})}, \mathbf{S}(\mathbf{W})}}$. Comparing $(\mathbf{Y} \perp\!\!\!\perp \mathbf{K_J}|\mathbf{W}, \mathbf{I})_{\mathcal{D}_{\overline{\mathbf{I}, \mathbf{K_J}(\mathbf{W})}, \mathbf{S}(\mathbf{W})}}$ and $(\mathbf{Y} \not\perp\!\!\!\perp \mathbf{K_J}|\mathbf{W}, \mathbf{I})_{\mathcal{D}_{\overline{\mathbf{I}, \mathbf{K_J}(\mathbf{W})}}}$, the only difference in the graph is the edges into $\mathbf{S}(\mathbf{W})$. Consider the case that there is some $S \in \mathbf{S}(\mathbf{W})$ in $p$. Consider $S$ that is closest to $Y$. $S$ cannot be a collider on $p$ since it is not an ancestor of any $W \in \mathbf{W}$, and thus $p$ will be blocked. Let us consider the 2 cases:

If $S$ has an outgoing edge towards $Y$ on $p$, then there is a directed path from $S$ to $Y$ because otherwise there has to be a collider $C \in \mathbf{W}$ in between $Y$ and $S$, and $S$ will be an ancestor of $C$. However, the subpath from $Y$ to $S$ in $p$ would then be m-connecting in $\mathcal{D}_{\overline{\mathbf{I}, \mathbf{K_J}(\mathbf{W})}, \mathbf{S}(\mathbf{W})}$, which is a contradiction.

If $S$ has an incoming edge from $Y$ on $p$, then there is a directed path $S$ to $U$ on $p$. Otherwise, there has to be a collider $C$ in the subpath between $S$ and $U$ on $p$. Such $C$ that is closest to $S$ makes $S$ an ancestor of $C \in \mathbf{W}$, a contradiction. Nevertheless, if there is a directed path from $S$ to $U$, $S$ is an ancestor of $U$. If $U$ is an ancestor of any $W \in \mathbf{W}$, $S$ will also be an ancestor of $W$, a contradiction. Thus $U$ cannot be an ancestor of any $W \in \mathbf{W}$, and the edges into $U$ will be removed in $\mathcal{D}_{\overline{\mathbf{I}, \mathbf{K_J}(\mathbf{W})}}$ which contradicts the assumption that $p$ is an m-connecting path.

Therefore, $S$ cannot be on $p$. While $S$ has to be an ancestor of some $W \in \mathbf{W}$ to activate $p$ if it is not on $p$, this contradicts the assumption that $S \in \mathbf{S}(\mathbf{W})$. To conclude, we show that $(\mathbf{Y} \perp\!\!\!\perp \mathbf{K_J}|\mathbf{W}, \mathbf{I})_{\mathcal{D}_{\overline{\mathbf{I}, \mathbf{K_J}(\mathbf{W})}}}$. Following the same process, we can also show that $(\mathbf{Y} \perp\!\!\!\perp \mathbf{K_I}|\mathbf{W}, \mathbf{J})_{\mathcal{D}_{\overline{\mathbf{J}, \mathbf{K_I}(\mathbf{W})}}}$. $\qquad\square$

Lemma B.2 shows that the graphical conditions for $P_{\mathbf{I}}(\mathbf{y}|\mathbf{w}) = P_{\mathbf{I}, \mathbf{J}}(\mathbf{y}|\mathbf{w}) = P_{\mathbf{J}}(\mathbf{y}|\mathbf{w})$ are sufficient to use another interventional distribution $P_{\mathbf{I}, \mathbf{J}, \mathbf{S}}$ as the intermediate distribution to show the equality by using rule 3 of Theorem 3.1 twice.

**Rule 4:** We begin by introducing a useful lemma.

**Lemma B.3.** *For disjoint* $\mathbf{X}, \mathbf{Y}, \mathbf{W} \subseteq \mathbf{V}, \mathbf{S} \subseteq \mathbf{V} \setminus (\mathbf{X} \cup \mathbf{Y}), \mathbf{S} \cap \mathbf{W} \neq \emptyset, \mathbf{S} \neq \mathbf{W}$, *then* $P_{\mathbf{x}}(\mathbf{y}|\mathbf{w}) = P_{\mathbf{x}, \mathbf{s}}(\mathbf{y}|\mathbf{w})$ *if:* $(\mathbf{S_R} \perp\!\!\!\perp \mathbf{Y}|\mathbf{W}, \mathbf{X})_{\mathcal{D}_{\overline{\mathbf{X}, \mathbf{S_R}(\mathbf{W})}}} \wedge (\mathbf{S_W} \perp\!\!\!\perp \mathbf{Y}|\mathbf{W} \setminus \mathbf{S_W}, \mathbf{X})_{\mathcal{D}_{\overline{\mathbf{X}}, \underline{\mathbf{S_W}}}}$ *where* $\mathbf{S_W} = \mathbf{S} \cap \mathbf{W}, \mathbf{S_R} = \mathbf{S} \setminus \mathbf{W}$.

*Proof.* Denote the statement $(\mathbf{S_R} \perp\!\!\!\perp \mathbf{Y}|\mathbf{W}, \mathbf{X})_{\mathcal{D}_{\overline{\mathbf{X}, \mathbf{S_R}(\mathbf{W})}}} \wedge (\mathbf{S_W} \perp\!\!\!\perp \mathbf{Y}|\mathbf{W} \setminus \mathbf{S_W}, \mathbf{X})_{\mathcal{D}_{\overline{\mathbf{X}}, \underline{\mathbf{S_W}}}}$ as $\mathcal{C}_1$. We first show that, by applying rule 3 and rule 2 in Theorem 3.1 sequentially, we can transform from $P_{\mathbf{x}}$ to $P_{\mathbf{x}, \mathbf{s}}$. Specifically, $P_{\mathbf{x}}(\mathbf{y}|\mathbf{w}) = P_{\mathbf{x}, \mathbf{s_R}}(\mathbf{y}|\mathbf{w})$ if $(\mathbf{S_R} \perp\!\!\!\perp \mathbf{Y}|\mathbf{W}, \mathbf{X})_{\mathcal{D}_{\overline{\mathbf{X}, \mathbf{S_R}(\mathbf{W})}}}$,

and $P_{\mathbf{x},\mathbf{s_R}}(\mathbf{y}|\mathbf{w}) = P_{\mathbf{x},\mathbf{s_R},\mathbf{s_W}}(\mathbf{y}|\mathbf{w})$ if $(\mathbf{S_W} \perp\!\!\!\perp \mathbf{Y}|\mathbf{W} \setminus \mathbf{S_W}, \mathbf{X}, \mathbf{S_R})_{\mathcal{D}_{\overline{\mathbf{X},\mathbf{S_R}},\underline{\mathbf{s_W}}}}$. By definition, $P_{\mathbf{x},\mathbf{s_R},\mathbf{s_W}}(\mathbf{y}|\mathbf{w}) = P_{\mathbf{x},\mathbf{s}}(\mathbf{y}|\mathbf{w})$.

Denote the statement $(\mathbf{S_R} \perp\!\!\!\perp \mathbf{Y}|\mathbf{W}, \mathbf{X})_{\mathcal{D}_{\overline{\mathbf{X},\mathbf{S_R}(\mathbf{W})}}} \wedge (\mathbf{S_W} \perp\!\!\!\perp \mathbf{Y}|\mathbf{W} \setminus \mathbf{S_W}, \mathbf{X}, \mathbf{S_R})_{\mathcal{D}_{\overline{\mathbf{X},\mathbf{S_R}},\underline{\mathbf{s_W}}}}$ as $\mathcal{C}_2$. Next, we show that $\mathcal{C}_1$ and $\mathcal{C}_2$ are equivalent. Notice that they share the same statement, and we just need to show that the other one holds true.

$(\mathcal{C}_1 \Rightarrow \mathcal{C}_2)$: Suppose otherwise, $(\mathbf{S_W} \not\perp\!\!\!\perp \mathbf{Y}|\mathbf{W}\setminus\mathbf{S_W}, \mathbf{X}, \mathbf{S_R})_{\mathcal{D}_{\overline{\mathbf{X},\mathbf{S_R}},\underline{\mathbf{s_W}}}}$, then there is an m-connecting path $p$ from $Y \in \mathbf{Y}$ to $U \in \mathbf{S_W}$ in $\mathcal{D}_{\overline{\mathbf{X},\mathbf{S_R}},\underline{\mathbf{s_W}}}$. Comparing $(\mathbf{S_W} \not\perp\!\!\!\perp \mathbf{Y}|\mathbf{W} \setminus \mathbf{S_W}, \mathbf{X}, \mathbf{S_R})_{\mathcal{D}_{\overline{\mathbf{X},\mathbf{S_R}},\underline{\mathbf{s_W}}}}$ and $(\mathbf{S_W} \perp\!\!\!\perp \mathbf{Y}|\mathbf{W} \setminus \mathbf{S_W}, \mathbf{X})_{\mathcal{D}_{\overline{\mathbf{X}},\underline{\mathbf{s_W}}}}$, the difference is the $\mathbf{S_R}$ in the conditioning set and the edges into $\mathbf{S_R}$. Consider the case that there is some $S \in \mathbf{S_R}$ in $p$. Since the edges into $S$ are removed, $S$ can only have outgoing edges, but conditioning on $S$ will then block $p$. Thus, $S$ cannot be in $p$. If $S$ is not in $p$, since the edges into $S$ are removed, conditioning on $S$ will not activate any path. Thus the supposition cannot hold.

$(\mathcal{C}_2 \Rightarrow \mathcal{C}_1)$: Suppose otherwise, $(\mathbf{S_W} \not\perp\!\!\!\perp \mathbf{Y}|\mathbf{W} \setminus \mathbf{S_W}, \mathbf{X})_{\mathcal{D}_{\overline{\mathbf{X}},\underline{\mathbf{s_W}}}}$. Then there is an m-connecting path $p$ from $Y \in \mathbf{Y}$ to $U \in \mathbf{S_W}$ in $\mathcal{D}_{\overline{\mathbf{X}},\underline{\mathbf{s_W}}}$. Comparing $(\mathbf{S_W} \perp\!\!\!\perp \mathbf{Y}|\mathbf{W} \setminus \mathbf{S_W}, \mathbf{X}, \mathbf{S_R})_{\mathcal{D}_{\overline{\mathbf{X},\mathbf{S_R}},\underline{\mathbf{s_W}}}}$ and $(\mathbf{S_W} \not\perp\!\!\!\perp \mathbf{Y}|\mathbf{W} \setminus \mathbf{S_W}, \mathbf{X})_{\mathcal{D}_{\overline{\mathbf{X}},\underline{\mathbf{s_W}}}}$, the difference is the $\mathbf{S_R}$ in the conditioning set and the edges into $\mathbf{S_R}$. Consider the case that there is some $S \in \mathbf{S_R}$ that is closest to $Y$ in $p$.

If $S$ is a colliser on $p$, then $S$ is an ancestor of some $W \in \mathbf{W}$. There is a m-connecting path from $S$ to $Y$ in $\mathcal{D}_{\overline{\mathbf{X},\mathbf{S_R}(\mathbf{W})}}$. A contradiction.

If $S$ has an outgoing edge towards $Y$ in $p$, there has to be a collider $C \notin \mathbf{W}$ between $S$ and $Y$ in $p$. Otherwise, the subpath between $S$ and $Y$ will be m-connecting in $\mathcal{D}_{\overline{\mathbf{X},\mathbf{S_R}(\mathbf{W})}}$ which contradicts the supposition. Since $p$ is m-connecting, $C$ has to be an ancestor of some $W \in \mathbf{W}$ in $\mathcal{D}_{\overline{\mathbf{X}},\underline{\mathbf{s_W}}}$. To block $p$ in $\mathcal{D}_{\overline{\mathbf{X},\mathbf{S_R}},\underline{\mathbf{s_W}}}$, there has to be some $S' \in \mathbf{S}$ that is in between $C$ and $W$. This will create an m-connecting path from $S'$ to $Y$ in $\mathcal{D}_{\overline{\mathbf{X},\mathbf{S_R}(\mathbf{W})}}$, which is a contradiction.

If $S$ has an outgoing edge towards $S_W$ in $p$, then $S$ has to be an ancestor of some $W \in \mathbf{W}$ in $\mathcal{D}_{\overline{\mathbf{X},\mathbf{S_R}(\mathbf{W})}}$, which is a contradiction.

Therefore, $S$ is not in $p$. There has to be a collider $C \notin \mathbf{W}$ in $p$. Consider such $C$ closest to $Y$. There is a directed path from $C$ to $S$ and a directed path from $S$ to some $W \in \mathbf{W}$ in $\mathcal{D}_{\overline{\mathbf{X}},\underline{\mathbf{s_W}}}$ for $p$ to be m-connecting. However, this makes $S$ an ancestor of $W$ in $\mathcal{D}_{\overline{\mathbf{X},\mathbf{S_R}(\mathbf{W})}}$, and the path created by concatenating the directed path from $C$ to $S$ and the subpath from $C$ to $Y$ in $p$ would be d-connecting in $\mathcal{D}_{\overline{\mathbf{X},\mathbf{S_R}(\mathbf{W})}}$. A contradiction. The supposition does not hold.

This concludes the proof of this lemma. $\qquad\square$

By applying Lemma B.3 twice, we can derive the graphical condition for $P_{\mathbf{I}}(\mathbf{y}|\mathbf{w}) = P_{\mathbf{I},\mathbf{J}}(\mathbf{y}|\mathbf{w}) = P_{\mathbf{J}}(\mathbf{y}|\mathbf{w})$ for two arbitrary interventions $\mathbf{I}, \mathbf{J}$. The following lemma shows that this is a sufficient condition.

**Lemma B.4.** *If* $\exists \mathbf{S} \subset \mathbf{V} \setminus (\mathbf{I} \cup \mathbf{J}), \mathbf{S_R} = \mathbf{S} \setminus \mathbf{W}, \mathbf{S_W} = \mathbf{S} \cap \mathbf{W}$, *such that* $(\mathbf{Y} \perp\!\!\!\perp \mathbf{R_J}, \mathbf{S_R}|\mathbf{W}, \mathbf{I})_{\mathcal{D}_{\overline{\mathbf{I},\mathbf{K_J}(\mathbf{W}),\mathbf{S_R}(\mathbf{W})}}} \wedge (\mathbf{Y} \perp\!\!\!\perp \mathbf{W_J}, \mathbf{S_W}|\mathbf{W} \setminus (\mathbf{S_W} \cup \mathbf{W_J}), \mathbf{I})_{\mathcal{D}_{\overline{\mathbf{I}},\underline{\mathbf{w_J},\mathbf{s_W}}}}$, *then* $(\mathbf{Y} \perp\!\!\!\perp \mathbf{R_J}|\mathbf{W}, \mathbf{I})_{\mathcal{D}_{\overline{\mathbf{I},\mathbf{R_J}(\mathbf{W})}}} \wedge (\mathbf{Y} \perp\!\!\!\perp \mathbf{W_J}|\mathbf{W} \setminus \mathbf{W_J}, \mathbf{I})_{\mathcal{D}_{\overline{\mathbf{I}},\underline{\mathbf{w_J}}}}$.

**Proof.** We first consider $(\mathbf{Y} \perp\!\!\!\perp \mathbf{R_J}|\mathbf{W}, \mathbf{I})_{\mathcal{D}_{\overline{\mathbf{I},\mathbf{R_J}(\mathbf{W})}}}$. Suppose otherwise $(\mathbf{Y} \not\perp\!\!\!\perp \mathbf{R_J}|\mathbf{W}, \mathbf{I})_{\mathcal{D}_{\overline{\mathbf{I},\mathbf{R_J}(\mathbf{W})}}}$, then there is an m-connecting path $p$ from $Y \in \mathbf{Y}$ to $U \in \mathbf{R_J}$ in $\mathcal{D}_{\overline{\mathbf{I},\mathbf{R_J}(\mathbf{W})}}$. From the given condition, we know that $(\mathbf{Y} \perp\!\!\!\perp \mathbf{R_J}|\mathbf{W}, \mathbf{I})_{\mathcal{D}_{\overline{\mathbf{I},\mathbf{K_J}(\mathbf{W}),\mathbf{S_R}(\mathbf{W})}}}$. The only difference is the edges into $\mathbf{S_R}(\mathbf{W})$ in $\mathcal{D}_{\overline{\mathbf{I}}}$. If there is any $S \in \mathbf{S_R}$ in $p$, then it has only outgoing edges. Since $p$ is m-connecting, the subpath from $S$ to $Y$ on $p$ is also m-connecting in $\mathcal{D}_{\overline{\mathbf{I},\mathbf{R_J}(\mathbf{W}),\mathbf{S_R}(\mathbf{W})}}$, which is a contradiction. Thus $\mathbf{S_R}$ cannot be on $p$. To block $p$ in $\mathcal{D}_{\overline{\mathbf{I},\mathbf{R_J}(\mathbf{W}),\mathbf{S_R}(\mathbf{W})}}$, there is a collider $C$ in $p$ that is an ancestor of some $W \in \mathbf{W}$. $C$ is activated in $p$ in $\mathcal{D}_{\overline{\mathbf{I},\mathbf{R_J}(\mathbf{W})}}$ but not activated in $\mathcal{D}_{\overline{\mathbf{I},\mathbf{R_J}(\mathbf{W}),\mathbf{S_R}(\mathbf{W})}}$. This requires some $S \in \mathbf{S_R}(\mathbf{W})$ to be an ancestor of $W$, which is impossible. Therefore, the supposition does not hold.

Next, we consider $(\mathbf{Y} \perp\!\!\!\perp \mathbf{W_J}|\mathbf{W}\setminus\mathbf{W_J},\mathbf{I})_{\mathcal{D}_{\overline{\mathbf{I}},\underline{\mathbf{W_J}}}}$. Suppose otherwise, $(\mathbf{Y} \not\perp\!\!\!\perp \mathbf{W_J}|\mathbf{W}\setminus\mathbf{W_J},\mathbf{I})_{\mathcal{D}_{\overline{\mathbf{I}},\underline{\mathbf{W_J}}}}$, then there is an m-connecting path $p$ from $Y \in \mathbf{Y}$ to $U \in \mathbf{W_J}$ in $\mathcal{D}_{\overline{\mathbf{I}},\underline{\mathbf{W_J}}}$. $p$ is blocked in $\mathcal{D}_{\overline{\mathbf{I}},\underline{\mathbf{W_J}},\mathbf{S_W}}$. Comparing $(\mathbf{Y} \perp\!\!\!\perp \mathbf{W_J}|\mathbf{W}\setminus\mathbf{W_J},\mathbf{I})_{\mathcal{D}_{\overline{\mathbf{I}},\underline{\mathbf{W_J}}}}$ and $(\mathbf{Y} \perp\!\!\!\perp \mathbf{W_J}|\mathbf{W}\setminus(\mathbf{S_W}\cup\mathbf{W_J}),\mathbf{I})_{\mathcal{D}_{\overline{\mathbf{I}\mathbf{W_J}},\underline{\mathbf{S_W}}}}$, the only difference is the edges outgoing from $\mathbf{S_W}$. If there is some $S \in \mathbf{S_W}$ in $p$, then $S$ has to be a collider. Consider such $S$ closest to $Y$. Since $p$ is m-connecting, then the subpath from $S$ to $Y$ is also m-connecting in $\mathcal{D}_{\overline{\mathbf{I}\mathbf{W_J}},\underline{\mathbf{S_W}}}$, which is a contradiction. Thus, $S$ cannot be in $p$. For $S$ to block $p$ in $\mathcal{D}_{\overline{\mathbf{I}\mathbf{W_J}},\underline{\mathbf{S_W}}}$ while not in $p$, it has to be an ancestor of some $W \in \mathbf{W}$ and a descendant of some collider in $p$. However, since $S \in \mathbf{W}$, it can still activate the collider in $p$ in $\mathcal{D}_{\overline{\mathbf{I}\mathbf{W_J}},\underline{\mathbf{S_W}}}$. Therefore, the supposition does not hold. $\qquad\square$

By applying Lemma B.4 twice, we can show that it is sufficient to transfer through $P_{\mathbf{I},\mathbf{J}}$. This concludes the proof of this theorem. $\qquad\square$

## B.2 Proof for Proposition 4.4

We show the graphical conditions on the augmented pair graphs are equivalent to those given in the generalized causal calculus rules.

**Proposition B.5.** *Consider a CBN ($\mathcal{D} = (\mathbf{V}\cup\mathbf{L},\mathbf{E}),P$) with latent variables $\mathbf{L}$ and its augmented pair graph $\mathrm{Aug}_{(\mathbf{I},\mathbf{J})}(\mathcal{D}) = (\mathbf{V}^{(\mathbf{I})}\cup\mathbf{V}^{(\mathbf{J})}\cup\{F\},\mathbf{E}^{(\mathbf{I})}\cup\mathbf{E}^{(\mathbf{J})}\cup\mathcal{E})$ with respect to a pair of interventions $\mathbf{I},\mathbf{J} \in \mathcal{I}$. Let $\mathbf{S} = \mathbf{I}\Delta\mathbf{J}$ be a set of nodes, $F$ is adjacent to $\mathbf{S}^{(\mathbf{I})},\mathbf{S}^{(\mathbf{J})}$. We have the following equivalence relations:*

*Suppose disjoint $\mathbf{Y},\mathbf{Z},\mathbf{W} \subseteq \mathbf{V}$. We have*

$$(\mathbf{Y} \perp\!\!\!\perp \mathbf{Z}|\mathbf{W},\mathbf{I})_{\mathcal{D}_{\overline{\mathbf{I}}}} \iff (\mathbf{Y}^{(\mathbf{I})} \perp\!\!\!\perp \mathbf{Z}^{(\mathbf{I})}|\mathbf{W}^{(\mathbf{I})},\mathbf{I}^{(\mathbf{I})},F)_{\mathrm{Aug}_{(\mathbf{I},\mathbf{J})}(\mathcal{D})} \qquad (3)$$

*Suppose $\mathbf{Y},\mathbf{W}$ are disjoint subsets of $\mathbf{V}\setminus\mathbf{S}$. We have*

$$\left\{\begin{array}{l} (\mathbf{Y} \perp\!\!\!\perp \mathbf{W_J}|\mathbf{W}\setminus\mathbf{W_J},\mathbf{I})_{\mathcal{D}_{\overline{\mathbf{I}},\underline{\mathbf{W_J}}}} \\ (\mathbf{Y} \perp\!\!\!\perp \mathbf{W_I}|\mathbf{W}\setminus\mathbf{W_I},\mathbf{J})_{\mathcal{D}_{\overline{\mathbf{J}},\underline{\mathbf{W_I}}}} \end{array}\right. \iff \left\{\begin{array}{l} (F \perp\!\!\!\perp \mathbf{Y}^{(\mathbf{I})}|\mathbf{W}^{(\mathbf{I})},\mathbf{I}^{(\mathbf{I})})_{\mathrm{Aug}_{(\mathbf{I},\mathbf{J})}(\mathcal{D})} \\ (F \perp\!\!\!\perp \mathbf{Y}^{(\mathbf{J})}|\mathbf{W}^{(\mathbf{J})},\mathbf{J}^{(\mathbf{J})})_{\mathrm{Aug}_{(\mathbf{I},\mathbf{J})}(\mathcal{D})} \end{array}\right. \qquad (4)$$

$$\left\{\begin{array}{l} (\mathbf{Y} \perp\!\!\!\perp \mathbf{R_J}|\mathbf{W},\mathbf{I})_{\mathcal{D}_{\overline{\mathbf{I},\mathbf{R_J}(\mathbf{W})}}} \\ (\mathbf{Y} \perp\!\!\!\perp \mathbf{R_I}|\mathbf{W},\mathbf{J})_{\mathcal{D}_{\overline{\mathbf{J},\mathbf{R_I}(\mathbf{W})}}} \end{array}\right. \iff \left\{\begin{array}{l} (F \perp\!\!\!\perp \mathbf{Y}^{(\mathbf{I})}|\mathbf{I}^{(\mathbf{I})},\mathbf{W}^{(\mathbf{I})})_{\mathrm{Aug}_{(\mathbf{I},\mathbf{J})}(\mathcal{D})} \\ (F \perp\!\!\!\perp \mathbf{Y}^{(\mathbf{J})}|\mathbf{J}^{(\mathbf{J})},\mathbf{W}^{(\mathbf{J})})_{\mathrm{Aug}_{(\mathbf{I},\mathbf{J})}(\mathcal{D})} \end{array}\right. \qquad (5)$$

*For disjoint $\mathbf{Y},\mathbf{Z},\mathbf{W} \subseteq \mathbf{V}$, where $\mathbf{K_I} = \mathbf{K}\setminus\mathbf{J}, \mathbf{K_J} = \mathbf{K}\setminus\mathbf{I}, \mathbf{W_I} = \mathbf{K_I}\cap\mathbf{W}, \mathbf{W_J} = \mathbf{K_J}\cap\mathbf{W}, \mathbf{R} = \mathbf{K}\setminus\mathbf{W}, \mathbf{R_I} = \mathbf{R}\cap\mathbf{K_I}, \mathbf{R_J} = \mathbf{R}\cap\mathbf{K_J}$*

$$\left\{\begin{array}{l} (\mathbf{Y} \perp\!\!\!\perp \mathbf{R_J}|\mathbf{W},\mathbf{I})_{\mathcal{D}_{\overline{\mathbf{I},\mathbf{R_J}(\mathbf{W})}}} \\ (\mathbf{Y} \perp\!\!\!\perp \mathbf{W_J}|\mathbf{W}\setminus\mathbf{W_J},\mathbf{I})_{\mathcal{D}_{\overline{\mathbf{I}},\underline{\mathbf{W_J}}}} \\ (\mathbf{Y} \perp\!\!\!\perp \mathbf{R_I}|\mathbf{W},\mathbf{J})_{\mathcal{D}_{\overline{\mathbf{J},\mathbf{R_I}(\mathbf{W})}}} \\ (\mathbf{Y} \perp\!\!\!\perp \mathbf{W_I}|\mathbf{W}\setminus\mathbf{W_I},\mathbf{J})_{\mathcal{D}_{\overline{\mathbf{J}},\underline{\mathbf{W_I}}}} \end{array}\right. \iff \left\{\begin{array}{l} (F \perp\!\!\!\perp \mathbf{Y}^{(\mathbf{I})}|\mathbf{I}^{(\mathbf{I})},\mathbf{W}^{(\mathbf{I})})_{\mathrm{Aug}_{(\mathbf{I},\mathbf{J})}(\mathcal{D})} \\ (F \perp\!\!\!\perp \mathbf{Y}^{(\mathbf{J})}|\mathbf{J}^{(\mathbf{J})},\mathbf{W}^{(\mathbf{J})})_{\mathrm{Aug}_{(\mathbf{I},\mathbf{J})}(\mathcal{D})} \end{array}\right. \qquad (6)$$

*Proof.* Consider Equation 3. For the right statement, conditioning on $F$ is equivalent to removing it and the subgraph induced by $\mathbf{V}^{(\mathbf{J})}$ from $\mathrm{Aug}_{(\mathbf{I},\mathbf{J})}(\mathcal{D})$. Then the statements on the two sides are equivalent.

Consider Equation 4. We need to show that $(\mathbf{Y} \perp\!\!\!\perp \mathbf{W_J}|\mathbf{W}\setminus\mathbf{W_J},\mathbf{I})_{\mathcal{D}_{\overline{\mathbf{I}},\underline{\mathbf{W_J}}}} \Leftrightarrow (F \perp\!\!\!\perp \mathbf{Y}^{(\mathbf{I})}|\mathbf{W}^{(\mathbf{I})},\mathbf{I}^{(\mathbf{I})})_{\mathrm{Aug}_{(\mathbf{I},\mathbf{J})}(\mathcal{D})}$.

($\Rightarrow$) Suppose otherwise $(F \not\perp\!\!\!\perp \mathbf{Y}^{(\mathbf{I})}|\mathbf{W}^{(\mathbf{I})},\mathbf{I}^{(\mathbf{I})})_{\mathrm{Aug}_{(\mathbf{I},\mathbf{J})}(\mathcal{D})}$, then there is an m-connecting path $p$ from $F$ to $Y^{(\mathbf{I})} \in \mathbf{Y}^{(\mathbf{I})}$ in $\mathrm{Aug}_{(\mathbf{I},\mathbf{J})}(\mathcal{D})$. Since $\mathbf{W}^{(\mathbf{I})}$ is conditioned on, $p$ cannot be a frontdoor path. Also, edges into $\mathbf{I}^{(\mathbf{I})}$ are removed, thus $p$ cannot be a backdoor path through $\mathbf{W_I}$. However, given $(\mathbf{Y} \perp\!\!\!\perp \mathbf{W_J}|\mathbf{W}\setminus\mathbf{W_J},\mathbf{I})_{\mathcal{D}_{\overline{\mathbf{I}},\underline{\mathbf{W_J}}}}$, there is no backdoor path through $\mathbf{W_J}$. The supposition does not hold.

($\Leftarrow$) Suppose otherwise $(\mathbf{Y} \not\perp\!\!\!\perp \mathbf{W_J}|\mathbf{W}\setminus\mathbf{W_J},\mathbf{I})_{\mathcal{D}_{\overline{\mathbf{I}},\underline{\mathbf{W_J}}}}$, then there is an m-connecting path $p$ from $Y^{(\mathbf{I})} \in \mathbf{Y}^{(\mathbf{I})}$ to $U \in \mathbf{W_J}$. It has to be a backdoor path from $U$. However, $(F \perp\!\!\!\perp$

$\perp \mathbf{Y^{(I)}}|\mathbf{W^{(I)}}, \mathbf{I^{(I)}})_{\mathrm{Aug}_{(\mathbf{I,J})}(\mathcal{D})}$ will not hold, if there is a backdoor path from $U$ to $Y$. Thus the supposition does not hold.

For the same reason, we can show that $(\mathbf{Y} \perp\!\!\!\perp \mathbf{W}_I|\mathbf{W} \setminus \mathbf{W_I}, \mathbf{J})_{\mathcal{D}_{\overline{\mathbf{J}}, \underline{\mathbf{W_I}}}} \Leftrightarrow (F \perp\!\!\!\perp \mathbf{Y^{(J)}}|\mathbf{W^{(J)}}, \mathbf{J^{(J)}})_{\mathrm{Aug}_{(\mathbf{I,J})}(\mathcal{D})}$.

Consider Equation 5. We need to show that $(\mathbf{Y} \perp\!\!\!\perp \mathbf{R_J}|\mathbf{W}, \mathbf{I})_{\mathcal{D}_{\overline{\mathbf{I}, \mathbf{R_J}(\mathbf{W})}}} \Leftrightarrow (F \perp\!\!\!\perp \mathbf{Y^{(I)}}|\mathbf{I^{(I)}}, \mathbf{W^{(I)}})_{\mathrm{Aug}_{(\mathbf{I,J})}(\mathcal{D})}$.

($\Rightarrow$) Suppose otherwise $(F \not\perp\!\!\!\perp \mathbf{Y^{(I)}}|\mathbf{I^{(I)}}, \mathbf{W^{(I)}})_{\mathrm{Aug}_{(\mathbf{I,J})}(\mathcal{D})}$, then there is an m-connecting path $p$ from $F$ to $Y^{(I)} \in \mathbf{Y^{(I)}}$ in $\mathrm{Aug}_{(\mathbf{I,J})}(\mathcal{D})$. Since $\mathbf{I^{(I)}}$ is conditioned on and the edges into $\mathbf{I^{(I)}}$ are removed, $p$ cannot be through $\mathbf{I^{(I)}}$. If $F$ has a frontdoor path to $Y^{(I)}$ through $\mathbf{R_J^{(I)}}$, then $Y$ to $\mathbf{R_J}$ will also be m-connecting in $\mathcal{D}_{\overline{\mathbf{I}, \mathbf{R_J}(\mathbf{W})}}$. Else if it is a backdoor path, it can only go through some $U^{(I)} \in \mathbf{R_J^{(I)}}(\mathbf{W})$; it contradicts $(\mathbf{Y} \perp\!\!\!\perp \mathbf{R_J}|\mathbf{W}, \mathbf{I})_{\mathcal{D}_{\overline{\mathbf{I}, \mathbf{R_J}(\mathbf{W})}}}$. Thus the supposition does not hold.

($\Leftarrow$) Suppose otherwise $(\mathbf{Y} \not\perp\!\!\!\perp \mathbf{R_J}|\mathbf{W}, \mathbf{I})_{\mathcal{D}_{\overline{\mathbf{I}, \mathbf{R_J}(\mathbf{W})}}}$, then there is an m-connecting path $p$ from $Y \in \mathbf{Y}$ to $U \in \mathbf{R_J}$ in $\mathcal{D}_{\overline{\mathbf{I}, \mathbf{R_J}(\mathbf{W})}}$. If $p$ has an outgoing from $U$, then $F$ would have a frontdoor path through $U^{(I)}$ to $Y^{(I)}$ that is m-connecting. This contradicts $(F \perp\!\!\!\perp \mathbf{Y^{(I)}}|\mathbf{I^{(I)}}, \mathbf{W^{(I)}})_{\mathrm{Aug}_{(\mathbf{I,J})}(\mathcal{D})}$. Thus $p$ can only be a backdoor path from $U$. If $U$ is not an ancestor of any $W \in \mathbf{W}$ in $\mathcal{D}_{\overline{\mathbf{I}}}$, the edges into $U$ are removed and $p$ do not exist in this case. Else if $U$ is an ancestor of some $W \in \mathbf{W}$, the same backdoor path would also be activated in $\mathrm{Aug}_{(\mathbf{I,J})}(\mathcal{D})$. Therefore, the supposition does not hold.

For the same reason, we can show that $(\mathbf{Y} \perp\!\!\!\perp \mathbf{R_I}|\mathbf{W}, \mathbf{J})_{\mathcal{D}_{\overline{\mathbf{J}, \mathbf{R_I}(\mathbf{W})}}} \Leftrightarrow (F \perp\!\!\!\perp \mathbf{Y^{(J)}}|\mathbf{J^{(J)}}, \mathbf{W^{(J)}})_{\mathrm{Aug}_{(\mathbf{I,J})}(\mathcal{D})}$.

Consider Equation 6. We need to show that $(\mathbf{Y} \perp\!\!\!\perp \mathbf{R_J}|\mathbf{W}, \mathbf{I})_{\mathcal{D}_{\overline{\mathbf{I}, \mathbf{R_J}(\mathbf{W})}}} \wedge (\mathbf{Y} \perp\!\!\!\perp \mathbf{W_J}|\mathbf{W} \setminus \mathbf{W_J}, \mathbf{I})_{\mathcal{D}_{\overline{\mathbf{I}}, \underline{\mathbf{W_J}}}} \Leftrightarrow (F \perp\!\!\!\perp \mathbf{Y^{(I)}}|\mathbf{I^{(I)}}, \mathbf{W^{(I)}})_{\mathrm{Aug}_{(\mathbf{I,J})}(\mathcal{D})}$.

($\Rightarrow$) Suppose otherwise $(F \not\perp\!\!\!\perp \mathbf{Y^{(I)}}|\mathbf{I^{(I)}}, \mathbf{W^{(I)}})_{\mathrm{Aug}_{(\mathbf{I,J})}(\mathcal{D})}$, then there is an m-connecting path $p$ from $F$ to $Y^{(I)} \in \mathbf{Y^{(I)}}$, in $\mathrm{Aug}_{(\mathbf{I,J})}(\mathcal{D})$. If $p$ is a frontdoor path through $\mathbf{S^{(I)}}$, since $\mathbf{I^{(I)}}, \mathbf{W^{(I)}}$ are conditioned on, only $\mathbf{R_J^{(I)}}$ could be in $p$. However, the path from $\mathbf{R_J}$ to $Y$ would be m-connecting in $\mathcal{D}_{\overline{\mathbf{I}, \mathbf{R_J}(\mathbf{W})}}$ which is a contradiction. If $p$ is a backdoor path from $F$ to $U \in \mathbf{S^{(I)}}$ to $Y^{(I)} \in \mathbf{Y^{(I)}}$, since the edges into $\mathbf{I^{(I)}}$ are removed, $U$ can only be from $\mathbf{R_J^{(I)}}$ or $\mathbf{W_J^{(I)}}$. If $U \in \mathbf{W_J^{(I)}}$, then it contradicts $(\mathbf{Y} \perp\!\!\!\perp \mathbf{W_J}|\mathbf{W} \setminus \mathbf{W_J}, \mathbf{I})_{\mathcal{D}_{\overline{\mathbf{I}}, \underline{\mathbf{W_J}}}}$. If $U \in \mathbf{R_J^{(I)}}$, $U$ could be either an ancestor of some $W^{(I)} \in \mathbf{W^{(I)}}$ or a non-ancestor of any $W^{(I)} \in \mathbf{W^{(I)}}$. For the case that $U$ is not an ancestor of any $W^{(I)} \in \mathbf{W^{(I)}}$, $p$ is blocked by $U$ as an inactivated collider. If $U$ is an ancestor of some $W^{(I)} \in \mathbf{W^{(I)}}$, then the subpath from $U$ to $\mathbf{Y^{(I)}}$ would indicate an m-connecting path from $\mathbf{R_J}$ to $Y$ in $\mathcal{D}_{\overline{\mathbf{I}}, \underline{\mathbf{W_J}}}$ which contradicts $(\mathbf{Y} \perp\!\!\!\perp \mathbf{R_J}|\mathbf{W}, \mathbf{I})_{\mathcal{D}_{\overline{\mathbf{I}, \mathbf{R_J}(\mathbf{W})}}}$. Thus, the supposition does not hold.

($\Leftarrow$) Suppose otherwise, there are two cases to consider, either $(\mathbf{Y} \not\perp\!\!\!\perp \mathbf{R_J}|\mathbf{W}, \mathbf{I})_{\mathcal{D}_{\overline{\mathbf{I}, \mathbf{R_J}(\mathbf{W})}}}$ or $(\mathbf{Y} \not\perp\!\!\!\perp \mathbf{W_J}|\mathbf{W} \setminus \mathbf{W_J}, \mathbf{I})_{\mathcal{D}_{\overline{\mathbf{I}}, \underline{\mathbf{W_J}}}}$.

If $(\mathbf{Y} \not\perp\!\!\!\perp \mathbf{W_J}|\mathbf{W} \setminus \mathbf{W_J}, \mathbf{I})_{\mathcal{D}_{\overline{\mathbf{I}}, \underline{\mathbf{W_J}}}}$, there is an m-connecting path $p$ from $Y \in \mathbf{Y}$ to $U \in \mathbf{W_J}$ in $\mathcal{D}_{\overline{\mathbf{I}}, \underline{\mathbf{W_J}}}$. $p$ cannot be a frontdoor path at $U$, since all edges outgoing from $\mathbf{W_J}$ are removed. While a valid backdoor path at $U$ indicates that there is also a valid path from $F$ to $Y^{(I)}$ through $U^{(I)}$ in $\mathrm{Aug}_{(\mathbf{I,J})}(\mathcal{D})$. Thus, this case is impossible.

If $(\mathbf{Y} \not\perp\!\!\!\perp \mathbf{W}_J|\mathbf{W} \setminus \mathbf{W}_J, \mathbf{I})_{\mathcal{D}_{\overline{\mathbf{I}}, \underline{\mathbf{W_J}}}}$, there is an m-connecting path $p$ from $Y \in \mathbf{Y}$ to $U \in \mathbf{R_J}$ in $\mathcal{D}_{\overline{\mathbf{I}, \mathbf{R_J}(\mathbf{W})}}$. $p$ cannot be a frontdoor path at $U$, because otherwise the path constructed by adding $F \rightarrow U^{(I)}$ to $p$ in $\mathrm{Aug}_{(\mathbf{I,J})}(\mathcal{D})$ will be m-connecting. Thus, $p$ can only be a backdoor path at $U$. If $U$ is not an ancestor of any $W \in \mathbf{W}$ in $\mathcal{D}_{\overline{\mathbf{I}}}$, the edges into $U$ are removed and such $p$ do not exist. Else if $U$ is an ancestor of some $W \in \mathbf{W}$ in $\mathcal{D}_{\overline{\mathbf{I}}}$, $W^{(I)}$ will activate the path constructed by adding

$F \to U^{(\mathbf{I})}$ to $p$ in $\mathrm{Aug}_{(\mathbf{I},\mathbf{J})}(\mathcal{D})$, which contradicts $(F \perp\!\!\!\perp \mathbf{Y}^{(\mathbf{I})}|\mathbf{I}^{(\mathbf{I})}, \mathbf{W}^{(\mathbf{I})})_{Aug_{(\mathbf{I},\mathbf{J})}(\mathcal{D})}$. Therefore, the supposition does not hold.

For the same reason, we can show that $(\mathbf{Y} \perp\!\!\!\perp \mathbf{R_I}|\mathbf{W}, \mathbf{J})_{\mathcal{D}_{\overline{\mathbf{J}, \mathbf{R_I}(\mathbf{W})}}} \wedge (\mathbf{Y} \perp\!\!\!\perp \mathbf{W_I}|\mathbf{W} \setminus \mathbf{W_I}, \mathbf{J})_{\mathcal{D}_{\overline{\mathbf{J}}, \underline{\mathbf{W_I}}}} \Leftrightarrow$
$(F \perp\!\!\!\perp \mathbf{Y}^{(\mathbf{J})}|\mathbf{J}^{(\mathbf{J})}, \mathbf{W}^{(\mathbf{J})})_{\mathrm{Aug}_{(\mathbf{I},\mathbf{J})}(\mathcal{D})}$. $\hfill\square$

**Proof of Proposition 4.4:** The follows from Proposition B.5. $\hfill\square$

## B.3 Proof for Lemma 4.6

*Proof.* The extra cycles contain $F$, while $F$ has only outgoing edges. Thus, the twin augmented MAGs are ancestral. Suppose there is an inducing path $\langle F, X_1, X_2, ..., X_k \rangle$, $F$ and $X_k$ are not adjacent. Then $X_{k-1}$ has a directed edge to $X_k$ in $\mathrm{MAG}(\mathrm{Aug}(\mathcal{D}))$ while $\mathrm{MAG}(\mathrm{Aug}(\mathcal{D}))$ is a MAG by definition. Thus, a contradiction arises, and the supposition does not hold. The twin augmented MAGs are maximal $\hfill\square$

## B.4 Proof for Theorem 4.7

*Proof.* **(If)** Suppose that the twin augmented MAGs $Twin_{(\mathbf{I},\mathbf{J})}(\mathcal{D}_1), Twin_{(\mathbf{I},\mathbf{J})}(\mathcal{D}_2)$ for all $\mathbf{I}, \mathbf{J} \in \mathcal{I}$ satisfy the 3 conditions. Then they induce the same m-separations and vice versa. Then by Proposition 4.4 that $\mathcal{D}_1$ and $\mathcal{D}_2$ impose the same constraints over the distribution tuples. Thus $\mathcal{P}_\mathcal{I}(\mathcal{D}_1, \mathbf{V}) = \mathcal{P}_\mathcal{I}(\mathcal{D}_2, \mathbf{V})$.

**(Only if)** Suppose for a pair of interventions $\mathbf{I}, \mathbf{J} \in \mathcal{I}$, $\mathcal{M}_1 = \mathrm{Twin}_{(\mathbf{I},\mathbf{J})}(\mathcal{D}_1), \mathcal{M}_2 = \mathrm{Twin}_{(\mathbf{I},\mathbf{J})}(\mathcal{D}_2)$ do not fully satisfy the 3 conditions. Then they must induce at least one different m-separation statement. We need to show that all the differences in m-separation statements induced by different $\mathcal{M}$ structures can be captured by some m-separation statements that are testable by the distribution tuples, and therefore, the difference in m-separation would be inducing different constraints on $\mathcal{P}_\mathcal{I}(\mathcal{D}_1, \mathbf{V})$ and $\mathcal{P}_\mathcal{I}(\mathcal{D}_2, \mathbf{V})$. Thus the condition that $\mathcal{P}_\mathcal{I}(\mathcal{D}_1, \mathbf{V}) = \mathcal{P}_\mathcal{I}(\mathcal{D}_2, \mathbf{V})$ will no longer hold, which is a contradiction.

We start by showing all testable m-separation statements. For an arbitrary twin augmented MAG $\mathcal{M} = (\mathbf{V}^{(\mathbf{I})} \cup \mathbf{V}^{(\mathbf{J})} \cup \{F\}, \mathbf{E}^{(\mathbf{I})} \cup \mathbf{E}^{(\mathbf{J})} \cup \mathcal{E})$, the testable m-separation statements are as follows:

$$\mathcal{T} = \{(\mathbf{X}^{(\mathbf{I})} \perp\!\!\!\perp \mathbf{Y}^{(\mathbf{I})}|\mathbf{Z}^{(\mathbf{I})}, \mathbf{I}^{(\mathbf{I})}, F)_\mathcal{M} : \mathbf{X}, \mathbf{Y} \subseteq \mathbf{V}, \mathbf{Z} \subseteq \mathbf{V} \setminus (\mathbf{X} \cup \mathbf{Y})\} \cup$$
$$\{(\mathbf{X}^{(\mathbf{J})} \perp\!\!\!\perp \mathbf{Y}^{(\mathbf{J})}|\mathbf{Z}^{(\mathbf{J})}, \mathbf{J}^{(\mathbf{J})}, F)_\mathcal{M} : \mathbf{X}, \mathbf{Y} \subseteq \mathbf{V}, \mathbf{Z} \subseteq \mathbf{V} \setminus (\mathbf{X} \cup \mathbf{Y})\} \cup$$
$$\{(F \perp\!\!\!\perp \mathbf{Y}^{(\mathbf{I})}|\mathbf{I}^{(\mathbf{I})}, \mathbf{Z}^{(\mathbf{I})})_\mathcal{M} \wedge (F \perp\!\!\!\perp \mathbf{Y}^{(\mathbf{J})}|\mathbf{J}^{(\mathbf{J})}, \mathbf{Z}^{(\mathbf{J})})_\mathcal{M} : \mathbf{Y} \subseteq \mathbf{V}, \mathbf{Z} \subseteq \mathbf{V} \setminus \mathbf{Y}\}$$

Next, we show that $\mathcal{M}_1$ and $\mathcal{M}_2$ should have the same skeleton.

First, we show that they have the same skeleton on $\mathbf{V}^{(\mathbf{I})}$. Suppose otherwise, in $\mathcal{M}_1$, $X^{(\mathbf{I})}$ and $Y^{(\mathbf{I})}$ are adjacent but they are non-adjacent in $\mathcal{M}_2$. Then it implies that $(X^{(\mathbf{I})} \not\perp\!\!\!\perp Y^{(\mathbf{I})}|\mathbf{Z}^{(\mathbf{I})}, F)_{\mathcal{M}_1} \wedge (X^{(\mathbf{I})} \perp\!\!\!\perp Y^{(\mathbf{I})}|\mathbf{Z}^{(\mathbf{I})}, F)_{\mathcal{M}_2}$ for some $\mathbf{Z} \subseteq \mathbf{V}$. Then we can further condition on $I^{(\mathbf{I})}$ while preserving the m-separations since edges into $\mathbf{I}^{(\mathbf{I})}$ are removed and thus conditioning on it will not activate any extra path. Therefore, we have $(X^{(\mathbf{I})} \not\perp\!\!\!\perp Y^{(\mathbf{I})}|\mathbf{Z}^{(\mathbf{I})}, \mathbf{I}^{(\mathbf{I})}, F)_{\mathcal{M}_1} \wedge (X^{(\mathbf{I})} \perp\!\!\!\perp Y^{(\mathbf{I})}|\mathbf{Z}^{(\mathbf{I})}, \mathbf{I}^{(\mathbf{I})}, F)_{\mathcal{M}_2}$ which is a pair of different testable statements in $\mathcal{T}$. Similarly, $\mathcal{M}_1$ and $\mathcal{M}_2$ also have the same skeleton in $\mathbf{V}^{(\mathbf{J})}$. What remains to be demonstrated is that the $F$ nodes share the same adjacencies in both graphs.

By the construction of twin augmented MAGs, $F$ is adjacent to $\mathbf{K}^{(\mathbf{I})}, \mathbf{K}^{(\mathbf{J})}$ in both $\mathcal{M}_1$ and $\mathcal{M}_2$. We need to show that $F$ has the same adjacencies to $X \notin \mathbf{K}$ in both graphs. Suppose otherwise, $F$ is adjacent to $X^{(\mathbf{I})}$ in $\mathcal{M}_1$, but non-adjacent to $X^{(\mathbf{I})}$ in $\mathcal{M}_2$. According to our construction of $\mathcal{M}_1, \mathcal{M}_2$, $F$ is adjacent to $X^{(\mathbf{I})}, X^{(\mathbf{J})}$ in $\mathcal{M}_1$ but non-adjacent to them in $\mathcal{M}_2$. We introduce the following lemma which shows that in this case, we can still find a pair of different testable m-separation statements which reflect this structural difference.

**Lemma B.6.** *Consider a causal graph $\mathcal{D} = (\mathbf{V} \cup \mathbf{L}, \mathbf{E})$ given a set of intervention targets $\mathcal{I} \subseteq 2^V$. Construct its twin augmented MAG $\mathcal{M} = (\mathbf{V}^{(\mathbf{I})} \cup \mathbf{V}^{(\mathbf{J})} \cup \{F\}, \mathbf{E}^{(\mathbf{I})} \cup \mathbf{E}^{(\mathbf{J})} \cup \mathcal{E})$, for $\mathbf{I}, \mathbf{J} \in \mathcal{I}$. If there exists minimal $\mathbf{W}_1, \mathbf{W}_2 \subseteq \mathbf{V}$, such that $(F \perp\!\!\!\perp X^{(\mathbf{I})}|\mathbf{I}^{(\mathbf{I})}, \mathbf{W}_1^{(\mathbf{I})})_\mathcal{M} \wedge (F \perp\!\!\!\perp X^{(\mathbf{J})}|\mathbf{J}^{(\mathbf{J})}, \mathbf{W}_2^{(\mathbf{J})})_\mathcal{M}$, then $(F \perp\!\!\!\perp X^{(\mathbf{I})}|\mathbf{I}^{(\mathbf{I})}, \mathbf{W}^{(\mathbf{I})})_\mathcal{M} \wedge (F \perp\!\!\!\perp X^{(\mathbf{J})}|\mathbf{J}^{(\mathbf{J})}, \mathbf{W}^{(\mathbf{J})})_\mathcal{M}$, where $\mathbf{W} = \mathbf{W}_1 \cup \mathbf{W}_2$.*

*Proof.* Suppose otherwise $(F \not\perp\!\!\!\perp X^{(\mathbf{I})} | \mathbf{I}^{(\mathbf{I})}, \mathbf{W}^{(\mathbf{I})})_{\mathcal{M}}$, meaning that conditioning on $\mathbf{W}^{(\mathbf{I})}$ activates extra paths from $F$ to $X^{(\mathbf{I})}$ in $\mathcal{M}$. Obviously, $\mathbf{I}^{(\mathbf{I})}$ can neither be on the paths nor activate a collider on the paths. Consider a m-connecting path $p^{(\mathbf{I})}$ from $F$ to $X^{(\mathbf{I})}$ given $\mathbf{W}^{(\mathbf{I})}$ in $\mathcal{M}$. Then there is a collider $C_1^{(\mathbf{I})}$ on $p^{(\mathbf{I})}$ which is activated by some $W_2^{(\mathbf{I})} \in \mathbf{W}_2^{(\mathbf{I})}$. Let the subgraph of $\mathcal{M}$ induced by $V^{(\mathbf{J})} \cup F$ be $G_J$. $F$ and $X^{(\mathbf{J})}$ are blocked by $\mathbf{W}_2$ meaning that the corresponding path $p^{(\mathbf{J})}$ in $G_J$ is either blocked by inactivated colliders or does not exist due to removed edges into $\mathbf{J}$. In both cases, $p^{(\mathbf{J})}$ is blocked by $\emptyset$. However, due to the minimality of $\mathbf{W}_2$, it has to block some path other than $p^{(\mathbf{J})}$. Consider $W_x \in \mathbf{W}_2$ which is closest to $X$ and is a descendant of a collider $C_1^{(\mathbf{J})}$ on $p^{(\mathbf{J})}$. Denote the path created by concatenating direct paths from $C_1^{(\mathbf{J})}$ to $W_x^{\mathbf{J})}$ and $C_1^{(\mathbf{J})}$ to $X^{(\mathbf{J})}$ as $p_x^{(\mathbf{J})}$. Suppose there is a path $p_1^{(\mathbf{J})}$ from $F$ to $X^{(\mathbf{J})}$ that is blocked by $W_x^{(\mathbf{J})}$. Since $p_1^{(\mathbf{J})}$ is blocked by $W_x^{(\mathbf{J})}$, $W_x^{(\mathbf{J})}$ cannot be a collider on $p_1^{(\mathbf{J})}$. If there is an edge into $W_x^{(\mathbf{J})}$ from $F$'s side, then the subpath of $p_x^{(\mathbf{J})}$ will be m-connecting. The subpath from $F$ to $W_x^{(\mathbf{J})}$ has to be blocked by inactivated colliders or conditioning on a non-collider. In either case, $W_x$ is not necessary for blocking $p_1^{(\mathbf{J})}$. Conversely, there has to be an outgoing edge from $W_x^{(J)}$ towards $F$. Consequently, on $p_1^{(\mathbf{J})}$, $F$ has an outgoing edge towards $W_x^{(\mathbf{J})}$, while $W_x^{(\mathbf{J})}$ has an outgoing edge towards $F$. Thus there has to be a collider in between $F$ and $W_x^{(\mathbf{J})}$. Consider such a collider $C_2^{(\mathbf{J})} \in \mathbf{W}_2^{(\mathbf{J})}$ that is closest to $F$. Nevertheless, according to the minimality of $\mathbf{W}_2$, $C_2^{(\mathbf{J})}$ has to block another path. If we repeat this process for $n = |\mathbf{V}|$ times, we show that $|\mathbf{W}_2| > n$, which is impossible. Thus the supposition that $(F \not\perp\!\!\!\perp X^{(\mathbf{I})} | \mathbf{I}^{(\mathbf{I})}, \mathbf{W}^{(\mathbf{I})})_{\mathcal{M}}$ does not hold. Similarly, we can show that $(F \perp\!\!\!\perp X^{(\mathbf{J})} | \mathbf{J}^{(\mathbf{J})}, \mathbf{W}^{(\mathbf{J})})_{\mathcal{M}}$ which concludes the proof. $\square$

Therefore, according to Lemma B.6, the structural difference implies that we can find some $\mathbf{W} \in \mathbf{W} \setminus \{X\}$ such that $(F \not\perp\!\!\!\perp X^{(\mathbf{I})} | \mathbf{I}^{(\mathbf{I})}, \mathbf{W}^{(\mathbf{I})})_{\mathcal{M}_1} \vee (F \not\perp\!\!\!\perp X^{(\mathbf{J})} | \mathbf{J}^{(\mathbf{J})}, \mathbf{W}^{(\mathbf{J})})_{\mathcal{M}_1}$ while $(F \perp\!\!\!\perp X^{(\mathbf{I})} | \mathbf{I}^{(\mathbf{I})}, \mathbf{W}^{(\mathbf{I})})_{\mathcal{M}_2} \wedge (F \perp\!\!\!\perp X^{(\mathbf{J})} | \mathbf{J}^{(\mathbf{J})}, \mathbf{W}^{(\mathbf{J})})_{\mathcal{M}_2}$, which is a pair of different testable m-separation statements. To conclude, $\mathcal{M}_1, \mathcal{M}_2$ have the same skeleton.

Next, we show that $\mathcal{M}_1$ and $\mathcal{M}_2$ have the same unshielded colliders. We start by showing that they have the same unshielded colliders in the vertex induced subgraph on $\mathbf{V}^{(\mathbf{I})}$ and $\mathbf{V}^{(\mathbf{J})}$. Suppose otherwise, $\langle X^{(\mathbf{I})}, Y^{(\mathbf{I})}, Z^{(\mathbf{I})} \rangle$ is an unshielded collider in $\mathcal{M}_1$ but not in $\mathcal{M}_2$. Since $Y^{(\mathbf{I})}$ is not a collider in $\mathcal{M}_2$, it has to be conditioned on to make $X^{(\mathbf{I})}$ and $Z^{(\mathbf{I})}$ m-separable. Then we have $(X^{(\mathbf{I})} \perp\!\!\!\perp Z^{(\mathbf{I})} | \mathbf{I}^{(\mathbf{I})}, \mathbf{W}^{(\mathbf{I})})_{\mathcal{M}_1} \wedge (X^{(\mathbf{I})} \not\perp\!\!\!\perp Z^{(\mathbf{I})} | \mathbf{I}^{(\mathbf{I})}, \mathbf{W}^{(\mathbf{I})}, Y^{(\mathbf{I})})_{\mathcal{M}_1}$ and $(X^{(\mathbf{I})} \perp\!\!\!\perp Z^{(\mathbf{I})} | \mathbf{I}^{(\mathbf{I})}, \mathbf{W}^{(\mathbf{I})})_{\mathcal{M}_2} \wedge (X^{(\mathbf{I})} \perp\!\!\!\perp Z^{(\mathbf{I})} | \mathbf{I}^{(\mathbf{I})}, \mathbf{W}^{(\mathbf{I})}, Y^{(\mathbf{I})})_{\mathcal{M}_2}$ for some $\mathbf{W} \subseteq \mathbf{V} \setminus \{Y\}$, which contains a pair of different testable statements.

Then we need to show that unshielded colliders which include $F$ nodes are also the same in both graphs. Due to the construction, $F$ can only have outgoing edges, thus it can only be an end node in the collider. Suppose $\langle F, Y^{(\mathbf{I})}, Z^{(\mathbf{I})} \rangle$ is an unshielded collider in $\mathcal{M}_1$ but not in $\mathcal{M}_2$. More specifically, $F \rightarrow Y^{(\mathbf{I})} \leftrightarrow\!\ast Z^{(\mathbf{I})}$ in $\mathcal{M}_1$ and $F \rightarrow Y^{(\mathbf{I})} \rightarrow Z^{(\mathbf{I})}$ in $\mathcal{M}_2$. There are 2 cases: $Y \in \mathbf{K}$ or $Y \notin \mathbf{K}$.

First, consider $Y \in \mathbf{K}$. $Y$ has to be in $\mathbf{J} \setminus \mathbf{I}$, then in the induced subgraph of $\mathbf{V}^{(\mathbf{J})}$, we cannot have $Y^{(\mathbf{J})} \leftrightarrow\!\ast Z^{(\mathbf{J})}$ in $\mathcal{M}_1$ or $\mathcal{M}_2$. Thus we can find some $\mathbf{W} \subseteq \mathbf{V} \setminus \{Y\}$, such that $(F \not\perp\!\!\!\perp Z^{(\mathbf{I})} | \mathbf{I}^{(\mathbf{I})}, \mathbf{W}^{(\mathbf{I})}, Y^{(\mathbf{I})})_{\mathcal{M}_1} \vee (F \not\perp\!\!\!\perp Z^{(\mathbf{J})} | \mathbf{J}^{(\mathbf{J})}, \mathbf{W}^{(\mathbf{J})}, Y^{(\mathbf{J})})_{\mathcal{M}_1}$, while $(F \perp\!\!\!\perp Z^{(\mathbf{I})} | \mathbf{I}^{(\mathbf{I})}, \mathbf{W}^{(\mathbf{I})}, Y^{(\mathbf{I})})_{\mathcal{M}_2} \wedge (F \not\perp\!\!\!\perp Z^{(\mathbf{J})} | \mathbf{J}^{(\mathbf{J})}, \mathbf{W}^{(\mathbf{J})}, Y^{(\mathbf{J})})_{\mathcal{M}_2}$, which is a pair of different testable m-separation statements.

Second, consider the case that $Y \notin \mathbf{K}$, but is adjacent to $F$ in both $\mathcal{M}_1$ and $\mathcal{M}_2$. Then there is an inducing path from $F$ to $Y^{(\mathbf{I})}$ or $Y^{(\mathbf{J})}$ in the augmented pair graphs. According to our construction of twin augmented MAGs, $F$ is adjacent to both $Y^{(\mathbf{I})}$ and $Y^{(\mathbf{J})}$ in both $\mathcal{M}_1$ and $\mathcal{M}_2$. Since $F$ is not adjacent to $Z^{(\mathbf{I})}, Z^{(\mathbf{J})}$ in $\mathcal{M}_2$, we know according to Lemma B.6 that there exists $\mathbf{W} \subseteq \mathbf{V}$ such that $(F \perp\!\!\!\perp Z^{(\mathbf{I})} | \mathbf{I}^{(\mathbf{I})}, \mathbf{W}^{(\mathbf{I})})_{\mathcal{M}_2} \wedge (F \perp\!\!\!\perp Z^{(\mathbf{J})} | \mathbf{J}^{(\mathbf{J})}, \mathbf{W}^{(\mathbf{J})})_{\mathcal{M}_2}$. $\mathbf{W}$ has to contain $Y$, otherwise $F$ is m-connecting to $Z^{(\mathbf{I})}$ or $Z^{(\mathbf{J})}$ in $\mathcal{M}_2$ through $Y^{(\mathbf{I})}$ or $Y^{(\mathbf{J})}$. We need to further show that there cannot be $Y^{(\mathbf{J})} \leftrightarrow\!\ast Z^{(\mathbf{J})}$ in $\mathcal{M}_2$. Suppose otherwise, if $Y^{(\mathbf{J})} \leftarrow Z^{(\mathbf{J})}$ in $\mathcal{M}_2$, then there is a cycle in $\mathcal{D}_2$. Else if $Y^{(\mathbf{J})} \leftrightarrow Z^{(\mathbf{J})}$ in $\mathcal{M}_2$, then there is a latent common ancestor of $Y, Z$ in $\mathcal{D}_2$. This will create an inducing path $\langle F, Y^{(\mathbf{I})}, Z^{(\mathbf{I})} \rangle$ in $\mathcal{M}_2$ which contradicts the condition that $F$ and $Z^{(\mathbf{I})}$ are non-adjacent. In $\mathcal{M}_1$, since there is $F \rightarrow Y \leftrightarrow\!\ast Z$, we have

$(F \not\perp\!\!\!\perp Z^{(\mathbf{I})} | \mathbf{I}^{(\mathbf{I})}, \mathbf{W}^{(\mathbf{I})})_{\mathcal{M}_1} \wedge (F \not\perp\!\!\!\perp X^{(\mathbf{J})} | \mathbf{J}^{(\mathbf{J})}, \mathbf{W}^{(\mathbf{J})})_{\mathcal{M}_1}$, which contains a pair of different testable statements in $\mathcal{T}$. Therefore, $\mathcal{M}_1$ and $\mathcal{M}_2$ have the same unshielded colliders.

Suppose $\langle U, W_1, W_2, ..., W_k, Y, Z \rangle$ is a discriminating path in both $\mathcal{M}_1$ and $\mathcal{M}_2$, $Y$ is a collider in $\mathcal{M}_1$ but a non-collider in $\mathcal{M}_2$. By our construction, $F$ can only be $U$ or $Y$ in the path. There are 3 cases.

First, if $F$ is not in the path. Then all the nodes in the path are in $\mathbf{V}^{(\mathbf{I})}$ or $\mathbf{V}^{(\mathbf{J})}$. Suppose the path is in $\mathbf{V}^{(\mathbf{I})}$. Then in $\mathcal{M}_2$, there exists some $\mathbf{W} \subseteq \mathbf{V}$, such that $(U^{(\mathbf{I})} \perp\!\!\!\perp Z^{(\mathbf{I})} | \mathbf{I}^{(\mathbf{I})}, F, \mathbf{W}^{(\mathbf{I})})_{\mathcal{M}_2}, W_i, Y \in \mathbf{W}, i \in [k]$. While in $\mathcal{M}_1$, we have $(U^{(\mathbf{I})} \not\perp\!\!\!\perp Z^{(\mathbf{I})} | \mathbf{I}^{(\mathbf{I})}, F, \mathbf{W}^{(\mathbf{I})})_{\mathcal{M}_1}$, since conditioning on $Y^{(\mathbf{I})}$ would activate the path. Thus there is a pair of testable m-separation statements.

Next, if $F$ is $Y$ in the path, this case is not valid, since $F$ can only have outgoing edges. Therefore, if $F$ is $Y$ in the path, it should have the same collider status in $\mathcal{M}_1$ and $\mathcal{M}_2$.

Lastly, $F$ is $U$ in the path. All of the other nodes in the path have to be all in either $\mathbf{V}^{(\mathbf{I})}$ or $\mathbf{V}^{(\mathbf{J})}$. Suppose that all other nodes are in $\mathbf{V}^{(\mathbf{I})}$. Since $F$ is non-adjacent to $Z^{(\mathbf{I})}$ in $\mathcal{M}_1$ and $\mathcal{M}_2$, we know that $F$ is also non-adjacent to $Z^{(\mathbf{J})}$ according to Lemma B.6. Thus in $\mathcal{M}_2$, there exists $\mathbf{W} \subseteq \mathbf{V}$, such that $(F \perp\!\!\!\perp Z^{(\mathbf{I})} | \mathbf{I}^{(\mathbf{I})}, \mathbf{W}^{(\mathbf{I})})_{\mathcal{M}_2} \wedge (F \perp\!\!\!\perp Z^{(\mathbf{J})} | \mathbf{J}^{(\mathbf{J})}, \mathbf{W}^{(\mathbf{J})})_{\mathcal{M}_2}, W_i, Y \in \mathbf{W}, i \in [k]$. $W_i, Y$ are in $\mathbf{W}$ because otherwise $F$ is m-connecting with $Z^{(\mathbf{I})}$ through the path. However, in $\mathcal{M}_1$ we have $(F \perp\!\!\!\perp Z^{(\mathbf{I})} | \mathbf{I}^{(\mathbf{I})}, \mathbf{W}^{(\mathbf{I})})_{\mathcal{M}_1} \wedge (F \perp\!\!\!\perp Z^{(\mathbf{J})} | \mathbf{J}^{(\mathbf{J})}, \mathbf{W}^{(\mathbf{J})})_{\mathcal{M}_1}$ because conditioning on $W_i^{(\mathbf{I})}, Y^{(\mathbf{I})}$ will activate every collider in the path from $F$ to $Z^{(\mathbf{I})}$. Therefore, there is a pair of different testable statements in $\mathcal{T}$. To conclude, for each discriminating path in $\mathcal{M}_1$ and $\mathcal{M}_2$, $F$ is a collider in $\mathcal{M}_1$ if and only if it is a collider in $\mathcal{M}_2$. Up to now, we showed that if $\mathcal{M}_1$ and $\mathcal{M}_2$ do not satisfy the three graphical conditions in the theorem, then there exists a testable m-separation statement that holds in one graph but not the other.

When there exists a pair of intervention targets $\mathbf{I}, \mathbf{J} \in \mathcal{I}$ such that $\mathcal{M}_1 =$ $\mathrm{Twin}_{(\mathbf{I},\mathbf{J})}(\mathcal{D}_1)$ and $\mathcal{M}_2 = \mathrm{Twin}_{(\mathbf{I},\mathbf{J})}(\mathcal{D}_2)$ do not satisfy either of the three conditions mentioned in the theorem statement, this implies that $\mathcal{D}_1$ and $\mathcal{D}_2$ are not $\mathcal{I}$-Markov equivalent. This is because there is a m-separation statement that appears as a condition in the definition of $\mathcal{I}$-Markov equivalence that is different in the two graphs $\mathcal{M}_1$ and $\mathcal{M}_2$. There is a m-separating path in $\mathcal{M}_1$ that is m-connecting in $\mathcal{M}_2$. We now show that $\mathcal{P}_{\mathcal{I}}(\mathcal{D}_2, \mathbf{V})$ contains tuples of distributions that are not in $\mathcal{P}_{\mathcal{I}}(\mathcal{D}_1, \mathbf{V})$. We leverage a key result from Meek, demonstrating that the set of unfaithful distributions has Lebesgue measure zero. Building on this, we construct a jointly Gaussian structural causal model that incorporates latent variables.

**Lemma B.7.** *Meek [2013b] Consider a causal DAG $D = (\mathbf{V}, \mathbf{E})$, where $(A \not\perp\!\!\!\perp B \mid C)_D$. Let $D_s = (\mathbf{V}_s, \mathbf{E}_s)$ be the subgraph that contains all the nodes in the m-connecting path that induces $(A \not\perp\!\!\!\perp B \mid C)_D$. Then any distribution $P$ over $\mathbf{V}_s$ where every adjacent pair of variables is dependent satisfies $(A \not\perp\!\!\!\perp B \mid C)_p$.*

The proof of Lemma B.7 uses weak transitivity and an inductive argument and can be found in Meek [2013b]. Suppose that $\mathbf{X}, \mathbf{Y}, \mathbf{Z} \subseteq \mathbf{V}$ such that $(\mathbf{X} \perp\!\!\!\perp \mathbf{Y} \mid \mathbf{Z}, F)_{\mathrm{Aug}_{(\mathbf{I},\mathbf{J})}(\mathcal{D}_1)}$ and $(\mathbf{X} \not\perp\!\!\!\perp \mathbf{Y} \mid \mathbf{Z}, F)_{\mathrm{Aug}_{(\mathbf{I},\mathbf{J})}(\mathcal{D}_2)}$. Suppose that both $\mathbf{X}, \mathbf{Y}$ are observed variables. In this case, any tuple of interventional distribution obtained from an observational distribution that is faithful to the causal graph with latent variables constitutes a valid example. Suppose $\mathbf{X} = F$ for some and $\mathbf{Y} \in \mathbf{V}$. Therefore, an $F$-node is m-connected to an observed node in $\mathrm{Aug}_{(\mathbf{I},\mathbf{J})}(\mathcal{D}_2)$ but not in $\mathrm{Aug}_{(\mathbf{I},\mathbf{J})}(\mathcal{D}_1)$.

Consider the causal graph $\mathcal{D}_2 = (\mathbf{V} \cup \mathbf{L}, \mathbf{E})$ with latent variables. Focus on the subgraph of $\mathcal{D}_2$ that includes all variables contributing to the m-connecting path of $(\mathbf{X} \not\perp\!\!\!\perp \mathbf{Y} \mid \mathbf{Z}, F)_{\mathrm{Aug}_{(\mathbf{I},\mathbf{J})}(\mathcal{D}_2)}$. An example can be found in Meek [2013b]. Let us call this subgraph $\mathcal{D}_{\mathrm{path}} = (V_{\mathrm{path}}, E_{\mathrm{path}})$. Consider a jointly Gaussian distribution on $V_{\mathrm{path}}$ that is faithful to $\mathcal{D}_{\mathrm{path}}$. Such a distribution exists by construction in Meek (Theorem 7, [Meek, 2013b]). Denote this distribution by $P_{\mathrm{path}}$. We will focus on $P_{\mathrm{path}}$ and later expand it by adding the remaining variables in $\mathcal{D}_{\mathrm{suff}}$ as jointly independent and independent of the variables in $D_{\mathrm{path}}$. Now, consider two intervention targets $\mathbf{I}$ and $\mathbf{J}$ on the CBN $(\mathcal{D}_{\mathrm{path}}, P_{\mathrm{path}})$, where $\mathbf{I}\Delta\mathbf{J} = K$. This implies that the distributions $P_{\mathbf{I}}$ and $P_{\mathbf{J}}$ account for the graphical separation of $F_I$. For this proof, we treat $F_I$ as a regime variable indicating when we switch between $P_{\mathbf{I}}$ and $P_{\mathbf{J}}$. This treatment is valid because we add only this single $F$ node, without introducing others. Define the distribution $P^*$ as follows:

$$P^*(\cdot \mid F_I = 0) = P_{\mathbf{I}}(\cdot), \quad P^*(\cdot \mid F_I = 1) = P_{\mathbf{J}}(\cdot),$$

and let $F_I$ follow a uniform distribution. We need to show that the invariances implied by the graph separation in the generalized causal calculus rules fail for $P_\mathbf{I}$ and $P_\mathbf{J}$. This is equivalent to demonstrating that $F_I$ is dependent on $\mathbf{Y}$ given $\mathbf{Z}$ in the distribution $P^*$. To construct the interventional distributions, we use a SCM that implies the given CBN. Let $\mathbf{x}$ represent the vector of all variables in the graph, including latent ones. Consider the SCM given as follows:

$$\mathbf{x} = \mathbf{Ax} + \mathbf{e},$$

where $\mathbf{A}$ is a lower triangular matrix describing the graph structure and parent-child relationships in $\mathcal{D}_{\text{path}}$, and $\mathbf{e}$ is the vector of exogenous Gaussian noise terms, $|\mathbf{x}| = n$.

Let $P_\mathbf{I}$ represent the distribution obtained by introducing the noise vector $\mathbf{e}_I$ into the system. The vector $\mathbf{e}_I$ is non-zero only in the rows $i$ where $x_i \in \mathbf{I}$. Furthermore, the matrix $\mathbf{A}$ is modified to a new matrix $\mathbf{A}_1$, which is identical to $\mathbf{A}$ except that all entries in the rows corresponding to $i$ ($\forall x_i \in \mathbf{I}$) are set to zero. This modification effectively removes the influence of the parents of the variables in the intervention set $\mathbf{I}$, as required by the definition of a hard intervention. Consequently, $P_\mathbf{I}$ qualifies as a valid hard interventional distribution. Similarly, let $\mathbf{e}_J$ denote the noise vector introduced for an intervention on $\mathbf{J}$. The matrix $\mathbf{A}$ is similarly replaced with $\mathbf{A}_2$, where all entries in the rows corresponding to $j$ ($\forall x_j \in \mathbf{J}$) are set to zero, achieving the same effect for the variables in $\mathbf{J}$. We show that in the combined distribution $p*$ using these $P_\mathbf{I}, P_\mathbf{J}$ every adjacent variable is dependent. Clearly, when $\mathbf{e}_I$ and $\mathbf{e}_J$ are different, $F$ variable is dependent with the variables in $\mathbf{K} := \mathbf{I}\Delta\mathbf{J}$, since $P^*(\mathbf{K} \mid F = 0) \neq P^*(\mathbf{K} \mid F = 1)$, which implies ($\mathbf{K} \not\perp\!\!\!\perp F \mid \emptyset)_{P^*}$. Therefore, we focus on establishing that every pair of variables that are adjacent are correlated except for the $F$ variable. The correlation of the variables in $\mathcal{D}_{\text{path}}$ is calculated as follows:

$$\mathbf{x} = \mathbf{A_1x} + \mathbf{e} + \mathbf{e}_I \Rightarrow (\mathbf{I}_n - \mathbf{A_1})\mathbf{x} = \mathbf{e} + \mathbf{e}_I \Rightarrow \mathbf{x} = (\mathbf{I}_n - \mathbf{A_1})^{-1}(\mathbf{e} + \mathbf{e}_I)$$

$$\mathbf{x} = \mathbf{A_2x} + \mathbf{e} + \mathbf{e}_J \Rightarrow (\mathbf{I}_n - \mathbf{A_2})\mathbf{x} = \mathbf{e} + \mathbf{e}_J \Rightarrow \mathbf{x} = (\mathbf{I}_n - \mathbf{A_2})^{-1}(\mathbf{e} + \mathbf{e}_J)$$

Where $\mathbf{I}_n$ is the identity matrix of size $n$ by $n$. Let $\mathbf{e}_1 = \mathbf{e} + \mathbf{e}_I$ and $\mathbf{e}_2 = \mathbf{e} + \mathbf{e}_J$. The correlation matrix between the observed variables, with respect to the distribution $P^*$ after marginalizing out the binary regime variable, is computed as follows:

$$E\left[\mathbf{xx}^T\right] = 0.5(\mathbf{I}_n - \mathbf{A_1})^{-1}E\left[\mathbf{e}_1\mathbf{e}_1^T\right](\mathbf{I}_n - \mathbf{A_1})^{-1^T} + 0.5(\mathbf{I}_n - \mathbf{A_2})^{-1}E\left[\mathbf{e}_2\mathbf{e}_2^T\right](\mathbf{I}_n - \mathbf{A_2})^{-1^T}$$

Let $\mathbf{D}_1 = E\left[\mathbf{e}_1\mathbf{e}_1^T\right]$ and $\mathbf{D}_2 = E\left[\mathbf{e}_2\mathbf{e}_2^T\right]$ represent the diagonal covariance matrices for the noise introduced by the hard interventions. Additionally, assume that all the noise variables, including $\mathbf{e}$, $\mathbf{e}_I$, and $\mathbf{e}_J$, follow zero-mean Gaussian distributions. Consider two adjacent variables, $x_i$ and $x_j$, within $\mathcal{D}_{\text{path}}$. We observe that the matrices $\mathbf{I}_n - \mathbf{A}_1$ and $\mathbf{I}_n - \mathbf{A}_2$ are full-rank, as $\mathbf{A}$ is a strictly lower triangular matrix, and the same holds for $\mathbf{A}_1$ and $\mathbf{A}_2$. As a result, the matrix inverses in these equations exist and are unique.

We now treat $\mathbf{D}_1$ and $\mathbf{D}_2$ as variables in this system. When performing a hard intervention, we can freely select the variance of each added noise term. Our objective is to demonstrate that there always exist hard interventions, represented by $\mathbf{D}_1$ and $\mathbf{D}_2$, such that $x_i$ and $x_j$ become dependent. Since both $x_i$ and $x_j$ are jointly Gaussian, they are dependent if and only if they are correlated. Therefore, we need to show that $E\left[x_ix_j\right] \neq 0$ for any adjacent pair $x_i, x_j$. If we set $\mathbf{D}_1 = \mathbf{D}_2 = \mathbf{0}$, we return to the observational system. By the assumption that the original distribution is faithful to the graph $\mathcal{D}_{\text{path}}$, any adjacent variables are dependent. This implies that the corresponding system of linear equations is not trivially zero. Hence, by randomly choosing the variances of the noise terms, we can ensure that, with probability 1, any adjacent pair of variables will be dependent (by applying a union bound).

Thus, we have shown that, in the graph $\mathcal{D}_{\text{path}}$ along with the $F$ variable, every pair of adjacent variables is dependent. Next, we can extend this distribution to include the variables outside $\mathcal{D}_{\text{path}}$. To do this, we select the remaining variables to be jointly independent and independent from those in $\mathcal{D}_{\text{path}}$. The interventional distributions can then be constructed by applying a similar hard intervention, where extra noise terms are added to the variables being intervened upon and replacing the matrix $\mathbf{A}$. The resulting set of interventional distributions will belong to $\mathcal{P}_\mathcal{I}(\mathcal{D}_2, \mathbf{V})$, but not to $\mathcal{P}_\mathcal{I}(\mathcal{D}_1, \mathbf{V})$. This is because m-separation should imply invariance across the interventional distributions, but we have constructed them in such a way that this condition does not hold. This concludes the proof. $\square$

## B.5 Proof for Proposition 5.2

*Proof.* We have shown in Theorem 4.7 that two causal graphs are $\mathcal{I}$-Markov equivalent if and only if their twin augmented MAGs satisfy the 3 conditions. Here we just need to show that the twin augmented MAGs follow the 3 conditions if and only if the $\mathcal{I}$-augmented MAGs follow the 3 conditions.

**(If:)** Notice that each twin augmented MAG $\mathrm{Twin}_{\mathbf{I},\mathbf{J}}(\mathcal{D})$ can be constructed by taking the graph union of $\mathrm{Aug}_{\mathbf{I}}(\mathcal{D},\mathcal{I})$ and $\mathrm{Aug}_{\mathbf{J}}(\mathcal{D},\mathcal{I})$ and removing irrelevant $F$ nodes. Therefore, if $\mathrm{Aug}_{\mathbf{I}}(\mathcal{D}_1,\mathcal{I})$ and $\mathrm{Aug}_{\mathbf{J}}(\mathcal{D}_2,\mathcal{I})$ have the same skeleton for any $\mathbf{I} \in \mathcal{I}$, $\mathrm{Twin}_{\mathbf{I},\mathbf{J}}(\mathcal{D}_1)$ and $\mathrm{Twin}_{\mathbf{I},\mathbf{J}}(\mathcal{D}_2)$ will also have the same skeleton. Furthermore, $\mathrm{Twin}_{\mathbf{I},\mathbf{J}}(\mathcal{D})$ and $\mathrm{Aug}_{\mathbf{I}}(\mathcal{D},\mathcal{I})$ have the same unshielded colliders within $\mathbf{V}^{(\mathbf{I})} \cup \{F^{(\mathbf{I},\mathbf{J})}\}$ because they have the same subgraph on $\mathbf{V}^{(\mathbf{I})} \cup \{F^{(\mathbf{I},\mathbf{J})}\}$. Since twin augmented MAGs do not have other $F$ nodes, $\mathrm{Twin}_{\mathbf{I},\mathbf{J}}(\mathcal{D}_1)$ and $\mathrm{Twin}_{\mathbf{I},\mathbf{J}}(\mathcal{D}_2)$ have the same unshielded colliders. Similarly, any discriminating path in $\mathrm{Aug}_{\mathbf{I}}(\mathcal{D},\mathcal{I})$ will also be preserved in $\mathrm{Twin}_{\mathbf{I},\mathbf{J}}(\mathcal{D})$ while there cannot be discriminating paths that pass through the $F$ node in $\mathrm{Twin}_{\mathbf{I},\mathbf{J}}(\mathcal{D})$. Therefore, $\mathrm{Twin}_{\mathbf{I},\mathbf{J}}(\mathcal{D}_1)$ and $\mathrm{Twin}_{\mathbf{I},\mathbf{J}}(\mathcal{D}_2)$ have the same collider status on discriminating paths.

**(Only if:)** Now we show that if $\mathrm{Twin}_{\mathbf{I},\mathbf{J}}(\mathcal{D}_1)$ and $\mathrm{Twin}_{\mathbf{I},\mathbf{J}}(\mathcal{D}_2)$ satisfy the 3 conditions for any $\mathbf{I},\mathbf{J} \in \mathcal{I}$, then $\mathrm{Aug}_{\mathbf{I}}(\mathcal{D}_1,\mathcal{I})$ and $\mathrm{Aug}_{\mathbf{I}}(\mathcal{D}_2,\mathcal{I})$ also satisfy the 3 conditions for any $\mathbf{I} \in \mathcal{I}$. Since $\mathrm{Aug}_{\mathbf{I}}(\mathcal{D}_1,\mathcal{I})$ is the graph union of $\mathrm{Twin}_{\mathbf{I},\mathbf{J}}(\mathcal{D}_1)$ on $\mathbf{V}^{(\mathbf{I})} \cup \{F^{(\mathbf{I},\mathbf{J})}\}$, all the adjacencies are kept and the same for $\mathrm{Aug}_{\mathbf{I}}(\mathcal{D}_2,\mathcal{I})$. Thus, they have the same adjacencies. For unshielded triples, if both ends are $F$ nodes, then they have to be unshielded colliders in both $\mathrm{Aug}_{\mathbf{I}}(\mathcal{D}_1,\mathcal{I})$ and $\mathrm{Aug}_{\mathbf{I}}(\mathcal{D}_2,\mathcal{I})$ given $F$ nodes have only outgoing edges by construction. If at most one endpoint of the unshielded triple is an $F$ node, then the same structure can be retrieved from the relevant twin augmented MAG. Therefore, $\mathrm{Aug}_{\mathbf{I}}(\mathcal{D}_1,\mathcal{I})$ and $\mathrm{Aug}_{\mathbf{I}}(\mathcal{D}_2,\mathcal{I})$ have the same unshielded colliders. Finally, we need to show that if $p = \langle U, W_1, W_2, ..., W_k, Y, Z \rangle$ is a discriminating path in both $\mathrm{Aug}_{\mathbf{I}}(\mathcal{D}_1,\mathcal{I})$ and $\mathrm{Aug}_{\mathbf{I}}(\mathcal{D}_2,\mathcal{I})$, then $p$ has the same collider status. If $p$ is in $\mathbf{V}^{(\mathbf{I})}$, then $p$ is also in $\mathrm{Twin}_{\mathbf{I},\mathbf{J}}(\mathcal{D}_1)$ and $\mathrm{Twin}_{\mathbf{I},\mathbf{J}}(\mathcal{D}_2)$ for any $\mathbf{J} \in \mathcal{I}$ and thus it shows the same collider status in both $\mathcal{I}$-augmented MAGs. If $F^{(\mathbf{I},\mathbf{J})}$ is an endpoint of $p$, then the path is also in $Twin_{\mathbf{I},\mathbf{J}}(\mathcal{D}_1)$ and $\mathrm{Twin}_{\mathbf{I},\mathbf{J}}(\mathcal{D}_2)$ with the same collider status. By the construction of $\mathcal{I}$-augmented MAGs, an $F$ node can only be $Y$ in $p$ if it is not the starting node. Since $F$ nodes only have outgoing edges, $p$ will have the same collider status in both $\mathrm{Aug}_{\mathbf{I}}(\mathcal{D}_1,\mathcal{I})$ and $\mathrm{Aug}_{\mathbf{I}}(\mathcal{D}_2,\mathcal{I})$, $\qquad\square$

## B.6 Proof for Theorem 6.3

The algorithm is learning the causal graph through finding the separating sets between each pair of nodes using the distributional invariance tests. The invariance tests are tied to the m-separation statements in the causal graph according to the h-faithfulness assumption, and the properties in Definition 4.1 are mapped to the m-separation statements in the twin augmented MAGs by Proposition 4.4. We show that twin augmented MAGs are combined to construct $\mathcal{I}$-augmented MAGs which preserve the m-separation statements in Proposition 5.2. Ideally, our algorithm would learn a structure that is close to the $\mathcal{I}$-augmented MAGs. Therefore, to show the soundness of Algorithm 1, we need to first define the $\mathcal{I}$-essential graph as follows:

**Definition B.8** ($\mathcal{I}$-essential graph). Given a causal graph $\mathcal{D} = (\mathbf{V} \cup \mathbf{L}, \mathbf{E})$ and a set of intervention targets $\mathcal{I} \subseteq 2^V$, the $\mathcal{I}$-essential graph of $\mathcal{D}$ related to $\mathcal{I}$, denoted as $\mathcal{E}_I(\mathcal{D},\mathcal{I})$ is the union graph of the $\mathcal{I}$-augmented MAGs of $\mathcal{D}'$ for all $\mathcal{D}'$ that are ADMGs $\mathcal{I}$-Markov equivalent to $\mathcal{D}$.

**Definition B.9** (Union Graph of $k$ ADMGs). Let $\mathcal{G}_1, \mathcal{G}_2, \ldots, \mathcal{G}_k$ be $k$ ADMGs, where each graph $\mathcal{G}_i = (\mathbf{V}_i, \mathbf{E}_i)$

The **union graph** of $k$ ADMGs, denoted as $\mathcal{G}_\cup = (\mathbf{V}_\cup, \mathbf{E}_\cup)$, where $\mathbf{V}_\cup = \bigcup_{i=1}^k \mathbf{V}_i$, for each pair of vertices $(X,Y)$, the edge set of $\mathcal{G}_\cup$ is determined by:

$$\mathcal{E}_\cup = \begin{cases} X \to Y, & \text{if } (X \to Y) \text{ appears in all } \mathcal{G}_i, X, Y \in \mathbf{V}_i. \\ X \leftrightarrow Y, & \text{if } (X \leftrightarrow Y) \text{ appears in all } \mathcal{G}_i, X, Y \in \mathbf{V}_i. \\ Xo\to Y, & \text{if } (X \to Y) \text{ appears in some } \mathcal{G}_i, (X \leftrightarrow Y) \text{ appears in some } \mathcal{G}_j, X, Y \in \mathbf{V}_i, \mathbf{V}_j. \\ Xo{-}oY, & \text{if } (X \leftarrow Y) \text{ appears in some } \mathcal{G}_i, (X \to Y) \text{appears in some } \mathcal{G}_j, X, Y \in \mathbf{V}_i, \mathbf{V}_j. \end{cases}$$

The $\mathcal{I}$-essential graphs denote the structure that is ultimately learnable by any causal discovery algorithm. Based on this definition, to demonstrate the soundness of our algorithm, it suffices to address the following two questions:

(a) Do all the $\mathcal{I}$-augmented graphs returned by Algorithm 1 have the same adjacencies as the $\mathcal{I}$-essential graphs?

(b) Are the orientation rules sound, i.e. any arrowhead/arrowtail learned by the algorithm is also present in the $\mathcal{I}$-essential graph?

We first address (a). Consider an $\mathcal{I}$-augmented graph $\mathcal{G}_I$, the edges either contain a $F$ node or not. There are no edges between any two $F$ nodes by our construction of $\mathcal{G}_\mathbf{I}$ and $\mathrm{Aug}_\mathbf{I}(\mathcal{D}, \mathcal{I})$. Consider the edge $(F^{(\mathbf{I}, \mathbf{J})}, Y^{(\mathbf{I})})$ in $\mathcal{G}_\mathbf{I}$. It is recovered because there does not exist $\mathbf{W} \subseteq \mathbf{V}$, such that $P_\mathbf{I}(y|\mathbf{w}) = P_\mathbf{J}(y|\mathbf{w})$ under h-faithfulness. According to Definition 4.1 and Proposition 4.4, it implies that $F^{(\mathbf{I}, \mathbf{J})}$ and $Y^{(\mathbf{I})}, Y^{(\mathbf{J})}$ are also adjacent in $Twin_{(\mathbf{I}, \mathbf{J})}(\mathcal{D}')$ for any $\mathcal{D}'$ that is $\mathcal{I}$-Markov equivalent to $\mathcal{D}$. Thus $F^{(\mathbf{I}, \mathbf{J})}$ and $Y^{(\mathbf{I})}$ are also adjacent in $\mathrm{Aug}_\mathbf{I}(\mathcal{D}', \mathcal{I})$. Consequently, they are also adjacent in the $\mathcal{I}$-essential graph. For the same reason, if $F^{(\mathbf{I}, \mathbf{J})}$ and $Y^{(\mathbf{I})}$ are not adjacent in $\mathcal{G}_I$, we can derive that they are also not adjacent in $\mathcal{E}_I(\mathcal{D}, \mathcal{I})$.

Next, consider the edges that do not contain $F$ nodes. There are no edges between two nodes in different domains in both $\mathcal{G}_\mathbf{I}$ and $\mathrm{Aug}_\mathbf{I}(\mathcal{D}, \mathcal{I})$ by our construction. We call the vertex induced subgraph on $\mathbf{V}^{(\mathbf{J})}, \mathbf{J} \in \mathcal{I}$ a domain. We just need to show that they have the same adjacencies within each domain. According to our algorithm, within each domain, all the edges are connected by applying the FCI algorithm, which is proved to be sound by Zhang [2008b]. Thus, $\mathcal{G}_\mathbf{I}[\mathbf{V}^{(\mathbf{I})}]$ will have the same adjacencies as $\mathrm{Aug}_\mathbf{I}(\mathcal{D}', \mathcal{I})[\mathbf{V}^{(\mathbf{I})}], \mathbf{I} \in \mathcal{I}$ for all $\mathcal{D}'$ that are $\mathcal{I}$-Markov equivalent to $\mathcal{D}$.

Finally, we address the soundness of orientation rules. In phase I of the learning algorithm, we use FCI rules within each domain to learn the skeletons. Zhang [2008b] showed that FCI is sound and complete. Thus, the orientations in the skeleton learned in this phase are shared across all $\mathrm{Aug}_\mathbf{I}(\mathcal{D}')$, for all $\mathcal{D}'$ that is $\mathcal{I}$-Markov equivalent to $\mathcal{D}$. In phase II, $F$ nodes are introduced and adjacent to the symmetric difference of the targets, and the nodes that are not separable from $F$. All the edges induced to $F$ nodes are oriented outgoing from $F$ nodes by Rule 8. This is also true in $\mathrm{Aug}_\mathbf{I}(\mathcal{D}', \mathcal{I})$ for all $\mathcal{D}'$ that is $\mathcal{I}$-Markov equivalent to $\mathcal{D}$. Thus, so far, all the orientations learned are sound. What remains to be shown is the soundness of the extra orientation rules.

**Remark:** In Algorithm 3, for the interventional domains, we return the separating set that includes the interventional target of the domain. This inclusion is required by the do-calculus rules. However, in practice when running Algorithm 1, we can remove the targets from the separating sets to save memory since removing them will not impose any modification on the orientation rules. To witness, the only rules with use separating sets are Rule 0 and Rule 4. Both are deciding if the node of interest is a collider. Since the targets are intervened, they have no incoming edges and thus removing them from the conditioning sets will not drop off any possible colliders. Therefore, one has the flexibility to include the targets in the separating set or not.

**Soundness of Rule** 0**:** If both end nodes are in $\mathbf{V}$, then the soundness is guaranteed by the soundness of FCI. Suppose $\langle F, X^{(I)}, Y^{(I)} \rangle$ is an unshielded collider identified by the learning algorithm, i.e., there is some $\mathbf{W} \subseteq \mathbf{V} \setminus \mathbf{I}$, such that, $F \perp\!\!\!\perp Y^{(I)} | \mathbf{I}^{(\mathbf{I})}, \mathbf{W}^{(I)}$, while $X \notin \mathbf{W} \cup \mathbf{I}$. Suppose otherwise, $\langle F, X^{(I)}, Y^{(I)} \rangle$ is not an unshielded collider, then it can only be $F \to X^{(I)} \to Y^{(I)}$. Since $F, Y^{(I)}$ are non-adjacent, the soundness of the skeleton indicates that there is a set of nodes that separates $F$ from $Y^{(I)}$. However, in this case, to separate $F$ from $Y^{(I)}$, $X^{(I)}$ has to be in the condition set; otherwise, the path $F \to X^{(I)} \to Y^{(I)}$ would be d-connecting.

**Soundness of Rule 9:** To address the soundness of Rule 9, we need to show that if $X^{(\mathbf{I})}, Y^{(\mathbf{I})}$ are adjacent, and $X \in \mathbf{I}$, then $Y$ can only be a descendant of $X$ in the interventional causal graph $\mathcal{D}_{\bar{\mathbf{I}}}$. Suppose otherwise, then $Y$ is an ancestor of $X$ or they have at least one common ancestor in $\mathcal{D}_{\bar{\mathbf{I}}}$. However, since $X$ is intervened, there will be no ancestor of $X$ in $\mathcal{D}_{\bar{\mathbf{I}}}$, which is a contradiction.

**Soundness of Rule 10:** We need to show that if we recover $X \to Y$ in the domain $G^{(\mathbf{I})}$, there cannot be $X^{(\mathbf{J})} \leftarrow Y^{(\mathbf{J})}$ in another domain of $\mathbf{J}$. Suppose otherwise, it indicates that there is a directed path from $Y$ to $X$ in $\mathcal{D}_{\bar{\mathbf{J}}}$. However, there is also a directed path from $X$ to $Y$ in $\mathcal{D}_{\bar{\mathbf{I}}}$. $\mathcal{D}_{\bar{\mathbf{J}}}$ and $\mathcal{D}_{\bar{\mathbf{I}}}$ are subgraphs of $\mathcal{D}$, thus there is at least 1 cycle in $\mathcal{D}$ which is a contradiction.

**Soundness of Rule 11:** $F^{(\mathbf{I},\mathbf{J})}$ is adjacent to $Y^{(\mathbf{I})}, Y^{(\mathbf{J})}$ while $Y \notin \mathbf{I}$ shows that there does not exist any $\mathbf{W} \subseteq \mathbf{V}$ such that $P_{\mathbf{I}}(y|\mathbf{w}) = P_{\mathbf{J}}(y|\mathbf{w})$. This means that there is no separating set that separates both $F^{(\mathbf{I},\mathbf{J})}, Y^{(\mathbf{I})}$ and $F^{(\mathbf{I},\mathbf{J})}, Y^{(\mathbf{J})}$. This indicates that there is an inducing path from $F^{(\mathbf{I},\mathbf{J})}$ to $Y^{(\mathbf{I})}$ or from $F^{(\mathbf{I},\mathbf{J})}$ to $Y^{(\mathbf{J})}$ relative to the latent variables in the augmented pair graph. Due to the definition of the inducing path, it has to go through $X^{(\mathbf{I})}$ or $X^{(\mathbf{J})}$ as $\mathbf{K} = \{X\}$. However, in $G^{(\mathbf{I})}$, since $X^{(\mathbf{I})}$ is intervened, $X^{(\mathbf{I})}$ cannot serve as a collider in the path. Consequently, we can infer that the inducing path has to go through $X^{(\mathbf{J})}$ to $Y^{(\mathbf{J})}$. Furthermore, $X^{(\mathbf{J})}$ has to be a parent of one of the end nodes of the path. Given that $F$ nodes are source nodes, we can orient $X^{(\mathbf{J})} \to Y^{(\mathbf{J})}$. $\qquad\square$

# C Algorithms

## C.1 Algorithm for Creating $F$ Nodes

---
**Algorithm 2** Algorithm for Creating $F$ Nodes

---
**Input:** Intervention set $\mathcal{I}$, intervention $\mathbf{I}$
Initialize $\mathcal{F} \leftarrow \emptyset$;
**for J in $\mathcal{I} \setminus \{\mathbf{I}\}$ do**
$\quad \mathcal{F} \leftarrow \mathcal{F} \cup \{F^{(\mathbf{I},\mathbf{J})}\}$
**Output:** The set of $F$ nodes $\mathcal{F}$

---

## C.2 Algorithm for Finding Separating Sets

---
**Algorithm 3** Algorithm for Finding Separating Set

---
**Input:** Target $\mathbf{I}$ and $\mathbf{J}$, interventional distributions $(P_{\mathbf{I}})_{\mathbf{I} \in \mathcal{I}}$, observable variables $\mathbf{V}$, $F$ nodes $\mathcal{F}$
Initialize $\mathcal{E} \leftarrow \emptyset$;
**if $\mathbf{I} = \mathbf{J}$ then**
$\quad$**for $X, Y$ in $\mathbf{V}$ do**
$\quad\quad SepFlag \leftarrow False$;
$\quad\quad$**for $\mathbf{W} \subset \mathbf{V}$ do**
$\quad\quad\quad$**if $P_{\mathbf{I}}(y|\mathbf{w}, x) = P_{\mathbf{I}}(y|\mathbf{w})$ then**
$\quad\quad\quad\quad SepFlag \leftarrow True$
$\quad\quad\quad\quad SepSet(X^{(\mathbf{I})}, Y^{(\mathbf{I})}) = \mathbf{W}^{(\mathbf{I})} \cup \mathcal{F} \cup \mathbf{I}^{(\mathbf{I})}$;
$\quad\quad$**if $SepFlag = False$ then**
$\quad\quad\quad \mathcal{E} \leftarrow \mathcal{E} \cup (X^{(\mathbf{I})}, Y^{(\mathbf{I})})$;
**else**
$\quad$**for $Y$ in $\mathbf{V}$ do**
$\quad\quad SepFlag \leftarrow False$;
$\quad\quad$**if $Y$ in $\mathbf{I}\Delta\mathbf{J}$ then**
$\quad\quad\quad$Pass;
$\quad\quad$**else**
$\quad\quad\quad$**for $\mathbf{W} \subseteq \mathbf{V}$ do**
$\quad\quad\quad\quad$**if $P_{\mathbf{I}}(y|\mathbf{w}) = P_{\mathbf{J}}(y|\mathbf{w})$ then**
$\quad\quad\quad\quad\quad SepFlag \leftarrow True$;
$\quad\quad\quad\quad\quad SepSet(F, Y^{(\mathbf{I})}) \leftarrow \mathbf{I}^{(\mathbf{I})} \cup \mathbf{W}^{(\mathbf{I})}$;
$\quad\quad\quad\quad\quad SepSet(F, Y^{(\mathbf{J})}) \leftarrow \mathbf{J}^{(\mathbf{J})} \cup \mathbf{W}^{(\mathbf{J})}$;
$\quad\quad$**if $SepFlag = False$ then**
$\quad\quad\quad \mathcal{E} \leftarrow \mathcal{E} \cup (F^{(\mathbf{I},\mathbf{J})}, Y^{(\mathbf{I})})$;
**Output:** Circle edges $\mathcal{E}$, separating sets $SepSet$

---

## C.3 Complexity Analysis

Since our algorithm is also constraint-based—like FCI and PC—the bulk of the computational cost comes from invariance tests, which can be seen as conditional independence tests involving the $F$-node. In the worst case, each pair of nodes may require an exponential number of tests to identify a separating set. However, in practice, many implementations (including FCI and PC) limit the size of the conditioning set (e.g., to a constant such as 3), which reduces the worst-case complexity to a polynomial bound of $O(n^5)$ (see, e.g., Spirtes et al. [2001]).

For the new orientation rules we introduce: Rules 8, 9, and 11 operate in $O(1)$ time since they only check the adjacencies of a fixed number of nodes. Rules 10 and 11 have a complexity of $O(n^2)$ as they potentially examine all pairs of nodes. In comparison, the FCI orientation rules can take up to $O(n^5)$ in the worst-case.

Thus, for each $\mathcal{I}$-augmented MAG, our learning algorithm has a similar worst-case complexity as the FCI algorithm, namely $O(n^5)$. If there are $k$ targets, the total complexity becomes $O(kn^5)$.

## D  An example of the Construction of Twin Augmented MAGs

Graph $\mathcal{D}_1$ in Figure 2a is the original graph with a confounder between $Z, Y$. Assume that the intervention set is $\mathcal{I} = \{\mathbf{I}_1 = \emptyset, \mathbf{I}_2 = \{Z\}\}$. Here we use the index of the targets in the superscripts of the variables for simplicity. Figure 2b shows how we build the augmented pair graph by adding the $F$ node and pointing it to both $Z$s in the two interventional subgraphs. In Figure 2c, we show the MAG of $\mathrm{Aug}_{(\emptyset, \{Z\})}(\mathcal{D}_1)$. The edge of $(F, Y^{(1)})$ is added because there is an inducing path $\langle F, Z^{(1)}, Y^{(1)} \rangle$ in $\mathrm{Aug}_{(\emptyset, \{Z\})}(\mathcal{D}_1)$, which means that $F$ is not m-separable from $Y^{(1)}$ and $F$ is an ancestor of $Y^{(1)}$. Now that $(F, Y^{(1)})$ is presented in $\mathrm{MAG}(\mathrm{Aug}_{(\emptyset, \{Z\})}(\mathcal{D}_1))$ but not $(F, Y^{(2)})$, according to Definition 4.5, we also add the edge of $(F, Y^{(2)})$ to finally construct the twin augmented MAG of $\mathcal{D}_1$, which is $\mathrm{Twin}_{(\emptyset, \{Z\})}(\mathcal{D}_1)$ as illustrated in Figure 2d.

Next, we repeat the same process for $\mathcal{D}_2$ as illustrated in Figure 2e 2f 2g. Comparing $\mathrm{Twin}_{(\emptyset, \{Z\})}(\mathcal{D}_1)$ and $\mathrm{Twin}_{(\emptyset, \{Z\})}(\mathcal{D}_2)$ (Figure 2h), we see that they do not satisfy the three conditions from Theorem 4.7 since they have different skeletons. Hence, we conclude that $\mathcal{D}_1$ and $\mathcal{D}_2$ are not $\mathcal{I}$-Markov equivalent with respect to $\mathcal{I}$. This example also highlights that hard interventions can be more informative than soft interventions because, under a soft intervention on $Z$, one cannot distinguish the two graphs, and they will be $\mathcal{I}$-Markov equivalent using the results from Kocaoglu et al. [2019]. However, with a hard intervention, we can distinguish the two graphs $\mathcal{D}_1$ and $\mathcal{D}_2$ from one another.

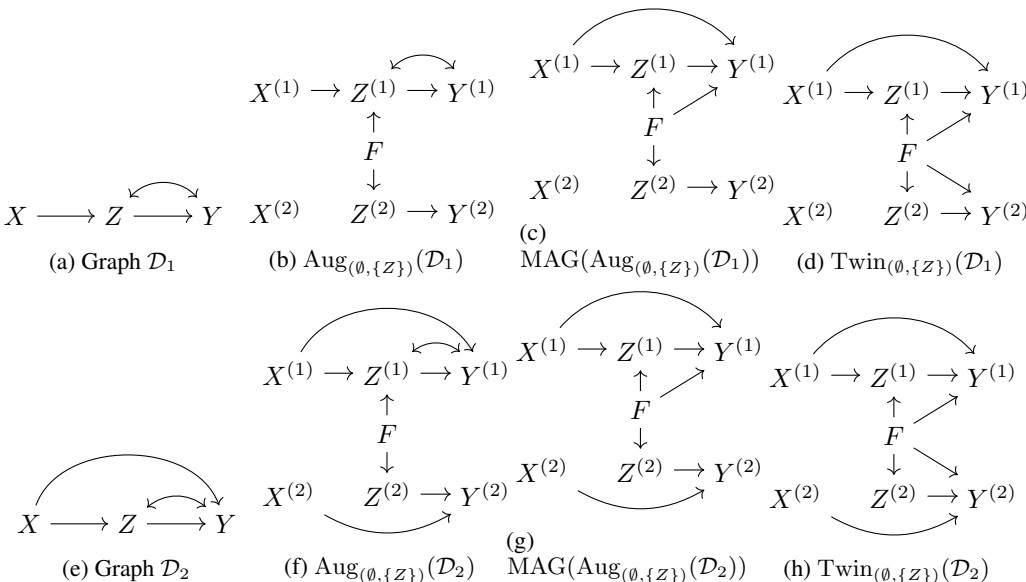

Figure 2: Illustration of the construction of twin augmented MAGs where $\mathcal{D}_1$ and $\mathcal{D}_2$ are not $\mathcal{I}-$Markov equivalent. (a) and (e) are two causal graphs, $\mathcal{D}_1$ and $\mathcal{D}_2$ respectively, given intervention targets $\mathcal{I} = \{\mathbf{I}_1 = \emptyset, \mathbf{I}_2 = \{Z\}\}$. (b) and (f) are the augmented pair graphs for $\mathcal{D}_1$ and $\mathcal{D}_2$ respectively. (c) and (g) are the MAG of the augmented pair graphs for $\mathcal{D}_1$ and $\mathcal{D}_2$ respectively. (d) and (h) are the twin augmented MAGs for $\mathcal{D}_1$ and $\mathcal{D}_2$ respectively. $F \to Y^{(1)}$ in $\mathrm{MAG}(\mathrm{Aug}_{(\emptyset, \{Z\})}(\mathcal{D}_1))$ and $\mathrm{MAG}(\mathrm{Aug}_{(\emptyset, \{Z\})}(\mathcal{D}_2))$ because there is an inducing path $\langle F, Z^{(1)}, Y^{(1)} \rangle$ in both augmented pair graphs. In the twin augmented graphs, we further add $F \to Y^{(2)}$ to make the adjacencies around the $F$ node symmetric.

# E   An Example of the Learning Process

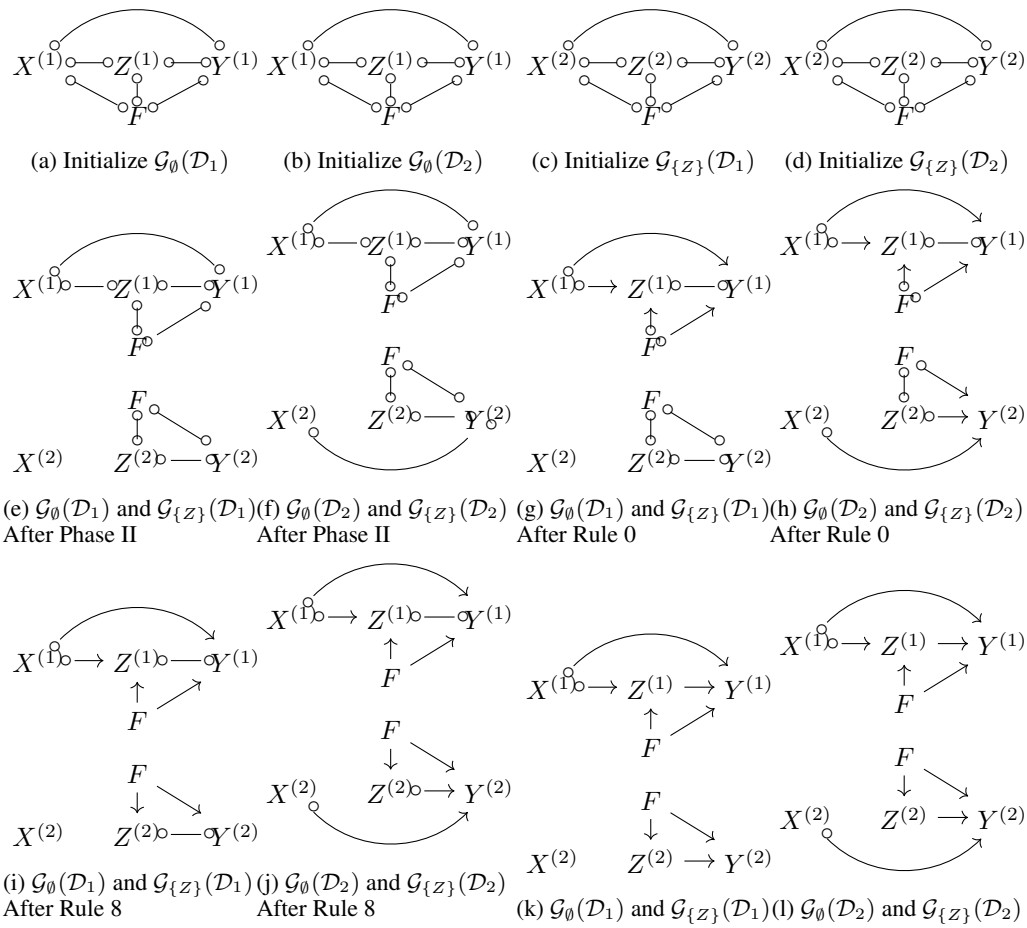

Figure 3: An example of the learning process of Algorithm 1.

Here we show an example of the learning process of Algorithm 1 in Figure 3. Consider the two causal graphs $\mathcal{D}_1, \mathcal{D}_2$ as shown in Figure 2a and Figure 2e respectively. Assume that we have access to $(P_{obs}, P_Z)$, i.e. $\mathcal{I} = \{\mathbf{I}_1 = \emptyset, \mathbf{I}_2 = \{Z\}\}$. In Phase I, we initialize the $\mathcal{I}$-augmented graphs under each interventional target for both graphs by constructing complete graphs with only circle edges within the observable nodes. After that, we create the $F$ nodes between each pair of intervention targets using Algorithm 2 and connect each $F$ node to all observable nodes. The initialized $\mathcal{I}$-augmented graphs are shown in Figure 3a, Figure 3b Figure 3c, and Figure 3d. Here we omit the superscript for $F$ nodes since there are only two domains. Then in Phase II, we learn skeletons by using Algorithm 3 to check the invariance statements. Specifically, for each pair of $X, Y \in \mathbf{V}$, if there exists some $\mathbf{W} \subseteq \mathbf{V}$ that separates $X, Y$ in any domain, we remove the circle edge between $X, Y$ accordingly. Similarly, if $F$ is separable from a vertex in $\mathbf{V}$ given a condition set $\mathbf{W} \subseteq \mathbf{V}$, i.e. $P_\emptyset(y|\mathbf{w}) = P_{\{Z\}}(y|\mathbf{w})$, then we remove the circle edge between $F, Y$ in both domains. Thus we construct the skeletons of $\mathcal{I}$-augmented graphs for $\mathcal{D}_1$ and $\mathcal{D}_2$ as shown in Figure 3e and Figure 3f respectively. The upper graphs are for the observational domain while the lower graphs are for the domain of $P_Z$. Next, in Phase III, we apply the orientation rules. We start by finding all the unshielded colliders using Rule 0. In $\mathcal{G}_\emptyset(\mathcal{D}_1)$, we notice that conditioning on $Z^{(1)}$, $F$ is dependent on $X^{(1)}$. Similarly, conditioning on $Y^{(1)}$, $F$ is dependent on $X^{(1)}$. We can then orient $X^{(1)}o\!\!-\!\!oZ^{(1)}$, $X^{(1)}o\!\!-\!\!oY^{(1)}$, $Fo\!\!-\!\!oZ^{(1)}$, and $Fo\!\!-\!\!oY^{(1)}$ as $X^{(1)}o\!\!\rightarrow Y^{(1)}$, $X^{(1)}o\!\!\rightarrow Z^{(1)}$, $Fo\!\!\rightarrow Z^{(1)}$, and $Fo\!\!\rightarrow Y^{(1)}$ respectively. The same structure appears in $\mathcal{G}_\emptyset(\mathcal{D}_2)$. However, in $\mathcal{G}_\emptyset(\mathcal{D}_2)$, we can identify the unshielded triplets $\langle X^{(2)}, Y^{(2)}, Z^{(2)}\rangle$ and $\langle X^{(2)}, Y^{(2)}, F\rangle$ which help us orient the colliders accordingly. The resulting graphs are plotted in Figure 3g and Figure 3h. Using Rule 8,

we orient all edges incident to $F$ out of $F$. The $\mathcal{I}$-augmented graphs after Rule 8 are presented in Figure 3i and Figure 3j. Since only $Z$ is intervened but not $Y$, we can orient the edge $Z^{(2)} \to Y^{(2)}$ in both $\mathcal{G}_{\{Z\}}(\mathcal{D}_1)$ and $\mathcal{G}_{\{Z\}}(\mathcal{D}_2)$ using Rule 9. While Rule 10 helps us to orient $Z^{(1)}o{\to} Y^{(1)}$, Rule 11 implies that we can orient $Z^{(1)} \to Y^{(1)}$ since $F$ points to $Y^{(1)}, Y^{(2)}$ in both $\mathcal{G}_{\emptyset}(\mathcal{D}_1)$ and $\mathcal{G}_{\emptyset}(\mathcal{D}_2)$ as $Y$ is not intervened. Thus, there has to be an inducing path from $F$ to $Y^{(1)}, Y^{(2)}$ that goes through $Z^{(1)}$ or $Z^{(2)}$ in both augmented pair graphs for $\mathcal{D}_1$ and $\mathcal{D}_2$. Notice that these edges cannot be oriented by the FCI rules. At this stage, none of the rules apply anymore; therefore, $\mathcal{G}_{\emptyset}(\mathcal{D}_1), \mathcal{G}_{\{Z\}}(\mathcal{D}_1)$ and $\mathcal{G}_{\emptyset}(\mathcal{D}_2), \mathcal{G}_{\{Z\}}(\mathcal{D}_2)$ are returned as the learned $\mathcal{I}$-augmented graphs of Algorithm 1. The final results are shown in Figure 3k and Figure 3l. Notice that the learned $\mathcal{I}$-augmented graphs of the two causal graphs have different adjacencies and unshielded colliders. Therefore, they can be distinguished by the conditions listed in Theorem 4.7.

## F Experiment Details

### F.1 Enumerate All ADMGs

In this experiment, we aim to demonstrate that the $\mathcal{I}$-MEC size under hard interventions is smaller than that under soft interventions in general. Here is how we set up the experiments. Given the number of observable variables $n$, we iterate through all possible ADMGs structures with $n$ nodes. The number of such ADMGs can be found as follows. We first generate all possible DAGs (not necessarily connected) with $n$ nodes. Then, for each pair of nodes in a DAG, there can be a bidirected edge or not. Thus, the number of ADMGs under consideration is $2^{\binom{n}{2}}$ multiplied by the number of all DAGs. Each time, we randomly pick an ADMG $G$ as the ground truth graph. Assuming that the size of the intervention targets is 2, we construct the twin augmented MAG $\mathcal{M}_1$ of $G$ following Definition 4.5 and the Augmented MAG $\mathcal{M}_2$ of $G$ following Definition 4 in Kocaoglu et al. [2019]. We then enumerate all possible ADMGs, and for each candidate ADMG $G'$, we construct its corresponding $\mathcal{M}'_1$ and $\mathcal{M}'_2$, and compare $\mathcal{M}_1, \mathcal{M}'_1$ following Theorem 4.7, $\mathcal{M}_2, \mathcal{M}'_2$ following Theorem 2 in Kocaoglu et al. [2019]. For each $n$, we repeat the experiment 30 times and calculate the average size of $\mathcal{I}$-MEC size together with standard errors. We consider two types of sampling ADMGs, random and complete. 'Random' means the ADMG is constructed by randomly adding directed and bidirected edges with a probability of 0.5 between each pair of nodes while not creating any cycle. 'Complete' means we first construct a random complete DAG and then randomly add bidirected edges with a probability of 0.5 between each pair. For intervention targets, we assume they are either atomic or an empty set. We only show the results for small $n$, as the number of ADMGs grows super-exponentially with $n$. All the experiments are run on an NVIDIA GeForce RTX 3090 graphics card. The numerical results are shown in Table 2. We notice that the number of ADMGs grows very fast with $n$. The results have a high standard error. This is due to the differences between the sampled graphs. Nevertheless, under hard interventions, the $\mathcal{I}$-MEC size tends to shrink the $\mathcal{I}$-MEC size more efficiently than with soft interventions on average, showing the power of hard interventions. Furthermore, as $n$ grows, the ratio of hard $\mathcal{I}$-MEC divided by the soft $\mathcal{I}$-MEC decreases, meaning that hard interventions extract more information on the causal graphs.

| $n$ | Mean under Hard | Mean under Soft | Graph | Ratio | Number of ADMGs |
|-----|-----------------|-----------------|-------|-------|-----------------|
| 2 | $2.03 \pm 0.15$ | $2.93 \pm 0.29$ | Random | $0.69 \pm 0.05$ | 6 |
| 2 | $2.37 \pm 0.12$ | $3.67 \pm 0.22$ | Complete | $0.65 \pm 0.05$ | 6 |
| 3 | $19.50 \pm 3.41$ | $30.57 \pm 4.36$ | Random | $0.64 \pm 0.11$ | 200 |
| 3 | $14.03 \pm 2.69$ | $24.70 \pm 4.12$ | Complete | $0.57 \pm 0.05$ | 200 |
| 4 | $677.13 \pm 227.72$ | $1218.83 \pm 361.83$ | Random | $0.56 \pm 0.18$ | 34,752 |
| 4 | $721.37 \pm 276.36$ | $1529.57 \pm 368.68$ | Complete | $0.47 \pm 0.07$ | 34,752 |

Table 2: Estimation of $\mathcal{I}$-MEC size by enumerating all ADMGs of the same size. We consider random ADMGs and ADMGs by adding bidirected edges to random complete DAGs. For each setting we sample 30 ground truth ADMGs and calculate the mean and standard error of $\mathcal{I}$-MEC size and the ratio.

## F.2 Sample ADMGs

When $n = 5$, the total number of valid ADMGs is 29,983,744, which is intractable to enumerate. Instead, we can sample ADMGs to estimate the expectation of the probability of a randomly sampled ADMG to be $\mathcal{I}$-Markov equivalent to a given ADMG using Hoeffding's Inequality. To do this, given $n$, we randomly sample a DAG that is a complete graph using the uniform CPDAG sampler by Wienöbst et al. [2021]. Then, for each sampled complete DAG, we add bidirected edges uniformly to each pair of nodes to construct a ground truth ADMG. To compare with the ADMG, we randomly pick two intervention targets that are either an empty set or atomic, and then randomly sample ADMGs following the same process and construct the augmented graphs. Suppose for each ground truth ADMG, we draw $M$ such random samples. For the $i$-th sample, $S_i = 1$, if it is $\mathcal{I}$-Markov equivalent to the true graph and $S_i = 0$ otherwise. We define $\mathcal{S} = \sum_{i=1}^{M} S_i$ which shows the number of $\mathcal{I}$-Markov equivalent ADMGs. We denote $\mathbb{E}_{\mathcal{S}}$ as the expectation we are approximating. Thus, according to Hoeffding's Inequality, we have:

$$P\left(\left|\frac{\mathcal{S}}{M} - \mathbb{E}_{\mathcal{S}}\right| \geq \epsilon\right) \leq \exp(-2M\epsilon^2) \tag{7}$$

We choose $\epsilon = 0.01$ and $\exp(-2M\epsilon^2) = 0.01$ for $M$ with $M = 23025$. For each setting, we randomly sample 50 ground truth ADMGs and then take the average. The results are shown in Table 3, Table 4, and Table 5. We can see that the estimated $\mathbb{E}_{\mathcal{S}}^{hard}$ is significantly lower than $\mathbb{E}_{\mathcal{S}}^{soft}$ meaning hard interventions can more efficiently learn the causal structure. Notice that as $n$ becomes larger, the number of ADMGs grows fast and thus the expectations get close to 0. Consequently, a much smaller $\epsilon$ would be necessary to approximate the expectations, leading to a much larger number of samples $M$.

| $n$ | Estimated $\mathbb{E}_{\mathcal{S}}^{hard}$ | Estimated $\mathbb{E}_{\mathcal{S}}^{soft}$ | Ratio |
|---|---|---|---|
| 2 | $0.417 \pm 0.010$ | $0.584 \pm 0.010$ | $0.715 \pm 0.011$ |
| 3 | $0.143 \pm 0.012$ | $0.235 \pm 0.016$ | $0.607 \pm 0.022$ |
| 4 | $0.058 \pm 0.011$ | $0.112 \pm 0.015$ | $0.514 \pm 0.024$ |
| 5 | $0.028 \pm 0.011$ | $0.061 \pm 0.013$ | $0.459 \pm 0.028$ |
| 6 | $0.015 \pm 0.010$ | $0.036 \pm 0.012$ | $0.420 \pm 0.030$ |

Table 3: Estimation of $\mathcal{I}$-MEC size on complete DAGs with different sizes and $0.5$ density of bidirected edges.

| $n$ | Estimated $\mathbb{E}_{\mathcal{S}}^{hard}$ | Estimated $\mathbb{E}_{\mathcal{S}}^{soft}$ | Ratio |
|---|---|---|---|
| 3 | $0.151 \pm 0.011$ | $0.264 \pm 0.015$ | $0.571 \pm 0.017$ |
| 4 | $0.067 \pm 0.010$ | $0.149 \pm 0.013$ | $0.451 \pm 0.017$ |
| 5 | $0.034 \pm 0.010$ | $0.091 \pm 0.013$ | $0.373 \pm 0.017$ |
| 6 | $0.019 \pm 0.010$ | $0.059 \pm 0.012$ | $0.317 \pm 0.017$ |

Table 4: Estimation of $\mathcal{I}$-MEC size with different $n$ and fixed $0.45n(n - 1)$ bidirected edges (approximately $0.9$ in density).

| $\rho$ | Estimated $\mathbb{E}_{\mathcal{S}}^{hard}$ | Estimated $\mathbb{E}_{\mathcal{S}}^{soft}$ | Ratio |
|---|---|---|---|
| 0.1 | $0.020 \pm 0.011$ | $0.024 \pm 0.011$ | $0.804 \pm 0.018$ |
| 0.3 | $0.022 \pm 0.011$ | $0.040 \pm 0.012$ | $0.566 \pm 0.030$ |
| 0.5 | $0.028 \pm 0.011$ | $0.061 \pm 0.013$ | $0.459 \pm 0.028$ |
| 0.7 | $0.031 \pm 0.011$ | $0.081 \pm 0.013$ | $0.395 \pm 0.021$ |
| 0.9 | $0.034 \pm 0.010$ | $0.093 \pm 0.011$ | $0.365 \pm 0.015$ |

Table 5: Estimation of $\mathcal{I}$-MEC size on completed DAGs with 5 nodes and different densities of bidirected edges. It shows that when the density of bidirected edges goes up, hard interventions shrink the $\mathcal{I}$-MEC size more efficiently than soft interventions.

### F.3 Comparison of Learning Objectives with Other Interventional Causal Discovery Algorithms

Most existing interventional causal discovery algorithms aim to learn a single causal graph. However, the goal of our learning algorithm is to learn a tuple of $\mathcal{I}$-augmented graphs that entails (a superset of) $\mathcal{I}$-Markov equivalent ADMGs. None of the previous approaches outputs the same object. Concretely, $\mathcal{I}$-FCI [Kocaoglu et al., 2019][6] (known targets)/$\psi$-FCI [Jaber et al., 2020] (unknown targets) assume soft interventions and cannot exploit the additional invariance constraints incurred by hard interventions. They use a single augmented MAG as the learning objective. JCI [Mooij et al., 2020] claims it works for any interventions but it is overly simplistic and the learned PAG is less informative (in Jaber et al. [2020] App. D shows that $\psi$-FCI can learn more than JCI). GIES [Hauser and Bühlmann, 2012] and IGSP [Wang et al., 2017] are score-based and aim at learning a single DAG under known targets. They neither guarantee inclusion-minimality nor provide equivalence-class certificates. Nevertheless, score-based methods usually impose parametric assumptions like linear function or additive Gaussian noise to the causal models. Such methods are not robust when the underlying mechanism deviates from those assumptions. Additionally, GIES and IGSP assume no latents and therefore their outputs are less informative. Because the outputs are of different types, a direct performance metric is unavailable to measure the performance of the methods. The previous sections have shown that empirically how much smaller the $\mathcal{I}$-MEC is vs. the soft $\mathcal{I}$-MEC produced by $\mathcal{I}$-FCI/$\psi$-FCI. Here we compare with GIES and IGSP with a simple example.

Consider the graph $[X \rightarrow, Z \rightarrow Y, Z \leftarrow L \rightarrow Y]$ where $L$ is a latent variable. We generate a Bayesian network with binary variables according to the graph. We take 100k samples from the observational distribution and 100k samples from $P_Z$. We choose 100k as it is more than enough to return correct CI tests. GIES returns $[Y \rightarrow Z \rightarrow X]$ which has the wrong causal order and does not belong to the same observational MEC. IGSP returns $[X \rightarrow Z \rightarrow Y, X \rightarrow Y]$ which is the observational MAG of the true graph, but it cannot tell anything to judge if there is a latent confounder between $Z, Y$. Under the same setting, when the simulated dataset is from a linear Gaussian structural causal model, both GIES and IGSP output $[X \rightarrow Z \rightarrow Y]$ which catches the right causal order but does not belong to the observational MEC. In both settings, our algorithm is able to learn the graphs illustrated in Figure 3k which preserve information from both domains. We use the Python implementation of GIES by Olga Kolotuhina and Juan L. Gamella [Gamella, 2025]. The IGSP implementation is from the `causaldag` package [Squires, 2018].

## G   Further Discussion

### G.1   Assumptions

#### G.1.1   Positivity

In theory, Algorithm 1 does not require a strict global positivity assumption. All results rely solely on h-faithfulness. "Strictly positive distribution" is added in some works on discovery as a convenience. This is because within the space of strictly positive parameterizations, the subset that violates faithfulness sits on algebraic varieties of lower dimension, hence has Lebesgue measure 0 (see Robins et al. [2003] and many follow-ups). In practice, with finite data, a zero or near-zero cell makes test statistics undefined or unstable, so implementations often impose minimum-count rules even when the theory would allow the zero.

The positivity assumption is required for identifiability (for example, Kivva et al. [2022], Kandasamy et al. [2019]). This is because that the manipulation of numerical expression explicitly require the denominator to be strictly positive. By contrast, structural discovery uses only qualitative independence relations. As long as we can evaluate a test where, no global support assumption is required.

#### G.1.2   Faithfulness

There are weaker faithfulness assumptions. However, the soundness of Algorithm 1 requires h-faithfulness. The $\mathcal{I}$-MEC we defined is based on the generalized do-calculus rules and so as the learning algorithm and hence h-faithfulness is minimal for the algorithm to be sound.

---

[6]The algorithm was proposed in Kocaoglu et al. [2019] and later named as $\mathcal{I}$-FCI by Li et al. [2023].

Like classic faithfulness, the set of parameterizations that violate h-faithfulness has Lebesgue measure zero for standard SEM families with continuous noise if we consider linear models; i.e., it holds generically (it also holds for many other models, see follow-ups of Robins et al. [2003]). It is therefore no stronger in a measure–theoretic sense than ordinary faithfulness, but it is stronger logically: it rules out a few additional, finely-tuned parameter combinations that would create the extra cancellations specific to our Rules 1-4—those cancellations correspond exactly to our new orientation rules, so dropping h-faithfulness would break thesoundness. From the high level, imagine a causal graph under faithfulness assumption. It could be the case that for certain conditioning values $Z = z$, two variables $X$ and $Y$ that are d-connected are independent. While faithfulness covers the case that $X$ is dependent with $Y$ given $Z$ (when the dependency metric is averaged over all $z$), h-faithfulness is similar to saying $X, Y$ are dependent given every $Z = z$. This analogy is exact if the intervened node is a source node, but we believe it gives some insight on the leap from observational faithfulness to h-faithfulness

### G.2 Comparison between Hard Interventions and Soft Interventions

One may expect that hard interventions can always extract more information about the causal graph than soft interventions when there are latents. Here, we show an example in Figure 4 where this is not true. Consider the ground truth ADMG $\mathcal{D}$ in Figure 4a with intervention targets $\mathcal{I} = \{\mathbf{I}_1 = \{X_1, X_2\}, \mathbf{I}_2 = \{Y_1, Y_2\}\}$. The $\mathcal{I}$-augmented MAGs under hard interventions are shown in Figure 4b and Figure 4c respectively. The augmented MAG under soft interventions is shown in Figure 4d. We suppress the superscript of $F$ nodes since they are the same. We notice that under hard interventions, the skeletons in both domains are empty graphs because the hard interventions remove all the bidirected edges in $\mathcal{D}$. However, under soft interventions, the skeleton of $\mathcal{D}$ is preserved since soft interventions do not modify the graphical structures. As a result, any graph that has only bidirected edges between $X_i$ and $Y_j$, $i, j \in \{1, 2\}$ is $\mathcal{I}$-Markov to $\mathcal{D}$ given $\mathbf{I}_1, \mathbf{I}_2$ as hard interventions while this is not true for soft interventions with the same targets. For example, consider the two graphs $\mathcal{D}_1, \mathcal{D}_2$ plotted in Figure 4e and Figure 4f. Given $\mathcal{I}$, their domain-specific skeletons will all be empty graphs as $\mathcal{D}$ when $\mathcal{I}$ is hard, thus they are $\mathcal{I}$-Markov equivalent to $\mathcal{D}$. However, with soft interventions, their skeletons can be preserved and thus not $\mathcal{I}$-Markov equivalent to $\mathcal{D}$ when $\mathcal{I}$ is soft. Therefore, for this kind of graphs and intervention targets, soft interventions may end up with a smaller $\mathcal{I}$-Markov equivalence class than hard interventions.

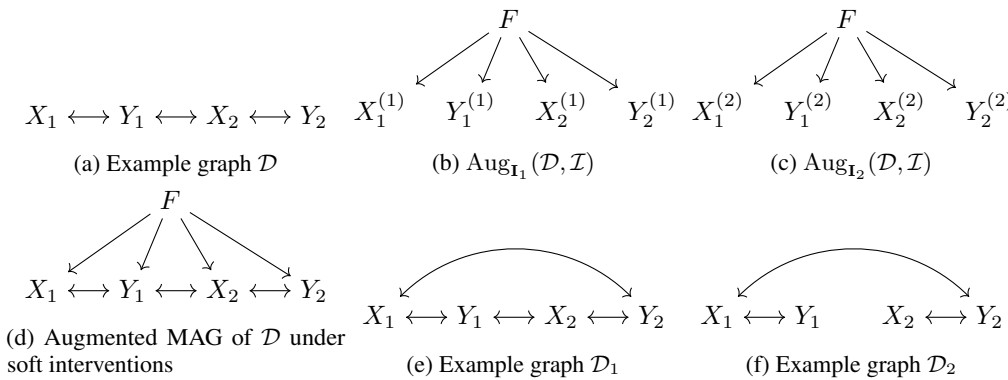

Figure 4: An example that soft interventions lead to a smaller $\mathcal{I}$-Markov equivalence class than hard interventions. (a) is the ground truth causal graph $\mathcal{D}$ with intervention targets $\mathcal{I} = \{\{X_1, X_2\}, \{, Y_1, Y_2\}\}$; (b) and (c) are the two $\mathcal{I}$-augmented MAGs under hard interventions; (d) shows the augmented MAG under soft interventions; (e) and (f) are two examples graphs that are $\mathcal{I}$-Markov equivalent to $\mathcal{D}$ when $\mathcal{I}$ is hard but not $\mathcal{I}$-Markov equivalent to $\mathcal{D}$ when $\mathcal{I}$ is soft.

If the observational distribution is provided, the skeleton is preserved for both hard and soft interventions. However, the following example in Figure 5 shows that even given the observational distribution, there still exist causal graphs which can be distinguished by soft interventions but not so with hard interventions. Consider the causal graphs $\mathcal{D}_1$ and $\mathcal{D}_2$ as shown in Figure 5a and Figure 5b with intervention targets $\mathbf{I}_1 = \emptyset, \mathbf{I}_2 = \{Y\}$, and $\mathbf{I}_3 = \{X, Y\}$. Here we use the index of the targets for superscripts and subscripts for simplicity. The augmented MAGs of $\mathcal{D}_1$ and $\mathcal{D}_2$

under soft interventions are shown in Figure 5c and Figure 5d respectively. We can see that the triple $\langle F^{(2,3)}, X, Y \rangle$ forms an unshielded collider in the augmented MAG of $\mathcal{D}_1$ but it is a non-collider in the augmented MAG of $\mathcal{D}_2$. Therefore, they do not satisfy the 3 conditions, meaning that they are not $\mathcal{I}$-Markov equivalent when $\mathcal{I}$ is soft. The $\mathcal{I}$-augmented MAGs for $\mathcal{D}_1$ and $\mathcal{D}_2$ are shown in Figure 5e, 5f, 5g and Figure 5h, 5i, 5j respectively. We can see that all the 3 pairs of corresponding $\mathcal{I}$-augmented MAGs satisfy the 3 conditions. Therefore, $\mathcal{D}_1$ and $\mathcal{D}_2$ are $\mathcal{I}$-Markov equivalent under hard interventions although the observational distribution is given. If an additional intervention target $\mathbf{I}_4 = \{X\}$ is given, $\mathcal{D}_1, \mathcal{D}_2$ will be non-$\mathcal{I}$-Markov equivalent with a different collider status of the triple $\langle F^{(1,4)}, X^{(4)}, Y^{(4)} \rangle$ in $\mathrm{Aug}_{\mathbf{I}_4}(\mathcal{D}_1, \mathcal{I})$ and $\mathrm{Aug}_{\mathbf{I}_4}(\mathcal{D}_2, \mathcal{I})$. We conclude that although hard interventions are stronger than soft interventions, they do not always lead to a smaller $\mathcal{I}$-Markov equivalence class than soft interventions. It may also imply that using a mixture of hard and soft interventions may be more efficient in causal discovery than using either of them alone.

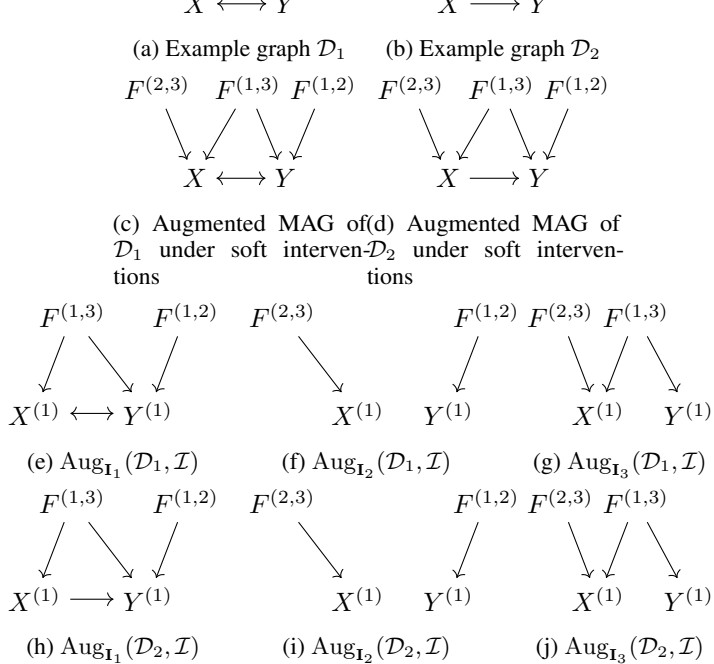

$$X \longleftrightarrow Y \qquad\qquad X \longrightarrow Y$$

(a) Example graph $\mathcal{D}_1$    (b) Example graph $\mathcal{D}_2$

(c) Augmented MAG of (d) Augmented MAG of $\mathcal{D}_1$ under soft interven- $\mathcal{D}_2$ under soft interventions tions

(e) $\mathrm{Aug}_{\mathbf{I}_1}(\mathcal{D}_1, \mathcal{I})$    (f) $\mathrm{Aug}_{\mathbf{I}_2}(\mathcal{D}_1, \mathcal{I})$    (g) $\mathrm{Aug}_{\mathbf{I}_3}(\mathcal{D}_1, \mathcal{I})$

(h) $\mathrm{Aug}_{\mathbf{I}_1}(\mathcal{D}_2, \mathcal{I})$    (i) $\mathrm{Aug}_{\mathbf{I}_2}(\mathcal{D}_2, \mathcal{I})$    (j) $\mathrm{Aug}_{\mathbf{I}_3}(\mathcal{D}_2, \mathcal{I})$

Figure 5: An example that given the observational distribution, soft interventions can distinguish the given two causal graphs while hard interventions cannot. (a), (b) are the ground truth causal graph $\mathcal{D}_1, \mathcal{D}_2$ respectively with intervention targets $\mathcal{I} = \{\emptyset, \{Y\}, \{X, Y\}\}$; (c), (d) are the augmented MAGs under soft interventions for $\mathcal{D}_1$ and $\mathcal{D}_2$ respectively; (e), (f), (g) are the $\mathcal{I}$-augmented MAGs under hard interventions for $\mathcal{D}_1$; (h), (i), (j) are the $\mathcal{I}$-augmented MAGs under hard interventions for $\mathcal{D}_2$. Notice that $\mathcal{D}_1$ and $\mathcal{D}_2$ are not $\mathcal{I}$-Markov equivalent when $\mathcal{I}$ is soft because $\langle F, X, Y \rangle$ have different unshielded collider status in the corresponding augmented MAGs. However, they are $\mathcal{I}$-Markov equivalent when $\mathcal{I}$ is hard because their $\mathcal{I}$-augmented MAGs corresponding to the same domains all satisfy the 3 conditions.

### G.3 Incompleteness of the Learning Algorithm

In this section, we will present an example in Figure 6 to show that Algorithm 1 is not complete, i.e., for some causal graphs and intervention targets $\mathcal{I}$, there exist circles marks in the $\mathcal{I}$-augmented graph returned by Algorithm 1 that can be an arrowhead in the $\mathcal{I}$-augmented MAG of a causal graph that is $\mathcal{I}$-Markov equivalent, and an arrowtail in another $\mathcal{I}$-augmented MAG. Let us consider the causal graph $\mathcal{D}$ as shown in Figure 6a with intervention targets $\mathcal{I} = \{\mathbf{I}_1 = \{X_2\}, \mathbf{I}_2 = \{X_4\}\}$. Here we use the domain index as the superscripts for simplicity. The $\mathcal{I}$-augmented MAGs $\mathrm{Aug}_{\mathbf{I}_1}(\mathcal{D}, \mathcal{I})$ and $\mathrm{Aug}_{\mathbf{I}_2}(\mathcal{D}, \mathcal{I})$ are shown in Figure 6b and Figure 6c respectively. The $\mathcal{I}$-augmented graphs $\mathcal{G}_{\mathbf{I}_1}(\mathcal{D}, \mathcal{I})$ and $\mathcal{G}_{\mathbf{I}_2}(\mathcal{D}, \mathcal{I})$ learned by Algorithm 1 are shown in Figure 6d and Figure 6e respectively. $X_2^{(1)} \rightarrow X_3^{(1)}$ and $X_4^{(2)} \rightarrow X_3^{(2)}$ are oriented by the inducing path rule. However, our algorithm cannot orient

$X_4^{(1)} \to X_3^{(1)}$ and $X_2^{(2)} \to X_3^{(2)}$. Without loss of generality, let's consider $X_4^{(1)} o\!\!\to X_3^{(1)}$ in $\mathcal{G}_{\mathbf{I}_1}(\mathcal{D})$. Suppose that there exists a graph $\mathcal{D}'$ such that there is $X_4^{(1)} \leftrightarrow X_3^{(1)}$ in $\mathcal{G}_{\mathbf{I}_1}(\mathcal{D}', \mathcal{I})$. Since we have $X_4^{(2)} \to X_3^{(2)}$ in $\mathcal{G}_{\mathbf{I}_2}(\mathcal{D}', \mathcal{I})$, this indicates that there is an inducing path and a directed path from $X_4$ to $X_3$ in $\mathcal{D}'_{\overline{X_4}}$ and thus in $\mathcal{D}'$ as $\mathcal{D}'_{\overline{X_4}}$ is a subgraph of $\mathcal{D}'$. The bidirected edge between $X_4^{(1)}$ and $X_3^{(1)}$ in $\mathcal{G}_{\mathbf{I}_1}(\mathcal{D}', \mathcal{I})$ indicates that the directed path from $X_4$ to $X_3$ is broken in $\mathcal{D}'_{\overline{X_2}}$. This can only happen when $X_2$ is in the directed path between $X_4$ and $X_3$ in $\mathcal{D}'$ and thus $\mathcal{D}'_{\overline{X_4}}$ as removing the edges into $X_4$ does not affect its descendants. Therefore, $X_4$ has a child other than $X_3$ and is an ancestor of $X_2$. Nevertheless, $X_4^{(2)}$ has only one possible child $X_3^{(2)}$ in $\mathcal{G}_{\mathbf{I}_2}(\mathcal{D}, \mathcal{I})$. Thus, the supposition does not hold. For the same reason, we can orient the circle mark in $X_2^{(2)} o\!\!\to X_3^{(2)}$ in $\mathcal{G}_{\mathbf{I}_2}(\mathcal{D}, \mathcal{I})$. These circle marks can only be oriented by comparing the graphical structures across domains. We need extra orientation rules to catch them.

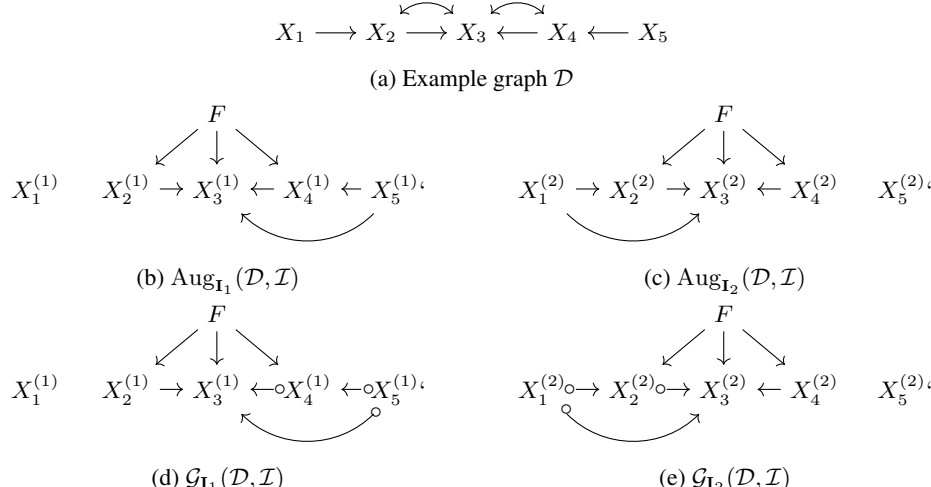

$$X_1 \longrightarrow X_2 \longrightarrow X_3 \longleftarrow X_4 \longleftarrow X_5$$

(a) Example graph $\mathcal{D}$

(b) $\mathrm{Aug}_{\mathbf{I}_1}(\mathcal{D}, \mathcal{I})$

(c) $\mathrm{Aug}_{\mathbf{I}_2}(\mathcal{D}, \mathcal{I})$

(d) $\mathcal{G}_{\mathbf{I}_1}(\mathcal{D}, \mathcal{I})$

(e) $\mathcal{G}_{\mathbf{I}_2}(\mathcal{D}, \mathcal{I})$

Figure 6: An example to show that Algorithm 1 is not complete. (a) is the ground truth causal graph $\mathcal{D}$ with intervention targets $\mathcal{I} = \{\{X_2\}, \{X_4\}\}$; (b) and (c) are the $\mathcal{I}$-augmented MAG under $\mathbf{I}_1$ and $\mathbf{I}_2$ respectively; (d) and (e) are the domain-specific $\mathcal{I}$-augmented graphs learned by Algorithm 1 under $\mathbf{I}_1$ and $\mathbf{I}_2$ respectively. The proposed Algorithm 1 cannot recover the circle marks at $X_2$ and $X_4$, but there does not exist a causal graph that has an arrowhead at the same places in their $\mathcal{I}$-augmented MAGs.

### G.4 Limitations, Future Work, and Broader Impact

**Limitations:** While our work provides a general characterization of $\mathcal{I}$-Markov equivalence classes, the resulting representation is a tuple of augmented graphs, rather than a single unified graph. This makes the characterization less straightforward to interpret. In addition, although our proposed algorithm is sound, it is not complete. We just illustrate this limitation with a concrete example provided in Appendix G.3.

**Future Work:** This work opens several promising directions for future research. First, our empirical results suggest that hard interventions tend to be more informative than soft interventions. It would be valuable to formally analyze this observation and establish theoretical conditions under which this holds. Additionally, while we have proven the soundness of our learning algorithm, extending it with additional orientation rules to achieve completeness remains an open challenge. Finally, our current framework assumes full access to interventional distributions. Developing methods that can learn causal graphs from limited interventional samples is an important direction for making these approaches more practical in real-world settings.

**Broader Impact:** This work contributes to advancing the theoretical and algorithmic foundations of causal discovery, with the potential to improve decision-making/causal inference in critical domains such as healthcare, economics, genomics, and social sciences [Sharma et al., 2022, Zhou et al., 2022, Hernán and Robins, 2016, Sachs et al., 2005, Taubman et al., 2014, Lee, 2008, Belyaeva et al.,

2021]. By combining observational and interventional data, the proposed methods can reduce reliance on large-scale experimentation, making causal analysis more accessible in data-scarce or resource-limited settings. This could empower practitioners to build more reliable models for understanding complex systems, ultimately benefiting scientific discovery and evidence-based policy-making.

However, the use of causal discovery methods also carries risks. Incorrect or biased causal inferences—whether due to data limitations, modeling assumptions, or algorithmic shortcomings—could lead to misguided conclusions, especially in sensitive applications like healthcare or policy-making. Furthermore, there is a risk of misuse if these methods are deployed without adequate validation or oversight, potentially reinforcing harmful biases or supporting flawed decision-making processes.

To mitigate these risks, it is crucial to promote transparency, reproducibility, and the inclusion of domain expertise when applying these methods. Future work should also explore techniques to quantify uncertainty in causal conclusions, improve model interpretability, and develop guidelines or safeguards for responsible deployment.

