# OpenReview forum: "Characterization and Learning of Causal Graphs from Hard Interventions"
_NeurIPS.cc/2025/Conference — NeurIPS 2025 poster_

### Official Review · Reviewer_QBQf · 2025-06-20

**Clarity:** 3
**Significance:** 3
**Originality:** 3
**Rating:** 5
**Confidence:** 4

**Summary:**

The paper characterizes $\cal I$ -Markov equivalence class for a given hard intervention set $\cal I$, by generalizing do-calculus under hard interventions. Prior work established similar results for only soft interventions, and to my knowledge, the present work is the first paper to extend those results specifically to hard interventions, leveraging the additional conditional independence constraints imposed by the hard interventions. Specifically, two graph structures, twin augmented MAG and $\cal I$-augmented MAGs, are proposed to capture $\cal I$-Markov equivalence (for hard interventions) graphically. In addition, a sound learning algorithm is proposed for finding this equivalence class. It is empirically shown that the $\cal I$-MEC under hard interventions is in general significantly smaller than the one under soft interventions, which validates the motivation of the paper to address the gap (for hard interventions) in the literature.

**Questions:**

### Questions
* If I understand correctly, the proofs only require the surgical properties of a hard intervention, that is intervened node has no parents left. A hard intervention, as common in literature, is defined as the deterministic do intervention via setting the intervened variables to constants. I’d like to ask whether the results are still valid for “stochastic hard interventions”, sometimes called perfect interventions. A remark about this could be helpful.
* Augmented pair graph. I’m curious, are you confident that the “augmented graph” construction in [19] is insufficient for developing the results in this paper? In other words, why is it truly necessary to extend that definition to “augmented pair graph”? This is not a criticism but rather an attempt to digest the results.
* Experiments: Were you able to run experiments for actually learning the graphs, for small graphs perhaps? If I’m not mistaken, the experiments in the paper focus only on the size of I-MEC.
* h-faithfulness: Do you have any comment on how strong this assumption is? I understand that some faithfulness assumption is required for CI-based causal discovery, I’m just curious if a milder form would still be sufficient (e.g., adjacency faithfulness is milder than standard faithfulness in classical causal discovery)

**Minor points**
- Given the use of “I-MEC” in prior work and the hard-intervention specific nature of this paper, I wonder whether a distinctive name, e.g., I-Hard-MEC, would have been more appropriate.

- Line 185: A soft intervention, in general, leaves the interventional graph the same. However, as a special case, it can change the graph, e.g., cutting one parent while leaving the other parents intact.

**Ethical Concerns:**

["NO or VERY MINOR ethics concerns only"]

**Final Justification:**

I've had an initial positive review, with score 5 (please see my initial review).

During the rebuttal process, I have also read the other reviews and authors’ responses. I think the “Background and our contribution” walk-through response to Reviewer 4BQV is a good summary for improving the exposition of the paper. I trust that with the promised changes—these explanations and adding some figures to the main text— the revised paper will be an easier read, and will showcase the importance and contributions better (which I already appreciate, as reflected in my positive view).

Thus, I maintain my positive score and recommend acceptance.

**Limitations:**

Yes.

**Paper Formatting Concerns:**

No.

**Quality:**

3

**Strengths And Weaknesses:**

### strengths
* Characterizing the interventional MEC under hard interventions, which the paper shows to be significantly smaller than the one for soft interventions in most cases, is surprisingly a hitherto unknown problem, to my knowledge. For this reason alone, I think the paper has strong contributions to causal discovery literature. The learning algorithm is not perfect (see my comments below); yet, the characterization results and the well-thought-out limitations discussion (in the appendix) suggest that this paper might draw significant attention and serve as the groundwork for future follow-up papers in the causal discovery community.
* The paper is mostly well-written. Due to the theoretical nature, some parts in the main body might be hard to grasp, though the appendix has plenty of explanations and examples to answer the majority of the possible questions of a reader.

### weaknesses
* Algorithm 1 is sound. However, i) it is not complete, ii) it requires intervention targets to be known, which is not always the case (e.g., off-target effects), iii) (minor) the runtime/complexity can be expensive, though it is approximately the same order as the standard FCI algorithm.
* This is a minor weakness of the paper and more of a comment since it mainly stems from the space limitation (but also from some design choices)---all graphical descriptions (of twin augmented MAGs and $\cal I$-augmented MAGs) are deferred to the appendix. This makes it more difficult to follow the results for the non-expert reader, e.g., the discussion in line 322 about learnable structures requires Figure 1c in the appendix. I trust that some figures will be moved to the main body in the final version of the paper. The reason I also say it’s a design choice is that, in my opinion, figures would have been more useful to convey the constructions and ideas than brief experiment results in the main paper (which is important but can be just explained more briefly by referring to results in the appendix).

---

> ### Author Rebuttal · Authors · 2025-07-29
>
> **We thank the reviewer for the encouraging assessment and the detailed, constructive questions.**
>
> **Limitation on the current algorithm**
>
> The learning algorithm we proposed is incomplete and requires the knowledge of the intervention targets. We leave these problems as future extensions. The algorithm can be expensive in runtime when the graph is large. Some techniques to speed up FCI (for example [1]) transfer directly to our algorithm..
>
> **Lack of visualizations in the main text**
>
> Due to the page limitation, we have to move the figures of examples with detailed explanation to the appendix. We will put some of them in the main text for the revision to better convey the concepts. Specifically, we will move Figure 1 in App. D on a detailed example of how to construct the twin augmented MAGs from the ADMGs to Section 4. We will also move Figure 2 in App. E on a detailed example of how to construct an $\mathcal{I}$-augmented MAG by combining the twin augmented MAGs and how it preserves the desired separations, to Section 5.
>
> **Clarification on hard interventions**
>
> The results are still valid for stochastic hard interventions. In deterministic interventions there is no randomness left in $X$, so conditioning on $X$ is harmless. In stochastic interventions $X$ does carry randomness, and putting it in the conditioning set may open collider paths (or block active paths), changing the independence structure. Therefore, the intervention target being in the conditioning set in d-separation statements should not be done if it is stochastic. We do that because there is no randomness left in the node to establish dependencies.
>
> **Necessity of augmented pair graphs**
>
> We start by introducing the generalized do-calculus rules under hard interventions. The rules map testable distributional invariances into graphical constraints which can be used for discovery. The augmented pair graph shows how these invariance statements are preserved when we are comparing two domains, especially for those statements across domains. Based on the augmented pair graphs, we further construct twin augmented MAGs and then put them together in a domain-centric $\mathcal{I}$-augmented MAG as the ultimate graphical representation. The $\mathcal{I}$-MAG in [3] integrates all domains into one graph. This is not applicable for soft interventions that does not modify the original causal graph. In our case, since hard interventions will modify the graphs, we have to use the domain-centric graphs to capture different graph skeletons in different domains. Therefore, while the $\mathcal{I}$-augmented MAGs are the ultimate graphical representation we use to characterize the equivalence class, the augmented pair graph is the first step of the construction of $\mathcal{I}$-augmented MAGs which differs from the case in [3] as the invariances statements under soft interventions can be preserved in one graph. This can also be seen from the set of do-calculus rules under soft interventions ([2]) which differs from the rules under hard interventions.
>
> **Experiments on learning small graphs**
>
> The experiments are focusing on empirically compare the size of $\mathcal{I}$-MEC under hard and soft interventions to show that hard interventions are, in general, more powerful than soft interventions in learning the causal graphs. The reason we did not **compare** with other methods that allows interventional data is that the output types vary and there is no unified global *accuracy* metrics that can measure the performance. For our learning algorithm, it returns a set of $\mathcal{I}$-augmented graphs, which represents a set of ADMGs. To witness, given an ADMG, we can construct the twin augmented MAGs and further $\mathcal{I}$-augmented MAGs following the steps in Def. 4.5 and Def 5.1. If the oriented arrowheads or arrowtails align with $\mathcal{I}$-augmented MAGs, this ADMG is entailed by the algorithm's outputs.
>
> I-FCI [3] (this is named by some follow-ups) or $\psi$-FCI [4] returns a single augmented graph called I-PAG. Our experiments in App. G implicitly shows that Algorithm 1 performs better than them given that Algorithm 1's outputs entail a smaller size of $\mathcal{I}$-Markov equivalent ADMGs. Other interventional causal discovery methods aim at returning one single graph. Such methods (for example, [6], [7]) often impose parametric assumptions on the SEMs like linear causal model or additive Gaussian noise and so on. Such methods degrade sharply when the assumptions do not hold. Additionally, they are not compatible with latents. Their output graph is not guaranteed to fall into the $\mathcal{I}$-MEC either. Due to the various output types, we did not **compare** the **performance** of Algorithm 1 with other methods. We will add some small scale experiments to show how our learning algorithm works in the revision.
>
> **Faithfulness assumption**
>
> We recognize that there are weaker faithfulness assumptions. However, the soundness of Algorithm 1 requires h-faithfulness. The $\mathcal{I}$-MEC we defined is based on the generalized do-calculus rules and so as the learning algorithm and hence h-faithfulness is minimal for the algorithm to be sound.
>
> Like classic faithfulness, the set of parameterizations that *violate* h-faithfulness has Lebesgue measure zero for standard SEM families with continuous noise if we consider linear models; i.e., it holds *generically* [5] (it also holds for many other models, see follow-ups of [5]).  It is therefore no stronger in a measure–theoretic sense than ordinary faithfulness, but it is stronger *logically*: it rules out a few additional, finely-tuned parameter combinations that would create the extra cancellations specific to our Rules 1-4—those cancellations correspond exactly to our new Rules 1–4, so dropping h-faithfulness would break soundness. We can also put it this way. Imagine a causal graph under faithfulness assumption. It could be the case that for certain conditioning values $Z=z$, two variables $X$ and $Y$ that are d-connected are independent. While faithfulness covers the case that $X$ is dependent with $Y$ given $Z$ (when the dependency metric is averaged over all $z$), h-faithfulness is similar to saying $X, Y$ are dependent given every $Z=z$. This analogy is exact if the intervened node is a source node, but we believe it gives some insight on the leap from observational faithfulness to h-faithfulness
>
> **Name of MEC**
>
> We will adopt a clearer term like $\mathcal{I}$**-Hard-MEC** and mention its relation to prior “MEC” work in the revision.
>
> **Soft intervention remark**
>
> We will correct the sentence to: “A *typical* soft intervention leaves the graph unchanged, but special cases (e.g., edge deletion) are possible; our results do not rely on such deletions.”
>
> **Again, thank you for the positive evaluation and the helpful suggestions. We hope that our rebuttal has clarified the reviewer's concerns, and we would be more than happy to engage in further discussions if the reviewer has additional questions.**
>
> References:
>
> [1] Colombo, Diego, Marloes H. Maathuis, Markus Kalisch, and Thomas S. Richardson. "Learning high-dimensional directed acyclic graphs with latent and selection variables." The Annals of Statistics (2012): 294-321.
>
> [2] Correa, Juan, and Elias Bareinboim. "A calculus for stochastic interventions: Causal effect identification and surrogate experiments." In Proceedings of the AAAI conference on artificial intelligence, vol. 34, no. 06, pp. 10093-10100. 2020.
>
> [3] Kocaoglu, Murat, Amin Jaber, Karthikeyan Shanmugam, and Elias Bareinboim. "Characterization and learning of causal graphs with latent variables from soft interventions." Advances in neural information processing systems 32 (2019).
>
> [4] Jaber, Amin, Murat Kocaoglu, Karthikeyan Shanmugam, and Elias Bareinboim. "Causal discovery from soft interventions with unknown targets: Characterization and learning." Advances in neural information processing systems 33 (2020): 9551-9561.
>
> [5] Robins, James M., Richard Scheines, Peter Spirtes, and Larry Wasserman. "Uniform consistency in causal inference." Biometrika 90, no. 3 (2003): 491-515.
>
> [6] Hauser, A. & Bühlmann, P. “Characterization and Greedy Learning of Interventional Markov Equivalence Classes of DAGs.” JMLR 13: 2409 – 2464, 2012.
>
> [7] Wang, Y., Solus, L., Yang, K. & Uhler, C. “Permutation-based Causal Inference Algorithms with Interventions.” NeurIPS 2017.

---

> > ### Comment · Reviewer_QBQf · 2025-08-02
> >
> > I thank the authors for their well-rounded rebuttal, it answers my main questions.
> >
> > I have also read the other reviews and authors’ responses. I think the “Background and our contribution” walk-through response to Reviewer 4BQV is a good summary for improving the exposition of the paper. I trust that with the promised changes—these explanations and adding some figures to the main text— the revised paper will be an easier read, and will showcase the importance and contributions better (which I already appreciate, as reflected in my positive view).
> >
> > I maintain my positive score.

---

> > > ### Author Response · Authors · 2025-08-03
> > >
> > > Thanks! We really appreciate your recognition of our work!

---

### Official Review · Reviewer_4BQV · 2025-07-04

**Clarity:** 1
**Significance:** 3
**Originality:** 3
**Rating:** 4
**Confidence:** 2

**Summary:**

The paper discusses the problem of learning a causal graph from interventional data. In particular, the authors consider the scenario where multiple datasets — each produced by a different hard intervention — are available to practitioners. The authors claim to generalize do-calculus and propose a novel causal discovery algorithm, of which they prove the soundness.

**Questions:**

- In Proposition 3.2, what do you mean by "tuple of hard-interventional distributions"? Is it a set of different interventions or is it a unique intervention $\mathbf{I}\in\mathcal{I}$ as it seems from your notation? If it is a unique intervention, how does it differ from Theorem 3.1? Otherwise, if it is a set of interventions, does the rule hold on all interventional distributions or are you considering the distribution where all interventions are performed?
- Why has the method not been compared with other existing interventional causal discovery frameworks? For instance, I-FCI, psi-FCI, JCI, GIES, IGSP. It would help my evaluation if the authors could expand on this.

**Ethical Concerns:**

["NO or VERY MINOR ethics concerns only"]

**Final Justification:**

The authors acknowledged the significant presentation problems in the paper. Already in the rebuttal, they improved the discussion of their theoretical results. Together with the aspects mentioned in the rebuttal of reviewer QBQf, the next revision of the paper should be sufficiently readable. The rebuttal convinced me of the novelty of their work.

**Limitations:**

yes

**Quality:**

3

**Strengths And Weaknesses:**

Strengths:
- To the best of my knowledge, the introduced algorithm is novel, as it builds on a novel and more general characterization of do-calculus introduced by the authors in this work.

Weaknesses:
- I had problems understanding the core points of the paper. Theoretical results in Sections 3, 4, and 5 lack an intuitive discussion or examples. For instance, a graphical representation of the concepts of augmented pair graphs and twin augmented MAGs could have clarified a quite heavy-to-read notation. To be entirely honest, do-calculus is not one of the core aspects of my expertise, and I am consequently adjusting my confidence with a low score.
- Related works reference several existing techniques for interventional and observational causal discovery. However, the authors only discuss the relation between their proposed method and FCI. The experimental section does not compare their method with any existing method.

---

> ### Author Rebuttal · Authors · 2025-07-30
>
> **We thank the reviewer for the helpful comments and apologize for the parts that were hard to follow.**
>
> **Clarification of the main paper**
>
> We agree the exposition can be improved. Due to the page limitation, we had to move some figures that illustrate the concepts to the appendix which made the main text less readable. In the revision we will: 1) move Figure 1 in App. D to Section 4 to illustrate the  construction of **twin-augmented MAG** and highlighting the extra edge that captures the hard-intervention constraints; 2) move Figure 2 to Section 5 to show how we construct $\mathcal{I}$-augmented MAGs by combining twin-augmented MAGs. A short narrative paragraph will precede each theorem to walk the reader through the intuition before the formal statement. We conclude the main idea of this paper below.
>
>  **Background and our contribution:**
>
> Learning a causal graph from **multiple hard-intervention distributions** has two pillars:
>
> (i) *do-calculus*, which maps cross-environment invariance statements to graphical separations;
>
> (ii) *equivalence-class characterizations*, which say **exactly** what can be learned from those invariances.
>
> For observational data this yields the well-known **Markov Equivalence Class (MEC)**.
>
> For interventional data, Hauser & Bühlmann [1] extended the idea to the **I-MEC**, but only for *observed* DAGs with no latents; later work (Yang et al. [2]) unified hard/soft interventions in the same fully-observed setting, and Kocaoglu ’19 [3] handled **soft** interventions with latent variables via an *augmented MAG*.
>
> Our paper closes the remaining gap: **hard interventions *with* latents**.
>
> Because a hard cut severs every parent of the target, its effect can be *global* (the dependencies between non-adjacent nodes can be affected) when hidden confounders exist (see example in Line 48). Such surgical property provides us with more testable conditional independence statements and which therefore, the augmented MAG structure in [3] fails in our setting.
>
> We therefore:
>
> 1. **Generalize do-calculus (Rules 1–4)** under hard interventions (Prop. 3.2) and define h-faithfulness (Def. 6.1) accordingly as the assumption for the learning algorithm. The generalized do-calculus rules map the testable invariances between hard-interventional distributions to the graphical constraints on the (mutilated) graphs.
>
> 2. Build the **augmented pair graph** (Def. 4.3) and further, the **twin-augmented MAG (TAM)** (Def. 4.5) (Figure 1 in App. D shows a detailed example of how we construct TAMs from the original ADMGs through augmented pair graphs), which preserves *all testable* invariances without mutilating the original ADMG.
>
> 3. Prove **Thm 4.7**: a complete graphical characterisation of the **$\mathcal{I}$-MEC** based on the graphical structures of the TAMs—the first such result for hard interventions under latents.
>
> 4. Compress the TAMs that contain the same interventional domain into a single, lean **$\mathcal{I}$-augmented MAG** (Def 5.1) (Figure 2 in App. E shows a detailed example of how we construct the $\mathcal{I}$-augmented MAGs). The $\mathcal{I}$-augmented MAGs are more compact and preserve all the testable separation statements. The union graph of all the $\mathcal{I}$-Markov equivalent $\mathcal{I}$-augmented MAGs forms the $\mathcal{I}$-essential graphs (Def. B.8) which become the ultimate target of our discovery procedure.
>
> 5. Derive **Algorithm 1**, an FCI-style learner that outputs a tuple of $\mathcal{I}$-augmented graphs (which represents a set of valid ADMGs) using FCI orientation rules together with additional, sound orientation rules (example walk-through in App. F, Figure 3).
>
> Together, these steps establish what is fundamentally learnable under hard interventions and provide a learning algorithm to achieve it. We will improve the main text, especially introduction and related works accordingly, to improve readability.
>
> **No comparison with other interventional methods**
>
> Unlike most causal discovery algorithms, the goal of our algorithm is *not* to return a single causal graph but a tuple of $\mathcal{I}$-augmented graphs that entail **$\mathcal{I}$-Markov equivalent graphs**; none of the listed algorithms outputs that object.  Concretely, **I-FCI [3] (known targets)/$\psi$-FCI [4] (unknown targets)** assume soft interventions and cannot exploit the additional invariance constraints; applying them to our setting recovers a *larger* MEC (App. G). They use a single augmented MAG as the learning objective. **JCI** claims it works for any interventions but it's overly simplistic; the learned PAG is less informative (in [4] App. D shows that $\psi$-FCI can learn more than JCI). **GIES, IGSP** are score-based and aim at learning a *single* DAG under known targets; they neither guarantee inclusion-minimality nor provide equivalence-class certificates. Nevertheless, scores-based methods usually impose parametric assumptions like linear function or additive Gaussian noise to the causal models. Such methods are not robust when the underlying mechanism deviates from those assumptions. Additionally, GIES and IGSP assume no latents and therefore cannot return ADMGs. Because the outputs are of different types, a direct performance metric is unavailable to measure the performance of the methods; instead we reported a *size* comparison (Table 2) that quantifies empirically how much smaller the $\mathcal{I}$-MEC is vs. the soft $\mathcal{I}$-MEC produced by I-FCI/$\psi$-FCI. To compare with GIES and IGSP, we use the graph $[X\rightarrow Z \rightarrow Y, Z\leftarrow L \rightarrow Y]$ where $L$ is a latent variable. We take 100k samples from observational distribution and 100k samples from $P_Z$. For the setting where $X, Y, Z$ are binary, we generate a random CPT based on the true graph and sample from it. GIES returns $[Y\rightarrow Z \rightarrow X]$ which is completely wrong. IGSP returns $[X\rightarrow Z \rightarrow Y, X\rightarrow Y]$ which is the observational MAG of the true graph, but it cannot tell anything to judge if there is a latent confounder between $Z, Y$. Under the same setting, when the simulated dataset is from a linear Gaussian SEM, both GIES and IGSP outputs $[X\rightarrow Z \rightarrow Y]$ which catches the right causal ordering but still is wrong and does not belong to the MEC. In both settings, our algorithm is able to learn the graphs illustrated in Figure 3 (k) which preserve information from both domains. We use the python implementation of GIES by Olga Kolotuhina and Juan L. Gamella. The IGSP implementation is from the `causaldag` package.
>
> We will make this distinction explicit in an extra sub section in App. G with the detailed experiment settings. We will also add the small scale experiments to compare the outputs of each method in the revision.
>
> **Meaning of “tuple of hard-interventional distributions”**
>
> Let $ \mathcal{I} = \lbrace \mathbf{I}\_1, \ldots, \mathbf{I}\_m \rbrace $ be the set of intervention targets.
> The “tuple” is the collection $ (P^{(1)}, \ldots, P^{(m)}) $
> where $P^{(j)}$ is the interventional distribution under the intervention target $\mathbf{I}\_j$.
> Proposition 3.2 is a generalization of Theorem 3.1. More specifically, Theorem 3.1 tells under what graphical constraints one can adjust the intervention or conditioning set from a given distribution. Proposition 3.2 generalizes the rules to any pair of arbitrary interventional distributions ($P_{\mathbf{I}}$ and $P_{\mathbf{J}}$ in the main paper). It is saying that for any pair of interventional distributions, the invariance statements hold under the corresponding graphical constraints. The new set of rules make the comparisons of testable invariances explicit.
>
> **Once again, thank you for the detailed feedback. We hope that our rebuttal has clarified the reviewer's concerns, and we would be more than happy to engage in further discussions if the reviewer has additional questions.**
>
> References:
>
> [1] Hauser, Alain, and Peter Bühlmann. "Characterization and greedy learning of interventional Markov equivalence classes of directed acyclic graphs." *The Journal of Machine Learning Research* 13, no. 1 (2012): 2409-2464.
>
> [2] Yang, Karren, Abigail Katcoff, and Caroline Uhler. "Characterizing and learning equivalence classes of causal dags under interventions." In International Conference on Machine Learning, pp. 5541-5550. PMLR, 2018.
>
> [3] Kocaoglu, Murat, Amin Jaber, Karthikeyan Shanmugam, and Elias Bareinboim. "Characterization and learning of causal graphs with latent variables from soft interventions." Advances in neural information processing systems 32 (2019).
>
> [4] Jaber, Amin, Murat Kocaoglu, Karthikeyan Shanmugam, and Elias Bareinboim. "Causal discovery from soft interventions with unknown targets: Characterization and learning." Advances in neural information processing systems 33 (2020): 9551-9561.

---

> > ### Comment · Reviewer_4BQV · 2025-08-04
> >
> > I thank the authors for their detailed walk-through on the main contents of the paper.
> >
> > I still have strong concerns on the presentation of the paper, as also mentioned by other reviewers. Improving the exposition would require significantly reshaping the paper. However, the provided walk-through of the paper is sufficient to evaluate the novelty of the paper, which effectively tackles an open problem in interventional causal discovery. I particularly appreciated the comparison with other interventional based causal discovery methods, which should definitely find a place in a next revision of the work.
> >
> > Given this, I am revising my recommendation accordingly to borderline accept.

---

> > > ### Author Response · Authors · 2025-08-04
> > >
> > > Thank you for taking the time to re-evaluate our work and for raising your recommendation. We appreciate your constructive comments and will incorporate the promised revisions.

---

### Official Review · Reviewer_HFqZ · 2025-07-07

**Clarity:** 2
**Significance:** 3
**Originality:** 2
**Rating:** 3
**Confidence:** 4

**Summary:**

The paper considers interventional equivalence class to the setting where hard interventions are available, in the presence of unobservables.  The core contributions of the paper are:

1) generalized do-calculus that relates pair of interventional distributions
2) Twin-Augmented MAG (TAM) that embeds a pair of intervention graphs with F-nodes
3) A graphical characterization of I-Markov equivalence class via TAM
4) Compression of k TAMs which preserves do-calculus constraints
5) An FCI-like discovery algorithm with new orientation rules

**Questions:**

“Revisiting the General Identifiability Problem” (Kivva et al., 2022) and “Minimum Intervention Cover of a Causal Graph” (Kandasamy et al., 2019) are examples of papers that derive constraints across multiple interventions using do-calculus (where the graph is given). It is unclear whether the TAM invariances introduce new algebraic equalities or re-express known identifiability constraints graphically. I understand the problem statements are different (they are inverse problems in some sense), but very closely related.

Since the paper deals with invariances of do-calculus across interventional distributions, besides the h-faithfulness requirement does the algorithm also needs positivity requirement?

Numerous inconsistencies suggest the manuscript was rushed.  Some core concepts are used before being defined.  I request the authors to proof-read the manuscript thoroughly.

Line 49: The bidrected edge is used before defining it.

Line 129: Inducing path is used before definition.  Does ancestor include the vertex?  I assume it is inclusive but it should be clear.

Line 131: What is "almost directed cycle"? -- please define it in the main paper or point out to the right section in Appendix.

Line 179:  In Rule 2, conditioning on z is not necessary after intervention on z.  It is redundant as z is already in intervention.

Line 197/199:  A minor suggestion:  Equate them to $P_{ \mathbf{I} \cup \mathbf{J}} (\mathbf{y} | \mathbf{w})$ / $\mathbf{P}_{ \mathbf{I} \cap \mathbf{J}} (\mathbf{y} | \mathbf{w})$.  I feel the proof maybe easy to derive with this addition.

Line 217: R_I, R_J, K_I, K_J first appear inside Proposition 3.2 yet is reused later. Move them to a global definition.

Line 229: "We introduce the augmented pair". -- It appears a new concept is being introduced, though is rightly pointed out that the notion of augmented pair and F-nodes is known in the literature.

Line 232: Superscripts (I) (J) are used without explanation.

Line 235: K should be in bold.

Line 258: Please add a short explanation regarding how to construct the MAG.  This would improve the flow.

Line 259:  MAG and Twin are in math mode.  Please use \mathrm or appropriate styling for all such text/abbrevation inside math mode.

Line 260:  The definition of S is confusing.  I understand S is defined inside Augmented pair definition but S is undefined here.

Line 263:  In $Y^{(\mathbf{I})}$ what is Y?

**Ethical Concerns:**

["NO or VERY MINOR ethics concerns only"]

**Final Justification:**

The rebuttal and discussions clarified some of my technical concerns, and I considered upgrading my score. However, I still found the manuscript difficult to read and believe it requires substantial revisions. Therefore, I will maintain my original score.

**Limitations:**

Yes.

**Paper Formatting Concerns:**

Nil.

**Quality:**

2

**Strengths And Weaknesses:**

The paper provides the first I-Markov equivalence characterization for hard interventions in the presence of unobservables.  The paper is well-structured starting with a modified version of do-calculus, to equivalence characterization, to compression of I-MAG, and finishing with a learning algorithm.  Proofs are detailed and appear sound.  Although the experiments are small-scale and synthetic (which is probably unavoidable), they quantify a clear reduction in equivalence-class size when moving from soft to hard interventions, illustrating noticeable benefit.

Weaknesses:
The contribution feels incremental: apart from the twin-augmented MAG, most machinery (augmented graphs, orientation rules, do-calculus derivations) already exists in the literature.  The hard intervention with latents is an important problem but the conceptual leap in my opinion is modest when viewed against two decades of equivalence class research.

---

> ### Author Rebuttal · Authors · 2025-07-29
>
> Thank you for the thoughtful assessment and the detailed style notes.
>
> **Novelty based on existing machineries**
>
> Prior augmented-graph formalisms ([1], [2], [6], [7]) cover **soft** interventions only or assume no latents.  Hard interventions add the *surgical* constraint “no incoming edges”, which induces **new cross-environment distributional invariances** (Proposition 3.2) that cannot be captured by those graphs. Nevertheless, the presence of latents would create non-local dependencies and can be affected by hard interventions on remote nodes. Our twin augmented MAG (TAM) embeds these additional constraints via the $F$ node edges and is thus more expressive than prior augmented graphs. This extension is what allows us to prove the first graphical characterisation of an $\mathcal{I}$-MEC under hard interventions with latents —- a result that did not exist.
>
> **Additional related works**
>
> Both papers assume the *graph is known* and derive algebraic constraints to decide the identifiability of *parameters (if interventional distributions can be computed from the observational distribution)* (Kivva [4]) or a minimum intervention *cover* (Kandasamy [5]). Our setting is like the **inverse** problem: our Twin-Augmented MAG translates cross-environment distributional invariance statements into d-separations without assuming the graph. These invariances are then used to (i) characterise the $\mathcal{I}$-MEC (Thm 4.7) with a compact graphical representation and (ii) drive Algorithm 1 for structure learning based on the graphical constraints corresponding to the invariances. Thus the two lines of work are complementary: one answers ‘Which causal effects are computable if the graph is known?’; the other answers ‘Which graphs are compatible with the data across multiple hard-intervention settings?’ We will include these additional related works and add this contrast at the end of Section 2.
>
> **Positivity assumption for the learning algorithm**
>
> In theory, Algorithm 1 does not require a strict global positivity assumption. All results rely solely on h-faithfulness. We are aware that  “strictly positive distribution” is added in some works on discovery as a convenience. This is because *within the space of strictly positive parameterisations* the subset that violates Faithfulness sits on algebraic varieties of lower dimension—hence has Lebesgue measure 0 (see [3] and many follow-ups). In practice, with finite data, a zero or near-zero cell makes test statistics undefined or unstable, so implementations often impose minimum-count rules even when the theory would allow the zero.
>
> We notice that the positivity assumption is required for identifiability (For example, [4], [5]). This is because that the manipulation of numerical expression explicitly require the denominator to be strictly positive. By contrast, structural discovery uses only qualitative independence relations. As long as we can evaluate a test where $P(S)>0$, no global support assumption is required.
>
> **Style/Notation issues**
>
> We will systematic­ally reorder definitions so every concept precedes first use for bidirected edge, inducing path, ancestor, and almost directed cycle. We will also fix any typos in the revision.
>
> For the conditioning set in Rule 2, Line 179, we add $Z$ explicitly due to its clear connection with the corresponding $F$ node d-separation statements, which require conditioning although it is not necessary. We will add a clarifying comment in the revised manuscript.
>
> In line 197/199, we will take the suggestion to add $P_{ \mathbf{I} \cup \mathbf{J} }(\mathbf{y}|\mathbf{w})$ to the equations to make it easier to understand. In the proof, we also show that using $P_{ \mathbf{I} \cup \mathbf{J} }$ as the intermediate distribution is both necessary and sufficient to derive the graphical constraint.
>
> We will make ${\mathbf{R}_{\mathbf{I}}}$ and so on a global definition to avoid repeated definitions.
>
> We will use `\mathrm` for notation of MAGs and twin augmented graphs.
>
> For the definition of Augmented pair graph, we will add a sentence to indicate that $F^{(\mathbf{I}, \mathbf{J} )}$ is an auxiliary node for a pair of targets $\mathbf{I}, \mathbf{J}$. The superscript shows the targets. We omit the superscript when it is clear from the context which pair of targets we are looking at.
>
> For the definition of twin augmented MAG, we will add a short sentence in Section 2 where we describe maximal ancestral graphs, on how to construct a MAG over an ADMG. In Line 260, we will replace $S$ with $V\in \mathbf{V}$ as a node such that $V^{(\mathbf{I})}$ or $V^{(\mathbf{J})}$ is adjacent to $F^{(\mathbf{I}, \mathbf{J} )}$ in $\mathrm{MAG}(Aug_{ (\mathbf{I}, \mathbf{J}) }(\mathcal{D}))$.
>
> In Line 263, $Y$ is any node in $\mathbf{V}$. We intended to clarify that any separation statement like $F \mathrel{\perp\mspace{-10mu}\perp} Y^{(\mathbf{I})}$ for $Y \in \mathbf{V}$ is not directly testable from comparing the CIs using just $P_{\mathbf{I}}, P_{\mathbf{J}}$. We can only decide this separation to be true when both $F \mathrel{\perp\mspace{-10mu}\perp} Y^{(\mathbf{I})}$ and $F \mathrel{\perp\mspace{-10mu}\perp} Y^{(\mathbf{J})}$ hold and that motivates our construction of Twin augmented MAGs by adding extra edges to the $F$ node.
>
> **MAG construction example**
>
> Section 4 will gain an example (figure now in App. D) showing how we construct twin augmented MAGs starting from an ADMG.
> Section 5 will gain an example (figure now in App. E) showing how we compress twin augmented MAGs to an $\mathcal{I}$-augmented MAG. We believe this, together with the reordered definitions, will substantially improve readability.
>
> **We appreciate the constructive feedback. We hope that our rebuttal has clarified the reviewer's concerns, and we would be more than happy to engage in further discussions if the reviewer has additional questions.**
>
> References:
>
> [1] Eaton, Daniel, and Kevin Murphy. "Belief net structure learning from uncertain interventions." *Journal of Machine Learning Research* 1 (2007): 1-48.
>
> [2] Pearl, Judea. "Models, reasoning and inference." Cambridge, UK: CambridgeUniversityPress 19, no. 2 (2000): 3.
>
> [3] Robins, James M., Richard Scheines, Peter Spirtes, and Larry Wasserman. "Uniform consistency in causal inference." *Biometrika* 90, no. 3 (2003): 491-515.
>
> [4] Kivva, Yaroslav, Ehsan Mokhtarian, Jalal Etesami, and Negar Kiyavash. "Revisiting the general identifiability problem." In Uncertainty in Artificial Intelligence, pp. 1022-1030. PMLR, 2022.
>
> [5] Kandasamy, Saravanan, Arnab Bhattacharyya, and Vasant G. Honavar. "Minimum intervention cover of a causal graph." In Proceedings of the AAAI Conference on Artificial Intelligence, vol. 33, no. 01, pp. 2876-2885. 2019.
>
> [6] Yang, Karren, Abigail Katcoff, and Caroline Uhler. "Characterizing and learning equivalence classes of causal dags under interventions." In International Conference on Machine Learning, pp. 5541-5550. PMLR, 2018.
>
> [7] Kocaoglu, Murat, Amin Jaber, Karthikeyan Shanmugam, and Elias Bareinboim. "Characterization and learning of causal graphs with latent variables from soft interventions." Advances in neural information processing systems 32 (2019).

---

> ### Comment · Reviewer_HFqZ · 2025-08-04
>
> Dear Authors,
>
> Thank you for your response.
>
> With respect to novelty, I understand the difference between hard and soft interventions -- My concern was mainly on the technical novelty in the proof techniques.  I did read the main paper and briefly went through Appendix, and I didn't find the technical leaps sufficient for the standards of NeurIPS.  I understand this is a subjective concern, unfortunately there is no easy way to objectify this.
>
> Also, with respect to existing literature on identifiability, I understand the differences between the problem statements.  I was mainly curious in understanding the overlaps in techniques (I believe there should be a significant overlap).

---

> > ### Author Response · Authors · 2025-08-04
> > **Reply to Reviewer HFqZ (Part 1 of 2)**
> >
> > Thank you for your interest in our paper's technical contribution. Below we explain in detail how our proof tools differ from the previous works:
> >
> > How our setting differs from soft interventions in proof techniques:
> >
> > The most relevant work is Kocaoglu 19 [1] which explores soft interventional $\mathcal{I}$-MEC. Soft interventions leave the graph intact, so [1] can transport any intermediate distribution between $P_I$ and $P_J$ with the ordinary three do-calculus rules to establish the generalized do-calculus rules.
> >
> > The transition from $P_I$ to $P_J$ requires going through an intermediate distribution that uses the set difference between $I$ and $J$. In the case of hard interventions, when the incoming edges of the targets in $I$ and $J$ are surgically cut, this previous set–difference distribution can no longer be used to relate $P_I$ and $P_J$. Proposition 3.2 proves that the minimal and only valid bridge is the union distribution $P_{I\cup J}$ instead. Finding this right in-between distribution to establish equality is one of our technical contributions. Below are more details on why this is sound.
> >
> > If we intervene on a larger set $I \cup J \cup S$ for some $S\neq \emptyset$, we remove the edges into $S$ while conditioning on it. The structural difference between using $I \cup J \cup S$ and $I \cup J$ is the set of edges outgoing from $S$ if we look at Rule 2 for example. We show that these edges do not construct any d-connecting path if using $I\cup J$. Hence any conditioning set $W$ that d-separates under $P_{I\cup J\cup S}$ automatically implies d-separations under $P_{I\cup J}$. The similar technique is also used for Rule 3 and Rule 4.
> >
> > Conversely, once $P_{I \cup J}$ satisfies the pair of separations, adding further cuts in $S$ merely cuts parents that are already absent; the equalities remain true. Thus using a larger intervention target for the in-between distribution yields strictly stronger invariances, and thus $P_{I \cup J}$ is sufficient. In short, any intermediary that intervenes on more than $I \cup J$ collapses back to the invariances implied by $P_{I \cup J}$, while smaller choices may admit spurious graphs.
> > This union rule has no analogue in [1], where the graph is unchanged and $P_{I \cap J}$ remains valid.
> >
> > Because hard cuts create invariances that are AND-statements across two different mutilated graphs, a single PAG cannot host them. We therefore introduce the Twin-Augmented MAG (TAM), duplicating every observed node and attaching exactly one twin of each pair to the context node $F$. Soft-intervention proofs never require such a construction.
> >
> > Another technical contribution:
> >
> > The “only-if’’ half (if the two twin MAGs do not fully obey the 3 conditions, there exists testable invariance statements) of Theorem 4.7 needs a new twin blocking lemma to map the non-adjacencies between $F$ nodes and other nodes to testable invariance statements. The non-adjacency between $F$ and $X$ in the domain of $I$ only indicates that $F$ is separable from $X$ given some $W_1$ in domain of $I$. This statement alone is not testable as Proposition 4.4 requires an AND statement in both domains. The lemma tells us the if we have $F$ non-adjacent to $X$ in both domains, i.e. $F\perp\mspace{-10mu}\perp X $ given some minimal $W_1$ in domain $I$ and $F \perp\mspace{-10mu}\perp X$ given some minimal $W_2$ in domain $J$, there exists $W = W_1 \cup W_2$ that separates $F$ and $X$ in both domains without triggering any extra paths. This is used to construct testable invariance statements across the interventional distributions $P_I$ and $P_J$. This argument is not needed in the soft-intervention works as all the domains have the same structure.
> >
> > This observation is critically used when justifying for unshielded colliders and collider status in discriminating paths as well.

---

> > > ### Author Response · Authors · 2025-08-04
> > > **Reply to Reviewer HFqZ (Part 2 of 2)**
> > >
> > > 2. Why our techiques do not overlap with those in the identifiability (ID) problems:
> > >
> > > Identifiability papers assume the causal graph is known and ask whether a symbolic causal effect, e.g. $P(Y| do(X))$, can be expressed in terms of $P(V)$ and a fixed list of interventional distributions.
> > >
> > > Hedge / C-component machinery ([2], [3], [4], [5]) works entirely inside the given graph: it recursively decomposes into latent bidirected components; if a "hedge" subgraph appears the effect is declared non-identifiable. That reasoning needs the true bidirected structure up-front and is impossible when the graph itself is the unknown we wish to discover, like in our setting. We focus on deriving if there can be any d-connecting paths between two sets of nodes given a conditioning set under different graph mutilations, which would otherwise be directly known given the graph like in the ID settings.
> > >
> > > [6], [7] and so on focus on algebraic derivations. Specifically, [6] casts identifiability as polynomial ideal membership and builds Gröbner bases to eliminate hidden kernel parameters. This algebra presupposes that every parent set is known and never orients edges; it manipulates kernels like $\sum_z P(y∣z,x)P(z)$ not cross-environment conditional-independence constraints.
> > >
> > > Our task is different: start with a fully unoriented graph, decide which arrowheads/arrowtails are forced by the empirical (cross-domain) CI/equality constraints induced by hard cuts. Algebraic substitutions or hedge arguments cannot formulate or resolve that problem. We therefore need the new hard-do calculus rules, the $\mathcal{I}$-augmented MAG representation and the proof techiques mentioned above. None of the techniques from the ID works are directly useful in our case, to the best of our knowledge.
> > >
> > > We hope this clarifies why our proofs require techniques not present in the discovery or identifiability literature. We appreciate your careful reading and are happy to address any further questions.
> > >
> > > References:
> > >
> > > [1] Kocaoglu, Murat, Amin Jaber, Karthikeyan Shanmugam, and Elias Bareinboim. "Characterization and learning of causal graphs with latent variables from soft interventions." Advances in neural information processing systems 32 (2019).
> > >
> > > [2] Tian, Jin, and Judea Pearl. "A general identification condition for causal effects." In Aaai/iaai, pp. 567-573. 2002.
> > >
> > > [3] Shpitser, Ilya, and Judea Pearl. "Identification of conditional interventional distributions." arXiv preprint arXiv:1206.6876 (2012).
> > >
> > > [4] Shpitser, Ilya, and Judea Pearl. "Complete identification methods for the causal hierarchy." (2008).
> > >
> > > [5] Kandasamy, Saravanan, Arnab Bhattacharyya, and Vasant G. Honavar. "Minimum intervention cover of a causal graph." In Proceedings of the AAAI Conference on Artificial Intelligence, vol. 33, no. 01, pp. 2876-2885. 2019.
> > >
> > > [6] Kivva, Yaroslav, Ehsan Mokhtarian, Jalal Etesami, and Negar Kiyavash. "Revisiting the general identifiability problem." In Uncertainty in Artificial Intelligence, pp. 1022-1030. PMLR, 2022.
> > >
> > > [7] Karvanen, Juha, Santtu Tikka, and Antti Hyttinen. "Do-search: a tool for causal inference and study design with multiple data sources." Epidemiology 32, no. 1 (2021): 111-119.

---

> > > > ### Author Response · Authors · 2025-08-07
> > > > **Follow up for Reviewer HFqZ**
> > > >
> > > > Dear Reviewer HFqZ,
> > > >
> > > > We would like to kindly ask whether our response has addressed your concerns. In particular, we tried to carefully address your comments on the related references on identifiability and highlighted the key differences in the problem statements and proof techniques to clarify our novelty and contribution. We sincerely hope that our clarifications were helpful in resolving your concerns.
> > > >
> > > > We truly appreciate your feedback and look forward to your response.
> > > >
> > > > Best,
> > > >
> > > > Authors

---

> > > > > ### Comment · Reviewer_HFqZ · 2025-08-07
> > > > >
> > > > > Dear Authors,
> > > > >
> > > > > Please give me a day or two, I would like to think a little bit.  I am sorry about the delay.  Thank you.

---

### Author Response · Authors · 2025-08-09
**Final Thoughts**

We thank the reviewers and AC for a very constructive discussion. We carefully considered every point raised and clarified several technical aspects during the rebuttal (hard-vs-soft intervention rules, stochastic hard interventions, the role of our pair graph construction and so on). Below we summarise our contributions, how we addressed concerns, and what we will revise.

**Contributions:**

Our paper addresses the open problem of **characterizing and learning with multiple hard-interventional distributions in the presence of latents**. Concretely, we list our contributions as follows:

1. We derive a **generalized do-calculus** (Prop. 3.2) for two arbitrary hard-interventional distributions. We show that the **union bridge** $P_{I\cup J}$ is the unique intermediary for cross-domain transport;
2. We construct the **Twin-Augmented MAG (TAM)** that embeds all testable distributional invariances, especially cross-environment invariances (which are AND-statements across mutilated graphs) in a single graph;
3. Based on TAM, we propose the first sound and complete **graphical characterization of the $\mathcal{I}$-MEC under hard interventions with latents** (Thm. 4.7);
4. We construct a **compression** from multiple TAMs to one **$\mathcal{I}$-augmented MAG** that preserves all testable invariances;
5. We introduce a **sound FCI-style discovery algorithm** with new orientation rules tied to hard-cut surgery. We prove the soundness of our algorithm (Thm. 6.3);
6. We conduct experiments on small graphs to show that hard $\mathcal{I}$-MEC size is in general smaller than soft $\mathcal{I}$-MEC size which justifies our intuition that hard interventions usually carry more information than soft interventions.

**Reviewer concerns and how we addressed them:**

- **Technical overlap & novelty (HFqZ):** We clarified how our proof technique differs from the most related work [1] which deals with soft interventions. One of the key differences is using the union-bridge as the unique intermediate distribution for the generalized do-calculus. Additionally, to prove the if-and-only-if graphical characterization of $\mathcal{I}$-MEC, we introduce new techniques to find testable invariance statements when the 3 conditions do not hold accordingly. These are not concerns for soft interventions since the graphical structures are not modified. We also clarified how our work differs from identifiability works in both problem statements and proof techniques.
- **Clarity, definitions before use, notation (HFqZ, 4BQV):** We will **reorder definitions** (bidirected edges, inducing paths, almost-directed cycles), introduce **global notation** ($R_I, K_I \ldots$), use $\mathrm{MAG}$ styling in math mode, and add **short construction notes** for MAGs.
- **Exposition & figures (4BQV, QBQf):** We will **move the TAM construction figure** (App. D) to Section 4 and the **$\mathcal{I}$-augmented MAG compression figure** (App. E) to Section 5, each with an intuition paragraph.
- **Comparisons & experiments (4BQV/QBQf):** We will add **small-scale learning experiments** and an explicit comparison section discussing I-FCI/$\psi$-FCI/JCI/GIES/IGSP and why outputs differ ($\mathcal{I}$-augmented graphs vs single PAG/DAG/augmented MAG), plus the empirical **$\mathcal{I}$-MEC size** reductions already shown.
- **Assumptions (positivity, h-faithfulness) (HFqZ, QBQf):** We clarified **no global positivity** is required in theory (only h-faithfulness) for our learning algorithm; we will state practical safeguards for finite samples. We also clarified that **stochastic hard interventions** still follow our analysis but under stochastic hard interventions, one should not place them in the conditioning set for do-calculus. We clarified the comparison between h-faithfulness and observational faithfulness.

**Revisions we will make:**
- Fix typos and unify notation (MAG styling, consistent $F$-node/context usage). Ensure that definitions precede uses.
- Rename our $\mathcal{I}$-MEC to distinguish it from other settings.
- Add a short clarification for stochastic hard interventions and soft interventions that do not change the graphical structure.
- Move the **TAM construction** and $\mathcal{I}$**-augmented MAG compression** figures into the main text with short intuition paragraphs.
- Expand **related work** to explicitly compare with previous works on identifiability.
- Include the **additional experiments and baselines with a discussion on the outputs to the appendix.**

We believe these changes will further clarify scope and significance and make the paper easier to read and verify. Thank you again for the thoughtful feedback and for the opportunity to improve our work.

References:

[1] Kocaoglu, Murat, Amin Jaber, Karthikeyan Shanmugam, and Elias Bareinboim. "Characterization and learning of causal graphs with latent variables from soft interventions." Advances in neural information processing systems 32 (2019).

---

### Decision · Program_Chairs · 2025-09-17

**Decision:**

Accept (poster)

**Comment:**

This paper makes a notable contribution by characterizing interventional Markov equivalence under hard interventions with latent variables, a problem not previously resolved. The authors introduce the Twin-Augmented MAG (TAM) and I-augmented MAG, establish a complete graphical characterization of the I-MEC in this setting, and develop a sound discovery algorithm with new orientation rules. Empirical evidence further shows that hard interventions yield smaller equivalence classes than soft interventions. Together, these results fill an important gap and are technically novel, rigorous, and well-motivated.

The main weakness concerns clarity and accessibility of presentation. The manuscript is notation-heavy, introduces definitions after use, and lacks intuitive figures. In addition, as Reviewer HFqZ noted, the technical advances may appear incremental without a clearer exposition of the proof techniques and their distinction from prior work. I strongly encourage the authors to follow through on their promised revisions: moving key figures to the main text, reordering definitions, expanding the related work discussion to highlight novelty, and improving the overall exposition. Addressing these points will substantially enhance readability and better convey the significance of the contribution.